# How Realistic Is Your Synthetic Data? Constraining Deep Generative Models for Tabular Data

**Mihaela Cătălina Stoian**[*, 1]**, Salijona Dyrmishi**[*, 2]**, Maxime Cordy**[2]**,**
**Thomas Lukasiewicz**[3, 1] **, Eleonora Giunchiglia**[3]
[1]University of Oxford, [2]University of Luxembourg, [3]Vienna University of Technology

## Abstract

Deep Generative Models (DGMs) have been shown to be powerful tools for generating tabular data, as they have been increasingly able to capture the complex distributions that characterize them. However, to generate realistic synthetic data, it is often not enough to have a good approximation of their distribution, as it also requires compliance with constraints that encode essential background knowledge on the problem at hand. In this paper, we address this limitation and show how DGMs for tabular data can be transformed into Constrained Deep Generative Models (C-DGMs), whose generated samples are guaranteed to be compliant with the given constraints. This is achieved by automatically parsing the constraints and transforming them into a Constraint Layer (CL) seamlessly integrated with the DGM. Our extensive experimental analysis with various DGMs and tasks reveals that standard DGMs often violate constraints, some exceeding 95% non-compliance, while their corresponding C-DGMs are never non-compliant. Then, we quantitatively demonstrate that, at training time, C-DGMs are able to exploit the background knowledge expressed by the constraints to outperform their standard counterparts with up to 6.5% improvement in utility and detection. Further, we show how our CL does not necessarily need to be integrated at training time, as it can be also used as a guardrail at inference time, still producing some improvements in the overall performance of the models. Finally, we show that our CL does not hinder the sample generation time of the models.

## 1 Introduction

Synthetic data generation represents an important area of machine learning (ML) due to its numerous applications in the real world. Indeed, synthetic data have been increasingly used to augment real data to improve the predictive performance of ML models (see, e.g., Han et al. (2005)), to remedy data scarcity (see, e.g., Choi et al. (2017)), and to promote fairness (see, e.g., van Breugel et al. (2021)), and they are now even used to generate brand-new datasets to ensure privacy in sensitive settings (see, e.g., Jordon et al. (2019); Yoon et al. (2020); Lee et al. (2021)).

Deep Generative Models (DGMs) have been shown to be powerful tools for generating tabular data, as they have been progressively more able to capture the complex distributions that characterize such data (see, e.g., Kim et al. (2023)). However, for generating realistic tabular data, it is insufficient to just learn a good distribution approximation; it is necessary to create samples that obey a set of constraints expressing background knowledge about known characteristics of the features and/or existing relationships among them. For example, if we are generating data from a clinical trial dataset, then, for every synthetic sample, we surely want the value associated with the *"maximum level of hemoglobin recorded"* column to be greater than or equal to the one associated with the *"minimum level of hemoglobin recorded"* column. Indeed, any sample violating such constraint is not realistic. Existing methods, while excellent at capturing complex distributions, are still not able to learn even from this simple background knowledge, and thus do not provide any guarantee of constraint satisfaction. In addition, currently, no work incorporates background knowledge in

---

*Equal contribution.

DGMs for tabular data except for GOGGLE (Liu et al., 2022), which, however, is only able to inject very simple background knowledge about existing correlations among features.

In this paper, we address this limitation, and we show how to integrate any background knowledge that can be expressed as a set of linear inequalities into different standard DGMs for tabular data. To this end, we introduce a novel approach able to transform different DGMs into corresponding Constrained Deep Generative Models (C-DGMs), i.e., DGMs whose generated samples are guaranteed to be compliant with the set of user-defined constraints. Our method takes as input linear inequality constraints and a DGM, and then automatically parses the constraints and generates a differentiable Constraint Layer (CL) that can be seamlessly integrated with the DGM. Our method thus returns a novel model, the C-DGM, whose sample space is guaranteed to be compliant with the constraints. To evaluate our approach, we conduct an extensive experimental analysis that involves testing Wasserstein GAN (WGAN) (Arjovsky et al., 2017), CTGAN (Xu et al., 2019), Table-GAN (Park et al., 2018), TVAE (Xu et al., 2019), and GOGGLE (Liu et al., 2022) on a collection of six tasks for which rich background knowledge is available: our datasets are annotated with up to 31 constraints, each including up to 17 different features. To this end, we first show that these models often violate the constraints, with WGAN even generating 100% of non-compliant samples for one dataset, and five models generating more than 95% non-compliant samples for another one (see Table 1). Then, we quantitatively demonstrate that the addition of the constraints in the topology of the network improves the performance both in terms of detection (i.e., how well the generated data matches the real data distribution) and in terms of utility (i.e., how well the generated data can replace real data to train ML models), with up to 6.5% improvement over all datasets. This validates the principle that generating compliant samples contributes to improving their quality: a challenging problem for which the state-of-the-art progresses only by small successive improvements (Liu et al., 2022; Kim et al., 2023). Further, we show how CL does not necessarily need to be added at training time, but it can simply be incorporated at inference time as a guardrail. This is important when the model is available only as a black-box system without any possibility to act on it. Finally, we show how CL does not hinder the sample generation time of the models. This paper thus follows the same principles outlined in Giunchiglia et al. (2023a), where the authors advocate for a more requirements-driven machine learning, where performance is just one of the requirements defining a model has to satisfy (others might include safety, fairness, robustness etc.).

**Contributions:** (i) We show that standard DGMs often generate synthetic data that are not aligned with the available background knowledge. (ii) We develop a method able to take as input any set of constraints expressed as linear inequalities and automatically create a differentiable constraint layer that can be seamlessly integrated with DGMs. (iii) We prove that the samples generated with C-DGMs are guaranteed to be compliant with the given constraints. This is the first method that is able to incorporate complex background knowledge into DGMs and guarantee that it is always satisfied. (iv) We show that C-DGMs outperform their standard counterparts in terms of both utility and detection. (v) We display how CL can be used as a guardrail at inference time. (vi) We quantitatively demonstrate that CL has negligible impact on the samples' generation time.

## 2 PROBLEM STATEMENT

Let $p_X$ be an unknown distribution over $X \in \mathbb{R}^D$, and let $\mathcal{D}$ be the training dataset consisting of $N$ i.i.d. samples drawn from $p_X$. The goal of standard generative modeling is to learn from $\mathcal{D}$ the parameters $\theta$ of a generative model such that the model distribution $p_\theta$ approximates $p_X$.

In *constrained generative modeling*, we assume to have access to a set of constraints expressing some background knowledge about the sample space of $p_X$, i.e., stating which samples are admissible and which are not. The goal is to learn the parameters $\theta$ of a generative model such that (i) the model distribution $p_\theta$ approximates $p_X$, and (ii) the sample space of $p_\theta$ is compliant with the constraints.

Let $\Pi$ be a finite set of constraints expressing such background knowledge, where each constraint is a linear inequality over a set of *variables* $\mathcal{X} = \{x_k \mid k = 1, \dots, D\}$, each variable uniquely corresponding to a feature of the dataset, and for this reason in the following we do not distinguish between variables and their corresponding features. Each constraint has the following form:

$$\sum_k w_k x_k + b \trianglerighteq 0, \tag{1}$$

with $w_k \in \mathbb{R}$, $b \in \mathbb{R}$, $\unrhd \in \{\geq, >\}$, and $x_k$ representing a continuous feature. Indeed, any set of linear inequalities using $\unrhd \in \{<, \leq, =, \geq, >\}$ can be easily converted into an equivalent one in which $\unrhd \in \{\geq, >\}$. In (1), when $\unrhd$ is $>$, we say that the linear inequality is *strict*.

**Example 2.1.** $\Pi = \{x_1 - x_2 \geq 0, x_2 - 5 > 0\}$ *expresses that for every generated sample $\tilde{x}$, the $1^{st}$ feature must be greater than or equal to the $2^{nd}$, and that its $2^{nd}$ feature must be greater than* 5.

Linear constraints represent an important class of constraints—as underlined by the existence of entire fields dedicated to its study (see e.g., linear programming)—and they possess many favourable properties, such as: (i) they are logically complete, as inconsistencies can be determined by computing linear combinations of the constraints themselves (Farkas, 1902), (ii) it is possible to compile them in a backtrack free representation that allows for computing satisfying samples in linear time in the size of the compiled representation (Dechter, 1999), and (iii) they always define a convex space.

A *sample* generated by a DGM is an assignment to the variables in $\mathcal{X}$. A sample $\tilde{x}$ *satisfies*:

1. the constraint (1) if $\sum_k w_k \tilde{x}_k + b \unrhd 0$, where $\tilde{x}_k$ is the value associated with the feature $x_k$,
2. a set $\Pi$ of constraints if it satisfies all the constraints in $\Pi$.

Contrarily, if $\tilde{x}$ does not satisfy a constraint (resp., $\Pi$) then $\tilde{x}$ *violates* the constraint (resp., $\Pi$). A set $\Pi$ of constraints is *satisfiable* if there exists a sample satisfying $\Pi$. A DGM model $m$ is *compliant* with $\Pi$ if all its generated samples satisfy $\Pi$.

Given a DGM $m$ generating samples that possibly do not satisfy $\Pi$, our goal is to create a C-DGM noted C-$m$ such that: (i) C-$m$ is compliant with $\Pi$, and (ii) for each sample $\tilde{x}$ generated by $m$, C-$m$ generates $\tilde{x}'$ such that $\tilde{x}'$ satisfies $\Pi$ and that, intuitively, is optimal in the sense that it minimally differs from $\tilde{x}$ (while taking into account the user preferences on which features should be changed).

In the following, given a constraint $\phi$ of form (1), a variable $x_k$ *appears positively* (resp., *negatively*) in $\phi$ if $w_k > 0$ (resp., $w_k < 0$). A variable *appears in* $\phi$ if it appears positively or negatively in $\phi$.

## 3 CONSTRAINED DEEP GENERATIVE MODELS

To achieve our goal, we build a differentiable *constraint layer* (CL) that (i) can be seamlessly integrated with multiple DGMs, (ii) guarantees the satisfaction of the constraints, and (iii) guarantees a possibly optimal output that minimally changes the initial DGM predictions. Given a generic DGM, CL can be added at training time right after the layer generating the samples. CL allows the gradients to backpropagate through it and thus the network can learn to exploit the background knowledge it encodes. To better ground the discussion, consider Figure 1, where we can see an overview of how to integrate our layer with a GAN-based model. In the Figure, the noise vector $z$ is fed into the generator, which in turn outputs a sample $\tilde{x}$. Often, $\tilde{x}$ needs to be transformed using a pre-defined mapping $f$ before using it, as the output space of the generator may not coincide with the feature space of the real data. Then, $f(\tilde{x})$ is passed to CL, which returns the corrected $\tilde{x}'$—now compliant with the constraints $\Pi$—and then $f^{-1}(\tilde{x}')$ is passed to the discriminator together with $f^{-1}(r)$, where $r \in \mathcal{D}$ is a real data point for our dataset. In the following, we focus on the construction of CL, and, for ease of presentation and without loss of generality, we assume that the output space of the generator coincides with the feature space of the real dataset. Thus, we assume that the generator generates a sample $\tilde{x}$, which then gets fed directly to CL, which in turn returns CL$(\tilde{x})$, the output of the C-DGM.

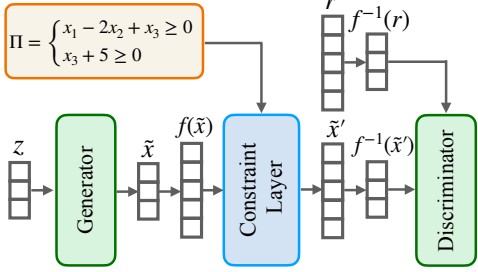

Figure 1: Overview on how to integrate CL into a GAN-based model.

Given a sample $\tilde{x}$, the basic idea of the constraint layer is to incrementally compute the value $\tilde{x}_i$ of the $i$th feature in the sample, minimally changing it whenever $\tilde{x}_i$ is not compatible with the values of the already analyzed features and the given constraints. To this end, let $\lambda : \mathcal{X} \mapsto [1, D]$ be a *variable ordering function* injectively mapping each variable in $\mathcal{X}$ to an integer in $[1, D]$. Such ordering is user-defined and defines the order of computation of the features. To ease the notation, and without

loss of generality, from now on we denote with $x_i$ the variable with order $i$ (and thus $\lambda(x_i) = i$). Then, given $\lambda$, we associate to each variable $x_i$ a set of constraints $\Pi_i$ inductively defined as follows:

1. if $i = D$, then $\Pi_i = \Pi$, and
2. if $i < D$ and $j = i + 1$, then

$$\Pi_i = \Pi_j \setminus (\Pi_j^- \cup \Pi_j^+) \cup \{red_j(\phi^1, \phi^2) \mid \phi^1 \in \Pi_j^-, \phi^2 \in \Pi_j^+\},$$

where (i) $\Pi_j^-$ (resp., $\Pi_j^+$) is the subset of $\Pi_j$ where $x_j$ appears negatively (resp., positively) and (ii) $red_j(\phi^1, \phi^2)$ is the *reduction of* $\phi^1$ *and* $\phi^2$ *over* $x_j$ defined, assuming $\phi^1 = \sum_k w_k^1 x_k + b^1 \unrhd^1 0$ and $\phi^2 = \sum_k w_k^2 x_k + b^2 \unrhd^2 0$ with $\unrhd^1, \unrhd^2 \in \{\geq, >\}$, as

$$\sum_{k \neq j} (w_k^1 |w_j^2| + w_k^2 |w_j^1|) x_k + b^1 |w_j^2| + b^2 |w_j^1| \unrhd 0,$$

in which $\unrhd$ is $\unrhd^1$ if $\unrhd^1 = \unrhd^2$, and $\unrhd$ is $>$ otherwise.

**Example 3.1.** *Continuing with the example,* $\Pi = \{x_1 - x_2 \geq 0, x_2 - 5 > 0\}$, $\Pi_2 = \Pi$, *and* $\Pi_1 = \{red_2(x_1 - x_2 \geq 0, x_2 - 5 > 0)\} = \{x_1 - 5 > 0\}$.

Consider now a generic model $m$ and a sample $\tilde{x}$ generated by $m$. To define $\mathrm{CL}(\tilde{x})$, we first compute the upper bounds (resp., lower bounds) associated with $\tilde{x}_i$ given the constraints and the computed values $\mathrm{CL}(\tilde{x})_j$ associated with the features $x_j$ with $j < i$. Indeed, each constraint $\phi$ of the form (1) in $\Pi_i^+$ (resp., $\Pi_i^-$) defines a lower (resp., upper) bound on $x_i$ given by the expression $\varepsilon_i^\phi = -\sum_{k \neq i} (w_k/w_i) x_k - b/w_i$, as $\phi$ is equivalent to $x_i - \varepsilon_i^\phi \unrhd 0$ (resp., $-x_i + \varepsilon_i^\phi \unrhd 0$).

Then, for each $i = 1, \ldots, D$, the values associated with the *least upper bound* ($ub_i$) and *greatest lower bound* ($lb_i$) for the feature $i$ given the values of $\mathrm{CL}(\tilde{x})_1, \ldots, \mathrm{CL}(\tilde{x})_{i-1}$ are defined to be:[1]

$$ub_i = \min \left( \bigcup_{\phi \in \Pi_i^-} \varepsilon_i^\phi (\mathrm{CL}(\tilde{x})) \right) \qquad lb_i = \max \left( \bigcup_{\phi \in \Pi_i^+} \varepsilon_i^\phi (\mathrm{CL}(\tilde{x})) \right), \qquad (2)$$

where $\varepsilon_i^\phi(\mathrm{CL}(\tilde{x}))$ corresponds to $\varepsilon_i^\phi$ instantiated with the values of $\mathrm{CL}(\tilde{x})$. Clearly, the fact that $\Pi$ is satisfiable and the definition of $ub_i$ and $lb_i$ in (2) which incorporates the values of $\mathrm{CL}(\tilde{x})_j$ for $j < i$, ensures $lb_i \leq ub_i$ for any $i = 1, \ldots, D$ (as formally stated below). However, assuming, e.g., that for a sample $\tilde{x}$, $\tilde{x}_i \leq lb_i$, it may not be possible to set the value of the feature $x_i$ to be exactly $lb_i$, because there exists a strict linear inequality $x_i > lb_i$ in $\Pi_i^+$. Indeed, in these cases, there does not exist a value $v$ for $x_i$ with $v > lb_i$ and minimum $|v - \tilde{x}_i|$, and the only way out is to select $v = lb_i + \epsilon$, where $\epsilon > 0$ is an arbitrarily chosen small value. For this reason, for every $i = 1, \ldots, D$, $\mathrm{CL}(\tilde{x})_i$ is defined as

$$\mathrm{CL}(\tilde{x})_i = \min^i(\max^i(\tilde{x}_i, lb_i), ub_i), \qquad (3)$$

where $\max^i(\tilde{x}_i, lb_i)$ is equal to $v = \max(\tilde{x}_i, lb_i)$ if there does not exist a strict inequality $\phi$ in $\Pi_i^+$ such that $v = \varepsilon_i^\phi(\mathrm{CL}(\tilde{x}))$, and is equal to $v + \epsilon$ with $\epsilon > 0$ small enough to ensure $\epsilon < ub_i - lb_i$, otherwise. The analogous intuition applies to $\min^i$.

**Example 3.2.** *Continuing with the example,* $\Pi = \{x_1 - x_2 \geq 0, x_2 - 5 > 0\}$, $\Pi_2 = \Pi$, $\Pi_1 = \{x_1 - 5 > 0\}$. *If* $\tilde{x}_1 = 7$ *and* $\tilde{x}_2 = 3$, *then* $CL(\tilde{x})_1 = 7$ *and* $CL(\tilde{x})_2 = 5.1$, *while if* $\tilde{x}_1 = \tilde{x}_2 = 3$, *then* $CL(\tilde{x})_1 = CL(\tilde{x})_2 = 5.1$, *such values computed with* $\epsilon = 0.1$.

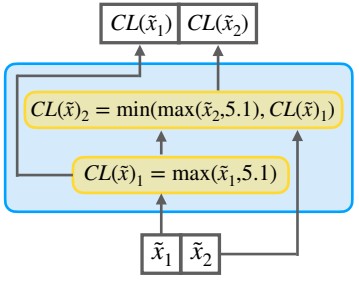

Figure 2: CL in Example 3.2.

Notice that for every sample it is always possible to compute, at least in theory, a value for $\epsilon$ such that $\mathrm{CL}(\tilde{x})$ satisfies the constraints in $\Pi$. In practice, in the implementation, we fix $\epsilon$ to be the minimum representable positive float.

**Theorem 3.3.** *Let $m$ be a deep generative model. Let $\Pi$ be a satisfiable and finite set of constraints over the sample space. Let $\lambda$ be a variable ordering and CL be the constraint layer built from $\lambda$ and $\Pi$. Then, the model C-$m$ obtained by incorporating CL in $m$ is compliant with $\Pi$.*

---

[1]We assume the function $\min(V)$ over a finite set $V$ of values in $\mathbb{R}$ to be defined as $\min(\emptyset) = +\infty$, and $\min(\{v\} \cup V') = v$ if $v \leq \min(V')$ and $\min(V')$ otherwise. Analogously for the function $\max(V)$.

See proof in Appendix A.1, which contains also the proof of the following corollary.

**Corollary 3.4.** *In the hypotheses of the theorem, for every sample $\tilde{x}$ produced by $m$ and for every $i = 1, \ldots, D$, it is always the case that $ub_i \geq lb_i$, and thus also*

$$CL(\tilde{x})_i = \min{}^i(\max{}^i(\tilde{x}_i, lb_i), ub_i) = \max{}^i(\min{}^i(\tilde{x}, ub_i), lb_i). \tag{4}$$

In addition to satisfying the constraints, our goal was to guarantee $CL(\tilde{x})$ optimality, capturing the intuition that it has to minimally differ from $\tilde{x}$, i.e., that it is not possible to reduce the distance $|CL(\tilde{x})_i - \tilde{x}_i|$ for each $i = 1, \ldots, D$, while satisfying the constraints. Such intuition is formalized by saying that $CL(\tilde{x})$ is *optimal (with respect to $\Pi$)* if

1. $CL(\tilde{x})$ satisfies $\Pi$, and
2. there does not exist another sample $\tilde{x}'$ different from $CL(\tilde{x})$ which satisfies $\Pi$ and such that for each $i = 1, \ldots, D$, $|\tilde{x}'_i - \tilde{x}_i| \leq |CL(\tilde{x})_i - \tilde{x}_i|$.

Indeed, if a sample $\tilde{x}$ satisfies the constraints, we expect $CL(\tilde{x})$ to return $\tilde{x}$ and thus be optimal. Notice that more than one value might satisfy the above definition of optimality. Indeed, given a sample $\tilde{x}$, the proposed definition has the advantageous properties that (i) if $\tilde{x}$ satisfies the constraints then $CL(\tilde{x}) = \tilde{x}$ is the unique optimal solution, and (ii) if $\tilde{x}$ does not satisfy the constraints then it is not possible to get "closer" to $\tilde{x}$ along all the dimensions while satisfying the constraints. This rather relaxed definition of optimality allows us to consider different methods to compute $CL(\tilde{x})$, each corresponding to a preference about the set of features which we want to minimally change. In our case, we want to minimally change those features whose distribution is better approximated by the unconstrained DGM (see Appendix C.3 for the definition of two different policies to compute $CL(\tilde{x})$). Indeed, though in a different setting, Giunchiglia et al. (2023b) showed that computing a correction without taking into account the confidence in the predictions consistently leads to worse performance.

**Theorem 3.5.** *Let $m$ be a deep generative model. Let $\Pi$ be a satisfiable and finite set of constraints over the sample space. Let $\lambda$ be a variable ordering and CL be the constraint layer built from $\lambda$ and $\Pi$. Let $\tilde{x}$ be a sample produced by $m$. If $\tilde{x}$ satisfies $\Pi$, then $CL(\tilde{x}) = \tilde{x}$ and $CL(\tilde{x})$ is optimal.*

The proof is in Appendix A.2. Excluding the lucky cases in which the sample $\tilde{x}$ already satisfies the constraints, we will show that $CL(\tilde{x})$ is ensured to be optimal, e.g., when $\Pi$ does not contain strict inequalities. In the general case, however, $CL(\tilde{x})$ may not be optimal since an optimal value for $CL(\tilde{x})$ may not exist. This is due to the presence in $\Pi$ of strict linear inequalities (see, e.g, in Example 3.2). However, for such cases we can prove that $CL(\tilde{x})$ can take values as close as needed to the original sample $\tilde{x}$ while satisfying the constraints. In order to make these notions precise, we introduce $CL^{\geq}(\tilde{x})$ as the value computed by CL for the constrained generative modeling problem in which $\Pi$ is replaced with $\Pi^{\geq}$. In $\Pi^{\geq}$, each strict linear inequality $\sum_k w_k x_k + b > 0$ in $\Pi$ is replaced with the corresponding non-strict one, i.e., $\sum_k w_k x_k + b \geq 0$. By definition, the samples satisfying $\Pi$ also satisfy $\Pi^{\geq}$, and $CL^{\geq}(\tilde{x})$ is optimal when considering $\Pi^{\geq}$ but may not be optimal when considering $\Pi$ (as $CL^{\geq}(\tilde{x})$ violates some strict linear inequality in $\Pi$). In such cases, $CL^{\geq}(\tilde{x})$ can be considered as the limit optimal solution when considering $\Pi$ and we prove that $CL(\tilde{x})$ can get arbitrarily close to $CL^{\geq}(\tilde{x})$ while satisfying the constraints in $\Pi$.

**Theorem 3.6.** *Let $m$ be a deep generative model. Let $\Pi$ be a satisfiable and finite set of constraints over the sample space. Let $\lambda$ be a variable ordering and CL be the constraint layer built from $\lambda$ and $\Pi$. Let $\tilde{x}$ be a sample produced by $m$.*

1. *$CL(\tilde{x})$ is optimal if $CL(\tilde{x}) = CL^{\geq}(\tilde{x})$, and*
2. *$CL(\tilde{x})$ tends to $CL^{\geq}(\tilde{x})$ as the $\epsilon$ values used to compute $CL(\tilde{x})$ tend to 0, otherwise.*

According to the theorem, $CL(\tilde{x})$ is guaranteed to be optimal when no $\epsilon$ values are introduced for computing $CL(\tilde{x})$, and if an $\epsilon$ value has to be introduced because of a strict linear inequality, $CL(\tilde{x})$ can still be as close as needed to the optimal solution $CL^{\geq}(\tilde{x})$ of the relaxed problem with $\Pi^{\geq}$.

## 4 EXPERIMENTAL ANALYSIS

We quantitatively evaluate different aspects of our approach to support the claims of our paper:

Table 1: Constraint violation rate for each model and dataset.

| Model/Dataset | URL | WiDS | LCLD | Heloc | FSP | News |
|---|---|---|---|---|---|---|
| WGAN | 11.1±1.6 | 98.2±0.2 | 100.0±0.0 | 57.0±13.0 | 70.7±8.3 | 45.6±9.6 |
| TableGAN | 4.9±1.4 | 96.4±2.4 | 6.1±0.9 | 45.6±16.3 | 71.6±8.7 | 72.6±5.3 |
| CTGAN | 3.1±2.6 | 99.9±0.0 | 11.8±2.7 | 41.6±12.1 | 74.3±5.2 | 54.3±10.1 |
| TVAE | 3.0±0.7 | 99.9±0.0 | 3.9±0.5 | 55.5±1.4 | 66.4±3.0 | 50.3±3.9 |
| GOGGLE | 5.9±6.6 | 78.2±11.6 | 13.1±2.9 | 47.3±7.0 | 63.7±17.6 | 44.8±7.2 |
| All C-models | **0.0±0.0** | **0.0±0.0** | **0.0±0.0** | **0.0±0.0** | **0.0 ±0.0** | **0.0±0.0** |

1. **Background knowledge alignment:** *How often do constraint violations occur?* Section 4.2 shows that standard DGMs often violate the constraints expressing background knowledge.
2. **Synthetic data quality:** *Does background knowledge improve the synthetic data quality?* Section 4.3 shows that C-DGMs outperform DGMs in terms of two metrics: utility and detection.
3. **Post-processing ability:** *Can CL act as a guardrail at inference time?* Section 4.4 shows that CL can be used as a guardrail while slightly improving the overall performance of the DGMs.
4. **Generation time:** *Does background knowledge injection affect generation time?* Section 4.5 shows that the sample generation time for C-DGMs is comparable with that of DGMs.

## 4.1 EXPERIMENTAL ANALYSIS SETTINGS

**Models:** We experimented with five DGM models, namely WGAN (Arjovsky et al., 2017), Table-GAN (Park et al., 2018), CTGAN (Xu et al., 2019), TVAE (Xu et al., 2019), and GOGGLE (Liu et al., 2022). Each model was augmented by our CL, resulting in C-WGAN, C-TableGAN, C-CTGAN, C-TVAE, and C-GOGGLE models. Refer to Appendix B.1 for details on the models.

**Datasets:** To evaluate the models, we selected real-world datasets for which a clear description of the feature relationships either was available or has been derived from our domain expertise. We limited our selection to datasets having at least 3 known constraints. As a result, we found 6 such datasets, 4 for binary classification tasks, 1 for multi-class classification, and 1 for regression. The datasets vary not only in size (2K to 1M rows), but also in feature number (24 to 109) and feature type (continuous, categorical, or mixed). The constraints between features are equally varied in number (from 4 to 31) as well as number of features appearing in each constraint (from 1 to 17). Additional details about the datasets and constraints statistics can be found in Appendix B.2 and B.3, respectively.

**Quality Evaluation:** We quantitatively evaluate two characteristics of the quality of generated samples: (i) *utility*, i.e., whether our synthetic data can be used as an alternative fake dataset to train other models, and (ii) *detection*, i.e., whether a classifier can be trained to tell apart the synthetic samples from the real data. In order to evaluate the utility of the generated samples, we follow the *Train on Synthetic, Test on Real* (TSTR) framework (Esteban et al., 2017; Jordon et al., 2019) which is a paradigm used in different papers (see, e.g., Kim et al. (2023)). Following this framework, we train multiple models (e.g., Random Forest, XGBoost etc.), validate them with original training data, and test them on real test data. To evaluate such models, we report the average F1-score, AUROC, and weighted F1-score for the classification datasets, and explained variance and mean absolute error for regression. On the other hand, to evaluate in terms of detection, we follow the popular evaluation method for tabular data generation (see, e.g., Liu et al. (2022)) and train six models (e.g., Decision Tree, AdaBoost etc.) to distinguish between the real and synthetic datasets. Again, we report average F1-score, AUROC and weighted F1-score. All the metrics are reported over 5 runs. For more details on the evaluation procedure, refer to Appendix B.4. Our full code is provided on GitHub.[2]

## 4.2 BACKGROUND KNOWLEDGE ALIGNMENT

*How often do constraint violations occur?* To assess the ability of standard DGMs to create samples that are aligned with the available background knowledge, we measure the *constraints' violation rate* (CVR) which, given a set of synthetic samples $\mathcal{S}$ and a set of constraints $\Pi$, represents the

---

[2]The code is available at https://github.com/mihaela-stoian/ConstrainedDGM.

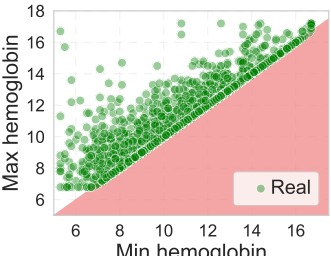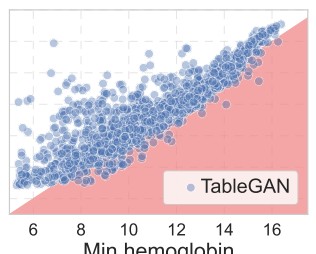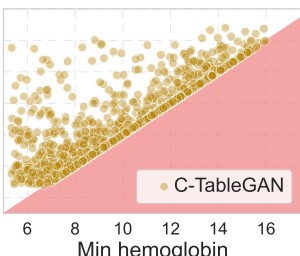

Figure 3: Real data and samples generated by TableGAN and C-TableGAN for WiDS.

percentage of samples in $\mathcal{S}$ violating $\Pi$. Table 1 shows the CVR for each tested model and dataset. In particular, the first five rows report the CVR for the standard DGMs, while the last row reports the CVR for all their respective constrained versions (i.e., C-WGAN, C-TableGAN, etc.) as it is always equal to zero for all models, datasets and runs. This is expected, as our C-DGMs offer theoretical guarantees to always generate samples compliant with the constraints. Let us now focus on the results obtained for the standard DGMs. Firstly, no standard DGM guarantees to produce only samples satisfying the constraints. Further, in more than half of the reported experiments, we have CVR > 50% (i.e., more than half of the generated samples violate the constraints), with peaks of CVR = 100% for WGAN tested on LCLD and CVR > 95% for four out of five DGMs tested on WiDS. Additionally, to give a better overview of the phenomenon, in Appendix C.1 we present the results according to two other metrics: (i) *constraints violation coverage* (CVC), which, given $\mathcal{S}$ and $\Pi$, represents the percentage of constraints in $\Pi$ that have been violated at least once by any of the samples in $\mathcal{S}$, and (ii) *samplewise constraints violation coverage* (sCVC), which represents the average over the samples in $\mathcal{S}$ of the percentage of the constraints violated by each sample. Finally, to visually demonstrate the difference between DGMs and the C-DGMs, we select three constraints where only two variables appear, and we create two-dimensional scatter-plots of the (i) real data, (ii) the samples generated by the DGMs and (iii) the ones generated by the C-DGMs, with the variables appearing in the constraint on the axes. In Figure 3 we present only the plots for TableGAN, the other figures can be found in Appendix C.1. In Figure 3, we consider the constraint *MaxHemoglobinLevel ≥ MinHemoglobinLevel* from the WiDS dataset, and we highlight in red the region violating the constraint. The Figure confirms our quantitative analysis, as TableGAN often violates the constraint, while C-TableGAN never does. Further, we can see how the distribution of the samples generated by C-TableGAN more closely matches the one of the real data.

## 4.3 SYNTHETIC DATA QUALITY

*Does background knowledge improve the synthetic data quality?* To validate our hypothesis, we compare the performance of the standard DGMs with their respective C-DGMs in terms of utility and detection, as specified in Section 4.1. The results of our experiments are summarised in Table 2, where we report the average utility performance (left) and the average detection performance (right) over all datasets with respect to: F1-score (F1), weighted F1-score (*w*F1), and Area Under the ROC Curve (AUC). As we can see from Table 2, except for two cases (F1 utility of C-CTGAN and AUROC detection of C-TVAE), our C-DGMs outperform their standard counterparts

Table 2: Results for every C-DGM and their respective standard version. The best results are in bold.

|  | **Utility (↑)** | | | **Detection (↓)** | | |
|---|---|---|---|---|---|---|
|  | F1 | *w*F1 | AUC | F1 | *w*F1 | AUC |
| WGAN | 0.463 | 0.488 | 0.730 | 0.945 | 0.943 | 0.954 |
| C-WGAN | **0.483** | **0.502** | **0.745** | **0.915** | **0.912** | **0.934** |
| TableGAN | 0.330 | 0.400 | 0.704 | 0.908 | 0.907 | 0.926 |
| C-TableGAN | **0.375** | **0.432** | **0.714** | **0.898** | **0.895** | **0.917** |
| CTGAN | **0.517** | 0.532 | 0.771 | 0.902 | 0.901 | 0.920 |
| C-CTGAN | 0.516 | **0.537** | **0.773** | **0.894** | **0.891** | **0.919** |
| TVAE | 0.497 | 0.527 | 0.767 | 0.869 | 0.868 | **0.892** |
| C-TVAE | **0.507** | **0.537** | **0.773** | **0.868** | **0.867** | 0.898 |
| GOGGLE | 0.344 | 0.373 | 0.624 | 0.926 | 0.926 | 0.943 |
| C-GOGGLE | **0.409** | **0.427** | **0.667** | **0.925** | **0.916** | **0.937** |

both in terms of utility and detection according to all metrics almost always, and never have worse

performance. Often, such difference in performance is non-negligible, as in the case of GOGGLE where we register a 6.5% difference in terms of utility when calculated in terms of F1-score.

Notice that for utility results we had to leave out of the average News regression dataset, as there we use metrics with different scales (i.e., Explained Variance and Mean Absolute Error). The utility results obtained on News are reported in the Appendix C.4.1 in Table 20. For News, in all C-DGMs we see an improvement or no change in the utility performance according to at least one of the two metrics. We report the utility and detection for individual datasets in Appendix C.4.1 and C.4.2. The hyperparameters used in each experiment are reported in Appendix B.5, Table 7.

## 4.4 POST-PROCESSING ABILITY

*Can CL act as a guardrail at inference time?* In many practical scenarios, users might not want to modify and retrain their model to add a constraint layer. For example, this might be the case for companies that already have their (possibly black-box) models and/or already generated data that simply needs to be aligned with the available background knowledge. In this Subsection, we thus show that CL can be added at inference time as a guardrail on any model without hindering the quality of the generated data. The results of our analysis are shown in Table 3, where P-DGM stands for a standard DGM with CL added as a post-processing step at inference time. The results demonstrate

Table 3: Results for every P-DGM and their respective standard version. The best results are in bold.

|  | Utility (↑) | | | Detection (↓) | | |
|---|---|---|---|---|---|---|
|  | F1 | *w*F1 | AUC | F1 | *w*F1 | AUC |
| WGAN | **0.463** | 0.488 | 0.730 | 0.945 | 0.943 | 0.954 |
| P-WGAN | 0.462 | **0.489** | **0.732** | **0.930** | **0.929** | **0.946** |
| TableGAN | **0.330** | **0.400** | 0.704 | 0.908 | 0.907 | 0.926 |
| P-TableGAN | 0.328 | 0.399 | **0.707** | **0.901** | **0.898** | **0.922** |
| CTGAN | **0.517** | **0.532** | **0.771** | 0.902 | 0.901 | **0.920** |
| P-CTGAN | 0.512 | 0.528 | 0.770 | **0.897** | **0.900** | 0.926 |
| TVAE | **0.497** | **0.527** | 0.767 | **0.869** | **0.868** | **0.892** |
| P-TVAE | 0.495 | 0.524 | **0.767** | 0.875 | 0.876 | 0.902 |
| GOGGLE | 0.344 | 0.373 | 0.624 | **0.926** | 0.926 | **0.943** |
| P-GOGGLE | **0.348** | **0.374** | **0.626** | 0.925 | **0.924** | 0.942 |

that P-DGMs outperform their standard counterparts 17 times out of 30 (in 1 other case having equal performance), thus showing the potential of using CL as a post-processor. As expected, the difference in results is often very little, as the CL layer is only applied at inference time. On the contrary, if we compare the P-DGMs models with the C-DGMs from Table 2, we can see that the C-DGMs almost always perform better (there are only two exceptions where the difference in performance is as little as 0.001). This is again expected, as C-DGMs are injected with the background knowledge at training time, thus making them able to exploit it.

## 4.5 IMPACT ON SAMPLES GENERATION TIME

*Does background knowledge injection affect generation time?* Another important aspect of synthetic data generators is their sample generation time (see, e.g., Kim et al. (2023); Xiao et al. (2022)). In order to show that the addition of CL has basically no impact on speed, for each dataset and model we measure the generation time of 1000 samples and report the average results over 5 runs in Table 4. The results show that the constrained versions are as fast or at most 0.03s slower than the unconstrained version for 15 and 14 cases (out of 30 cases), respectively, with only one case 0.27s

Table 4: Sample generation time in seconds.

|  | URL | WiDS | LCLD | Heloc | FSP | News |
|---|---|---|---|---|---|---|
| WGAN | 0.02 | 0.03 | 0.01 | 0.00 | 0.00 | 0.01 |
| C-WGAN | 0.02 | 0.04 | 0.01 | 0.01 | 0.01 | 0.02 |
| TableGAN | 0.18 | 3.21 | 0.17 | 0.17 | 0.18 | 0.20 |
| C-TableGAN | 0.19 | 3.19 | 0.18 | 0.18 | 0.18 | 0.19 |
| CTGAN | 0.13 | 0.26 | 0.08 | 0.06 | 0.08 | 0.14 |
| C-CTGAN | 0.14 | 0.27 | 0.08 | 0.06 | 0.08 | 0.14 |
| TVAE | 0.12 | 0.27 | 0.06 | 0.06 | 0.06 | 0.12 |
| C-TVAE | 0.13 | 0.27 | 0.07 | 0.06 | 0.07 | 0.13 |
| GOGGLE | 0.71 | 3.99 | 9.91 | 0.16 | 0.06 | 2.01 |
| C-GOGGLE | 0.71 | 3.86 | 10.18 | 0.16 | 0.06 | 2.04 |

slower than the unconstrained version. This means that the constrained layer introduces almost no overhead to the sampling process in many practical scenarios. This result is particularly interesting

given that our representation of the constraints may have an exponential size (Dechter, 1999). Indeed, we did not experience any exponential blow-up, which happens in the worst case, when the constraints have many variables in common.

## 5 RELATED WORK

Our work lies at the intersection of standard tabular data synthesis and neuro-symbolic AI for its ability to incorporate background knowledge into neural architectures.

**Tabular Data Synthesis.** Several approaches based on DGMs have been specifically designed to address particular challenges in generating tabular data such as mixed types of features, and imbalanced categorical data. Notable among these are GAN-based approaches like TableGAN (Park et al., 2018), CTGAN (Xu et al., 2019), OCT-GAN (Kim et al., 2021), and IT-GAN (Lee et al., 2021). These methods leverage the power of GANs to model the underlying data distribution and generate synthetic samples that closely resemble real-world tabular data. Few approaches focus especially on particular domains like healthcare where privacy is important (see, e.g., (Choi et al., 2017; Che et al., 2017)). Following privacy concerns, two approaches, i.e., DPGAN (Xie et al., 2018) and PATE-GAN (Jordon et al., 2019), incorporate differential privacy techniques to ensure that the generated synthetic data does not reveal sensitive information about the individuals in the original dataset. Alternative to GANs, Xu et al. (2019) proposed TVAE as a variation of the standard Variational AutoEncoder, while TabDDPM (Kotelnikov et al., 2023) and STaSy (Kim et al., 2023) were proposed following the achievements of score-based models. Finally, Liu et al. (2022) proposed GOGGLE, a model that uses graph learning to infer relational structure from the data.

**Neuro-symbolic AI Methods.** Neuro-symbolic AI raised great interest in the recent past for many reasons (see, e.g., (Raedt et al., 2018; d'Avila Garcez & Lamb, 2023), among which there is the ability to incorporate complex background knowledge in deep learning models. Standard approaches in the field incorporate logical constraints in the training of a neural network by introducing additional terms in the loss function that penalize the network for violating them (see, e.g., (Diligenti et al., 2012; 2017; Xu et al., 2018; Fischer et al., 2019; Badreddine et al., 2022; Stoian et al., 2023)). These approaches, while often easy to incorporate into neural models, do not give any guarantee that the constraints are actually satisfied. Alternative approaches, e.g., DeepProbLog (Manhaeve et al., 2018), rely on a solver to inject background knowledge as a set of definite clauses both at training and inference time. A more recent work (van Krieken et al., 2023) in this line proposed using neural networks for performing approximate inference in polynomial time to address the scalability problem of probabilistic neuro-symbolic learning frameworks. More closely to our approach, we find the recent methods that are able to incorporate the background knowledge into the topology of the network itself. However, these methods are either only able to deal with constraints expressed in propositional logic (Ahmed et al., 2022) or even less expressive constraints (Giunchiglia & Lukasiewicz, 2021) or become quickly intractable for even moderately complex logical constraints (Hoernle et al., 2022). Regarding the application of neuro-symbolic AI in generative tasks: Liello et al. (2020) incorporates propositional logic constraints on GANs for structured objects generation, while Misino et al. (2022) shows how to integrate ProbLog (Raedt et al., 2007) with VAEs. Our work differs from the last two both in the application domain and in the type of background knowledge we incorporate.

## 6 DISCUSSION AND CONCLUSIONS

Our work introduces a novel method able to translate complex constraints, expressed as linear inequalities, into a seamlessly integrated layer within Deep Generative Models (DGMs), thereby yielding Constrained Deep Generative Models (C-DGMs). This infusion of domain-specific knowledge directly into the network's architecture facilitates the generation of more realistic tabular data samples, as C-DGMs are guaranteed to comply with the specified constraints. Furthermore, our experiments reveal that including our layer during the training phase enhances the quality of the generated samples, thus underlining the importance of constraint integration not only during data generation but also throughout the model's learning process. **Limitations:** In this paper, we focus on constraints that can be expressed as linear inequalities. While this covers a wide range of scenarios, some relationships among features may demand more expressive constraint representations. In the future, we expect many developments in this work where even more complex constraints are considered.

AUTHOR CONTRIBUTIONS

**Mihaela Cătălina Stoian**: conceptualization, methodology, writing, constraint layer and software implementation, experimental analysis, model training, debugging; **Salijona Dyrmishi**: conceptualization, experimental design and analysis, writing, software implementation, model training, debugging; **Maxime Cordy**: supervision, editing, writing reviewing; **Thomas Lukasiewicz**: supervision, editing, writing reviewing; **Eleonora Giunchiglia**: conceptualization, methodology, formalization, writing, supervision.

ACKNOWLEDGMENTS

Mihaela Cătălina Stoian is supported by the EPSRC under the grant EP/T517811/1. Salijona Dyrmishi is supported by the Luxembourg National Research Funds (FNR) AFR Grant 14585105. This work was also supported by the Alan Turing Institute under the EPSRC grant EP/N510129/1, by the AXA Research Fund, by the EPSRC grant EP/R013667/1, and by the EU TAILOR grant. We would like to thank Andrew Ryzhikov, Salah Ghamizi, and Thibault Simonetto for the useful discussions. We also thank Thibault Simonetto for his contribution to the scaler and exploration of domain constraints. We also acknowledge the use of the EPSRC-funded Tier 2 facility JADE (EP/P020275/1), GPU computing support by Scan Computers International Ltd. and the HPC facilities of the University of Luxembourg.

ETHICS STATEMENT

While our method enables the generation of synthetic data, we recognize the importance of responsible and ethical use. It is conceivable that individuals could misuse our approach to create high-quality fake data for deceptive purposes, such as unauthorized access or selling counterfeit information. However, synthetic data generation can also provide many societal benefits, such as privacy protection as shown in (Park et al., 2018; Yoon et al., 2020).

REPRODUCIBILITY STATEMENT

To ensure the reproducibility of this paper, we include all the necessary details in the Appendix. Appendix A includes detailed proofs for the technical statements presented in the paper. Appendix B provides all the details on the experimental analysis settings, in particular: (i) Section B.2 provides the descriptions and links to the datasets, (ii) Section B.4 details the evaluation protocol we followed, and (iii) Section B.5 reports how the hyperparameters search was conducted, with Table 7 containing the best hyperparameter configuration for each dataset and model. The code is publicly available at https://github.com/mihaela-stoian/ConstrainedDGM.

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

# A  THEOREMS

## A.1  PROOF OF THEOREM 3.3

*Proof.* The proof is by induction on the number $n$ of variables in $\Pi$. We recall that $\{x_1, \ldots, x_D\}$ is the set of variables and that we assumed, w.l.o.g., $\lambda(x_i) = i$.

For the base case $n = 0$, $\Pi$ does not contain variables and, since $\Pi$ is satisfiable, for each constraint $\sum_k w_k x_k + b \trianglerighteq 0$ in $\Pi$, (i) each $w_k = 0$, (ii) $b \trianglerighteq 0$, and (iii) each sample satisfies $\Pi$.

Assume that $n > 1$ variables appear in $\Pi$. Let $x_k$ be the variable with the highest $\lambda(x_k)$ value occurring in $\Pi$.

Then, $x_D, x_{D-1}, \ldots, x_{k+1}$ do not occur in $\Pi$, $\Pi_D = \Pi_{D-1} = \ldots = \Pi_k = \Pi$, and

$$\Pi_{k-1} = \Pi_k \setminus (\Pi_k^- \cup \Pi_k^+) \cup \{red_k(\phi^1, \phi^2) \mid \phi^1 \in \Pi_k^-, \phi^2 \in \Pi_k^+\}. \tag{5}$$

Consider an arbitrary sample $\tilde{x}$.

In $\Pi_{k-1}$, by construction, less than $n$ variables appear. Thus, for the inductive hypothesis, assume $\mathrm{CL}(\tilde{x})$ satisfies $\Pi_{k-1}$. Proving that $\mathrm{CL}(\tilde{x})$ satisfies $\Pi_k$ is equivalent to proving that $\mathrm{CL}(\tilde{x})$ satisfies both $\Pi_k^+$ and $\Pi_k^-$, since we know from eq. (5) that $\Pi_k \subseteq \Pi_{k-1} \cup \Pi_k^- \cup \Pi_k^+$.

The proof is in two steps. We first prove by contradiction that either (i) $lb_k < ub_k$ or (ii) $lb_k = ub_k$ and there exist two constraints $\phi_1 \in \Pi_k^-$ and $\phi_2 \in \Pi_k^+$ such that: $ub_k = \varepsilon_k^{\phi^1}(\mathrm{CL}(\tilde{x}))$ and $lb_k = \varepsilon_k^{\phi^2}(\mathrm{CL}(\tilde{x}))$, and both $\phi^1$ and $\phi^2$ are not strict inequalities. Then, in the second step, we prove that $\mathrm{CL}(\tilde{x})$ satisfies each constraint $\phi \in \Pi_k^+ \cup \Pi_k^-$.

**First step.** Assume that either (i) $lb_k > ub_k$ or (ii) $lb_k = ub_k$ and at least one between $\phi^1$ and $\phi^2$ is a strict inequality. If $lb_k > ub_k$ or $lb_k = ub_k$, then $lb_k \neq -\infty$ and $ub_k \neq +\infty$ and thus both $\Pi_k^-$ and $\Pi_k^+$ are not empty. Let $\phi^1$ (resp. $\phi^2$) be the constraint in $\Pi_k^-$ (resp. $\Pi_k^+$) such that $ub_k = \varepsilon_k^{\phi^1}(\mathrm{CL}(\tilde{x}))$ (resp. $lb_k = \varepsilon_k^{\phi^2}(\mathrm{CL}(\tilde{x}))$). Such constraints $\phi_1$ and $\phi_2$ exist since $\Pi$ is finite. Then, by definition, $red_k(\phi^1, \phi^2)$ is equivalent to $\varepsilon_k^{\phi^1} - \varepsilon_k^{\phi^2} \geq 0$ if both $\phi^1$ and $\phi^2$ are non-strict inequalities, and to $\varepsilon_k^{\phi^1} - \varepsilon_k^{\phi^2} > 0$ if at least one between $\phi^1$ and $\phi^2$ is a strict inequality.

We know that $red_k(\phi^1, \phi^2) \in \Pi_{k-1}$ and that, by the inductive hypothesis, $\mathrm{CL}(\tilde{x})$ satisfies $\Pi_{k-1}$. Thus, $\mathrm{CL}(\tilde{x})$ satisfies $red_k(\phi^1, \phi^2)$, which (taken together with the definition of $red_k(\phi^1, \phi^2)$ above) implies that $\varepsilon_k^{\phi^1}(\mathrm{CL}(\tilde{x})) - \varepsilon_k^{\phi^2}(\mathrm{CL}(\tilde{x})) \geq 0$ if both $\phi^1$ and $\phi^2$ are non-strict inequalities, and that $\varepsilon_k^{\phi^1}(\mathrm{CL}(\tilde{x})) - \varepsilon_k^{\phi^2}(\mathrm{CL}(\tilde{x})) > 0$ if at least one between $\phi^1$ and $\phi^2$ is a strict inequality. However, by our assumption, $ub_k = \varepsilon_k^{\phi^1}(\mathrm{CL}(\tilde{x}))$ and $lb_k = \varepsilon_k^{\phi^2}(\mathrm{CL}(\tilde{x}))$. Thus, we have that $ub_k \geq lb_k$ if both $\phi^1$ and $\phi^2$ are non-strict inequalities, and that $ub_k > lb_k$ if at least one between $\phi^1$ and $\phi^2$ is a strict inequality, deriving a contradiction.

**Second step.** We now prove that $\mathrm{CL}(\tilde{x})$ satisfies each constraint $\phi \in \Pi_k^+$. The statement holds since

1. if $\phi$ is a non-strict inequality, by definition, we have that $\mathrm{CL}(\tilde{x})_k \geq lb_k \geq \varepsilon_k^\phi(\mathrm{CL}(\tilde{x}))$, and

2. if $\phi$ is a strict inequality, we have two cases. In the first one, $lb_k = \varepsilon_k^\phi(\mathrm{CL}(\tilde{x}))$ and, since $lb_k < ub_k$, it is possible to choose an $\epsilon > 0$ such that $\mathrm{CL}(\tilde{x})_k = lb_k + \epsilon < ub_k$, and thus $\mathrm{CL}(\tilde{x})_k > lb_k = \varepsilon_k^\phi(\mathrm{CL}(\tilde{x}))$. In the second case, $lb_k > \varepsilon_k^\phi(\mathrm{CL}(\tilde{x}))$ and then $\mathrm{CL}(\tilde{x})_k \geq lb_k > \varepsilon_k^\phi(\mathrm{CL}(\tilde{x}))$.

Analogously, $\mathrm{CL}(\tilde{x})$ satisfies each constraint $\phi \in \Pi_k^-$.

$\square$

## A.2 PROOF OF THEOREM 3.5

Let $\tilde{x}$ be a sample satisfying the constraint in $\Pi$. We recall that, w.l.o.g., we assume $\lambda(x_i) = i$. We will prove by contradiction that if $\tilde{x}$ satisfies $\Pi$, then $\text{CL}(\tilde{x}) = \tilde{x}$.

Assume $\text{CL}(\tilde{x}) \neq \tilde{x}$. Let $i$ be the lowest index such that $\text{CL}(\tilde{x})_i \neq \tilde{x}_i$. Then, $\text{CL}(\tilde{x})_i = \min^i(\max^i(\tilde{x}_i, lb_i), ub_i) \neq \tilde{x}_i$, and thus either $\tilde{x}_i < lb_i$ or $\tilde{x}_i > ub_i$, both cases being impossible given

1. the definitions of $lb_i$ and $ub_i$,

2. the hypothesis that $\tilde{x}$ satisfies the constraints in $\Pi$, and

3. the fact that all the constraints in $\Pi_i^- \cup \Pi_i^+$ are entailed by $\Pi$ and thus satisfied by $\tilde{x}$.

## A.3 PROOF OF THEOREM 3.6

We prove the two statements of the theorem in separate lemmas, after a first introductory lemma.

**Lemma A.1.** *In the hypotheses of Theorem 3.6, $CL^{\geq}(\tilde{x})$ is optimal with respect to $\Pi^{\geq}$.*

*Proof.* Let $\tilde{x}$ be a sample. We recall that, w.l.o.g., we assume $\lambda(x_i) = i$.

We show that for every $i = 1, \ldots, D$, there does not exist another sample $\tilde{x}'$ satisfying $\Pi^{\geq}$ such that for each $1 \leq j < i$, $\tilde{x}'_j = \text{CL}^{\geq}(\tilde{x})_j$ and $|\tilde{x}'_i - \tilde{x}_i| < |\text{CL}^{\geq}(\tilde{x})_i - \tilde{x}_i|$. The proof is by induction on $i$.

For the base case ($i = 1$), we know that in $\Pi_1^{\geq}$ the only variable that can appear is $x_1$, thus we have four cases:

1. $\Pi_1^{\geq}$ is either empty or contains a constraint $c \geq 0$ which is always satisfied since $\Pi$ (and thus $\Pi^{\geq}$) is satisfiable. In this case, $\text{CL}^{\geq}(\tilde{x})_1 = \tilde{x}_1$ and the statement trivially holds;

2. $\Pi_1^{\geq}$ is equivalent to a single constraint $\{x_1 + a \geq 0\}$ in which case if $\tilde{x}_1 \geq -a$ then $\text{CL}^{\geq}(\tilde{x})_1 = \tilde{x}_1$ otherwise $\text{CL}^{\geq}(\tilde{x})_1 = -a$ and the statement trivially holds;

3. $\Pi_1^{\geq}$ is equivalent to a single constraint $\{-x_1 + b \geq 0\}$ in which case if $\tilde{x}_1 \leq b$ then $\text{CL}^{\geq}(\tilde{x})_1 = \tilde{x}_1$ otherwise $\text{CL}^{\geq}(\tilde{x})_1 = b$ and the statement trivially holds;

4. $\Pi_1^{\geq}$ is equivalent to a pair of constraints $\{x_1 + a \geq 0, -x_1 + b \geq 0\}$. Since $\Pi$ is satisfiable, $a + b \geq 0$. Then, if $\tilde{x}_1 < -a$ then $\text{CL}^{\geq}(\tilde{x})_1 = -a$, if $-a \leq \tilde{x}_1 \leq b$ then $\text{CL}^{\geq}(\tilde{x})_1 = \tilde{x}_1$, and if $\tilde{x}_1 > b$ then $\text{CL}^{\geq}(\tilde{x})_1 = b$. For each of the three cases, the statement trivially holds.

Assume $i = j + 1 > 1$. Assume by contradiction that there exists another sample $\tilde{x}'$ satisfying $\Pi^{\geq}$ such that for each $1 \leq j < i$, $\tilde{x}'_j = \text{CL}^{\geq}(\tilde{x})_j$ and $|\tilde{x}'_i - \tilde{x}_i| < |\text{CL}^{\geq}(\tilde{x})_i - \tilde{x}_i|$. By construction of $\Pi_i^{\geq}$, we know that only $x_1, \ldots, x_i$ appear in $\Pi_i^{\geq}$. If we substitute each variable $x_j$ with $j < i$ with $\text{CL}^{\geq}(\tilde{x})_j$ in $\Pi_i^{\geq}$, then (i) in the resulting set of constraints the only variable is $x_i$, (ii) we are back to the base case, and (iii) the statement follows.

$\square$

**Lemma A.2.** *In the hypotheses of Theorem 3.6, $CL(\tilde{x})$ is optimal if $CL(\tilde{x}) = CL^{\geq}(\tilde{x})$.*

*Proof.* The proof is a direct consequence of the facts that (i) $\text{CL}(\tilde{x})$ satisfies the constraints in $\Pi$, (ii) $\text{CL}(\tilde{x}) = \text{CL}^{\geq}(\tilde{x})$, (iii) $\text{CL}^{\geq}(\tilde{x})$ is optimal wrt $\Pi^{\geq}$ (from Lemma A.1), and (iv) the samples satisfying $\Pi$ are a subset of the samples satisfying $\Pi^{\geq}$. $\square$

**Lemma A.3.** *In the hypotheses of Theorem 3.6, $CL(\tilde{x})$ tends to $CL^{\geq}(\tilde{x})$ as the $\epsilon$ values used to compute $CL(\tilde{x})$ tend to 0.*

*Proof.* Let $\epsilon_1, \ldots, \epsilon_k$ ($k \geq 0$) be the $\epsilon$ values (assumed, w.l.o.g., to be distinct) in $\mathrm{CL}(\tilde{x})$. If $k = 0$ then $\mathrm{CL}(\tilde{x}) = \mathrm{CL}^{\geq}(\tilde{x})$, and the statement trivially holds.

Consider $\mathrm{CL}(\tilde{x})_i$, $i = 1, \ldots, D$, and substitute $\epsilon_j$ with a newly introduced variable $v_j$. $\mathrm{CL}(\tilde{x})_i$ is a composition of continuous functions in the newly introduced variables and, thus, it is continuous. For an arbitrary continuous function $f : \mathbb{R}^k \to \mathbb{R}$ we know that $\lim_{x \to x_0} f(x) = f(x_0)$. Thus,

$$\lim_{[v_1, \ldots, v_k] \to [0, \ldots, 0]} \mathrm{CL}(\tilde{x})_i = \mathrm{CL}^{\geq}(\tilde{x})_i,$$

and hence

$$\lim_{[v_1, \ldots, v_k] \to [0, \ldots, 0]} \mathrm{CL}(\tilde{x}) = \mathrm{CL}^{\geq}(\tilde{x}).$$

$\square$

# B  EXPERIMENTAL ANALYSIS SETTINGS

## B.1  MODELS

In our experimental analysis, we use five base models:

- **WGAN** (Arjovsky et al., 2017) is a GAN model trained with Wasserstein loss in a typical generator discriminator GAN-based architecture. In our implementation, WGAN uses a MinMax transformer for the continuous features and one-hot encoding for categorical ones. It has not been designed specifically for tabular data.

- **TableGAN** (Park et al., 2018) is among the first GAN-based approaches proposed for tabular data generation. In addition to the typical generator and discriminator architecture for GANs, the authors proposed adding a classifier trained to learn the relationship between the labels and the other features. The classifier ensures a higher number of semantically correct produced records. TableGAN uses a MinMax transformer for the features.

- **CTGAN** (Xu et al., 2019) uses a conditional generator and training-by-sampling strategy in a generator-discriminator GAN architecture to model tabular data. The conditional generator generates synthetic rows conditioned on one of the discrete columns. The training-by-sampling ensures that the data are sampled according to the log-frequency of each category. Both help to better model the imbalanced categorical columns. CTGAN transforms discrete features using one-hot encoding and a mode-based normalization for continuous features. A variational Gaussian mixture model (Camino et al., 2018) is used to estimate the number of modes and fit a Gaussian mixture. For each continuous value, a mode is sampled based on probability densities, and its mean and standard deviation are used to normalize the value.

- **TVAE** (Xu et al., 2019) was proposed as a variation of the standard Variational AutoEncoder to handle tabular data. It uses the same transformations of data as CTGAN and trains the encoder-decoder architecture using evidence lower-bound (ELBO) loss.

- **GOGGLE** (Liu et al., 2022) is a graph-based approach to learning the relational structure of the data as well as functional relationships (dependencies between features). The relational structure of the data is learned by building a graph where nodes are variables and edges indicate dependencies between them. The functional dependencies are learned through a message passing neural network (MPNN). The generative model generates each variable considering its surrounding neighborhood.

## B.2  DATASETS

We use 6 real-world datasets covering both classification and regression tasks. An overview of these datasets' statistics can be found in Table 5. For the selection, we focused on datasets with at least three feature relationship constraints that either were provided with the description of the datasets or we could derive with our domain expertise. The selected datasets are listed below:

Table 5: Datasets statistics.

| Dataset | # Train | # Val | # Test | # Features | # Cat. | # Cont. | Task (# classes) |
|---|---|---|---|---|---|---|---|
| URL | 7K | 2K | 2K | 64 | 20 | 44 | Binary classification |
| WiDS | 22K | 6K | 7K | 109 | 9 | 100 | Binary classification |
| LCLD | 494K | 199K | 431K | 29 | 8 | 21 | Binary classification |
| Heloc | 8K | 2K | 0.2K | 24 | 8 | 16 | Binary classification |
| FSP | 2K | 0.2K | 0.2K | 28 | 0 | 28 | Multi-class class. (7) |
| News | 31K | 7K | 1K | 60 | 14 | 46 | Regression |

- URL[3] (Hannousse & Yahiouche, 2021) is used to perform webpage phishing detection with features describing statistical properties of the URL itself as well as the content of the page.
- WiDS[4] is used to predict if a patient is diagnosed with a particular type of diabetes named Diabetes Mellitus, using data from the first 24 hours of intensive care.
- LCLD[5] is used to predict whether the debt lent is unlikely to be collected. In particular, we use the feature-engineered dataset from Simonetto et al. (2022), inspired from the LendingClub loan data. The dataset captures features related to the loan as well as client history.
- FICO's Home Equity Line of Credit dataset (Heloc[6]) from the FICO xML Challenge is used to predict whether customers will repay their credit lines within 2 years. Similarly to LCLD, the dataset has features related to the credit line and the client's history.
- FSP[7] (Buscema, 1998) is used to predict 7 types of surface defects in stainless steel plates. The features approximately capture the geometric shape of the defect and its outline.
- News[8] (Fernandes et al., 2015) is used to predict the number of times a news article will be shared on social networks. The features capture properties of the text, as well as the publishing time.

For URL, WiDS, and LCLD, we used the train-val-test splits provided by Simonetto et al. (2022). For Heloc we used the train-test split of 80-20 and 20% of the training set was later split for validation. Finally, for FSP and News we split the data into 80-10-10% sets, and for the former (which is a multiclass classification dataset) we preserved the class imbalance.

### B.3 Constraints Datasheet

Here we give an overview of the structure of our constraints. To this end, we define $F$ to be the number of features appearing in at least one constraint, and we remind that $D$ is the total number of features. Then, given a constraint $\phi$, we define $F_\phi^+$ (resp. $F_\phi^-$) to be the number of features appearing positively (resp. negatively) in $\phi$, and $F_\phi = F_\phi^+ + F_\phi^-$ to be the number of features appearing in $\phi$.

As we can see from Table 6, the constraints characteristics greatly vary from one dataset to the other. Indeed, we can have from 4 to 31 constraints annotated for each dataset, with as little as 15% of the variables appearing in at least a constraint in News to as much as 56.88% in WiDS. Further, we can see that for almost all datasets, the average number of features appearing per constraint is 2.00, except for URL, which has a constraint where as many as 17 different features appear, and LCLD where we have 2 constraints where a single variable appears.

### B.4 Evaluation protocol

For evaluating the utility of the DGM/C-DGM models presented in our paper, we followed closely the protocol from Kim et al. (2023) which we also reproduce here.

---

[3] Link to dataset: https://data.mendeley.com/datasets/c2gw7fy2j4/2

[4] Link to dataset: https://www.kaggle.com/competitions/widsdatathon2021

[5] Link to dataset: https://figshare.com/s/84ae808ce6999fafd192

[6] Link to the dataset: https://huggingface.co/datasets/mstz/heloc

[7] Link to dataset: https://www.kaggle.com/datasets/uciml/faulty-steel-plates

[8] Link to dataset: https://archive.ics.uci.edu/dataset/332/online+news+popularity

Table 6: Constraints statistics.

| Dataset | # Constr. | $F$ / $D$ | Avg. $F_\phi$ | Avg. $F_\phi^+$ | Avg. $F_\phi^-$ |
|---|---|---|---|---|---|
| URL | 8 | 24 / 64 | 4.25 | 1.00 | 3.25 |
| WiDS | 31 | 62 / 109 | 2.00 | 1.00 | 1.00 |
| LCLD | 4 | 5 / 29 | 1.50 | 0.75 | 0.75 |
| Heloc | 7 | 10 / 24 | 2.00 | 1.00 | 1.00 |
| FSP | 4 | 7 / 28 | 2.00 | 1.00 | 1.00 |
| News | 5 | 9 / 60 | 2.00 | 1.00 | 1.00 |

1. First, we generate a synthetic dataset, split into training, validation and test partitions using the same proportions as the real dataset.

2. Then, we perform a hyperparameter search using the synthetic training data partition to train different classifiers/regressors.

   For the binary classification datasets (i.e., URL, WiDS, LCLD, and Heloc) we use: Decision Tree (Wu et al., 2008), AdaBoost (Schapire, 2013), Multi-layer Perceptron (MLP) (Haykin, 1994), Random Forest (Ho, 1995), XGBoost (Chen & Guestrin, 2016), and Logistic Regression (Cox, 1958) classifiers. For multi-class classification datasets, (i.e., FSP) we use Decision Tree, MLP, Random Forest, and XGBoost classifiers. For the regression dataset, (i.e., News) we use MLP, XGBoost, and Random Forest regressors and linear regression. For all the classifiers and regressors above, we considered the same hyperparameter settings as those from Table 26 of (Kim et al., 2023) and picked the best hyperparameter configuration using the real validation set according to the F1-score.

3. Finally, we tested the selected best models on the real test set and averaged the results across all the classifiers/regressors to get performance measurements for the DGM/C-DGM predictions according to three different metrics: F1-score, weighted F1-score, and Area Under the ROC Curve.

The above procedure was repeated 5 times for each DGM/C-DGM model, and the results were averaged separately for each of the metrics.

For evaluating the DGM/C-DGM models in terms of detection, we slightly adapted the procedure presented by Kim et al. (2023). We first created the training, validation, and test sets by concatenating real and synthetic data, including their targets as usual features, and adding a new target column that specifies whether the data is real or not. By construction, these datasets are binary classification datasets and, thus, are suitable for the hyperparameter search procedure presented in Kim et al. (2023) using the 6 different binary classifiers mentioned above. We proceeded to pick the best model using the newly-created validation set, and then we obtained the final detection performance on the newly-created test data (which combines the real and the synthetic data).

### B.5 HYPERPARAMETER SEARCH

Prior to experimenting with our C-DGM methods, we conducted an extensive hyperparameter search to reveal the best configurations. We always chose the best settings according to the utility performance measured either by the average over the F1-score, weighted F1-score, and Area Under the ROC Curve for the binary and multi-class classification datasets or by the Mean Absolute Error for the regression dataset.

**Initial phase.** For GOGGLE, we used the same optimiser and learning rate set as Liu et al. (2022). Specifically, we used Adam (Kingma & Ba, 2015) with a set of 5 different learning rates: $\{1 \times 10^{-3}, 5 \times 10^{-3}, 1 \times 10^{-2}\}$. For TVAE, we used Adam with a set of 5 different learning rates: $\{5 \times 10^{-6}, 1 \times 10^{-5}, 1 \times 10^{-4}, 2 \times 10^{-4}, 1 \times 10^{-3}\}$. And for each of the other DGM models, we used three different optimizers, Adam, RMSProp (Hinton, 2014), SGD (Ruder, 2016), each with a different set of learning rates:

- for WGAN $\{1 \times 10^{-4}, 1 \times 10^{-3}\}$, $\{5 \times 10^{-5}, 1 \times 10^{-4}, 1 \times 10^{-3}\}$, and $\{1 \times 10^{-4}, 1 \times 10^{-3}\}$, respectively.

- for TableGAN, $\{5 \times 10^{-5}, 1 \times 10^{-4}, 2 \times 10^{-4}, 1 \times 10^{-3}\}$, $\{1 \times 10^{-4}, 2 \times 10^{-4}, 1 \times 10^{-3}\}$ and $\{1 \times 10^{-4}, 1 \times 10^{-3}\}$, respectively.

- for CTGAN, $\{5 \times 10^{-5}, 1 \times 10^{-4}, 2 \times 10^{-4}\}$, $\{1 \times 10^{-4}, 2 \times 10^{-4}, 1 \times 10^{-3}\}$ and $\{1 \times 10^{-4}, 1 \times 10^{-3}\}$, respectively.

Then, for each of the above optimizer-learning rate pairs, we tested three different batch sizes, depending on the DGM model: $\{64, 128, 256\}$ for WGAN, $\{128, 256, 512\}$ for TableGAN, $\{70, 280, 500\}$ for CTGAN and TVAE, and $\{64, 128\}$ for GOGGLE. The batch sizes for CTGAN are multiples of 10, to allow for using the CTGAN's recommended PAC (Lin et al., 2018) value of 10, among other values.

**Further search.** The initial phase of hyperparameter search allowed us to narrow down the space of configurations and focus on further investigating the most promising one. For the second phase, we varied the following parameters for each DGM, keeping fixed the rest from the best model of the initial phase:

- for WGAN we varied initially PAC values to 4, 8, and 16 and then discriminator iterations to 1,2, and 10.

- for TableGAN we varied separately *i)* generator layers dimensions to 128 from 100 initially and *ii)* embedding dimensions by doubling the value.

- for CTGAN we varied the value of PAC within $\{1, 5, 15\}$.

- for TVAE we varied the multiplier of the loss within $\{1, 2, 3, 4\}$.

The best hyperparameters settings are presented in Table 7. We used these configurations for the experiments presented in our paper. The same hyperparameters were then used for C-DGMs. Furthermore, for C-DGMs we reported the variable ordering hyperparameter that needs to be given to our CL. We give a full description of the orderings and of the impact they have on the performance in Appendix C.3.

## C  RESULTS

### C.1  BACKGROUND KNOWLEDGE ALIGNMENT

Besides constraints violation rate (CVR) presented in the main text, we measure two additional metrics: (i) *constraints violation coverage* (CVC), which, given a set of samples $\mathcal{S}$ and $\Pi$, represents the percentage of constraints in $\Pi$ that have been violated at least once by any of the samples in $\mathcal{S}$, and (ii) samplewise *constraints violation coverage* (sCVC), which represents the average over the samples in $\mathcal{S}$ of the percentage of the constraints violated by each sample.

Table 8 shows that for 5 out of 6 datasets, all the samples generated by unconstrained DGMs violate at least 50% of the constraints, with CVC reaching up to 100% for WiDS, FSPand News. This entails that the models actually struggle with the majority of the constraints specified, and that the problem cannot be solved by just fixing a very small subset of the available constraints. Meanwhile, all our C-DGMs guarantee that not a single constraint is violated by any of the samples.

Regarding sCVC, the results in Table 9 show that for all models and datasets, on average each sample can violate up to 29.8% of the constraints. As a reminder, even a single constraint violated indicates an unfeasible example in a real setting. The unconstrained DGM models learn different representations that produce different rankings for sCVC. For example, WGAN violates on average the most constraints per samples in LCLD. But, for TableGAN, CTGAN, TVAE and GOGGLE, LCLD is among the datasets with the lowest sCVC. Again, all our C-DGMs ensure that sCVC is 0.0, meaning not a single constraint is violated for all the samples.

We complement our quantitative analysis of background knowledge alignment for the generated synthetic samples with some visualizations. In particular, we consider three different constraints where only two variables appear, and then for each one of them, we create four two-dimensional scatter plots with the variables appearing in the constraint on the axes. The first scatter plot represents the real data, while the other three represent the samples generated by each of the DGMs and the C-DGMs. We now consider them one by one.

Table 7: Best hyperparameter settings used for DGMs/C-DGMs in our experiments.

| Model/Dataset | Hyperparameter | URL | WiDS | LCLD | Heloc | FSP | News |
|---|---|---|---|---|---|---|---|
| WGAN | Batch size | 64 | 128 | 128 | 64 | 128 | 128 |
| | Optimiser | Adam | RMSProp | SGD | Adam | RMSProp | RMSProp |
| | Learning rate | 0.0001 | 0.001 | 0.001 | 0.0001 | 0.001 | 0.001 |
| | Epochs | 300 | 80 | 30 | 200 | 20 | 50 |
| | Discriminator iters | 10 | 10 | 5 | 10 | 10 | 10 |
| | Ordering | Rnd | KDE | KDE | KDE | KDE | KDE |
| TableGAN | Batch size | 128 | 128 | 128 | 128 | 128 | 128 |
| | Optimiser | Adam | RMSProp | Adam | Adam | Adam | RMSProp |
| | Learning rate | 0.001 | 0.001 | 0.001 | 0.001 | 0.001 | 0.001 |
| | Epochs | 300 | 50 | 25 | 200 | 200 | 50 |
| | Ordering | Corr | Corr | KDE | Corr | KDE | KDE |
| CTGAN | Batch size | 500 | 500 | 500 | 500 | 500 | 500 |
| | Optimiser | Adam | RMSProp | Adam | Adam | Adam | Adam |
| | Learning rate | 0.0002 | 0.0005 | 0.0002 | 0.0002 | 0.0002 | 0.0002 |
| | Epochs | 150 | 50 | 20 | 500 | 300 | 100 |
| | Ordering | KDE | Corr | Rnd | Corr | Corr | Corr |
| TVAE | Batch size | 70 | 70 | 500 | 500 | 70 | 70 |
| | Optimiser | Adam | Adam | Adam | Adam | Adam | Adam |
| | Learning rate | 0.0002 | 0.000005 | 0.00001 | 0.000005 | 0.00001 | 0.00001 |
| | Epochs | 150 | 50 | 40 | 150 | 500 | 50 |
| | Loss factor | 2 | 2 | 4 | 2 | 1 | 3 |
| | Ordering | KDE | KDE | Corr | Corr | Corr | Corr |
| GOGGLE | Batch size | 128 | 128 | 128 | 64 | 128 | 64 |
| | Optimiser | Adam | Adam | Adam | Adam | Adam | Adam |
| | Learning rate | 0.005 | 0.01 | 0.001 | 0.001 | 0.005 | 0.001 |
| | Epochs | 1000 | 200 | 200 | 1000 | 1000 | 250 |
| | Patience | 50 | 50 | 50 | 50 | 50 | 50 |
| | Ordering | Random | Corr | Corr | Random | Corr | KDE |

Table 8: Constraints violation coverage (CVC) for all datasets and models under study.

| Model/Dataset | URL | WIDS | LCLD | HELOC | Faults | News |
|---|---|---|---|---|---|---|
| WGAN | $11.1 \pm 1.6$ | $100.0 \pm 0.0$ | $75.0 \pm 0.0$ | $100.0 \pm 0$ | $75.0 \pm 0.0$ | $84.8 \pm 8.7$ |
| TableGAN | $24.5 \pm 3.7$ | $100.0 \pm 0.0$ | $50.0 \pm 0.0$ | $100.0 \pm 0$ | $75.0 \pm 0.0$ | $100.0 \pm 0.0$ |
| CTGAN | $16.5 \pm 3.8$ | $100.0 \pm 0.0$ | $50.0 \pm 0.0$ | $99.4 \pm 1.3$ | $75.0 \pm 0.0$ | $100.0 \pm 0.0$ |
| TVAE | $12.5 \pm 0.0$ | $100 \pm 0.0$ | $50.0 \pm 0.0$ | $100.0 \pm 0.0$ | $75.0 \pm 0.0$ | $100.0 \pm 0.0$ |
| GOGGLE | $17.5 \pm 6.9$ | $99.2 \pm 1.7$ | $50.0 \pm 0.0$ | $100.0 \pm 0.0$ | $75.0 \pm 0.0$ | $100.0 \pm 0.0$ |
| All C-models | $\mathbf{0.0 \pm 0.0}$ | $\mathbf{0.0 \pm 0.0}$ | $\mathbf{0.0 \pm 0.0}$ | $\mathbf{0.0 \pm 0.0}$ | $\mathbf{0.0 \pm 0.0}$ | $\mathbf{0.0 \pm 0.0}$ |

1. In Figure 4, we consider again the constraint *MaxHemoglobinLevel − MinHemoglobinLevel* $\geq 0$ appearing in WiDS dataset from Section 4.2 in the main text. We visualize the outputs for the remaining models WGAN, CTGAN, TVAE and GOGGLE and their constrained counterparts C-WGAN, C-CTGAN, C-TVAE and C-GOGGLE.

2. In Figure 5, we illustrate the behavior of the real and synthetic samples with respect to a constraint from the Heloc dataset which requires that the number of trades that have been insolvent for 60 days must be greater than the number of trades that have been insolvent for 90 days, i.e., *NumInsolventTradesGreaterThan60Days* $\geq$ *NumInsolventTradesGreaterThan90Days*.

3. In Figure 6, we illustrate the behavior of the real and synthetic samples with respect to a constraint from the FSP dataset that requires *X_Maximum* $\geq$ *X_minimum*, where *X_minimum* and *X_maximum* refer to the value for the $X$ coordinate in images captured of steel plates.

Table 9: Samplewise constraints violation coverage (sCVC) for all models and datasets under study.

| Model/Dataset | URL | WIDS | LCLD | HELOC | Faults | News |
|---|---|---|---|---|---|---|
| WGAN | $1.4 \pm 0.2$ | $21.2 \pm 1.3$ | $29.8 \pm 0.2$ | $10.5 \pm 2.5$ | $23.3 \pm 4.3$ | $12.1 \pm 2.4$ |
| TableGAN | $0.6 \pm 0.2$ | $14.4 \pm 2.8$ | $1.5 \pm 0.2$ | $9.0 \pm 3.3$ | $22.9 \pm 3.2$ | $24.1 \pm 3.3$ |
| CTGAN | $0.4 \pm 0.3$ | $21.6 \pm 0.5$ | $3.0 \pm 0.7$ | $7.6 \pm 0.3$ | $24.4 \pm 2.4$ | $15.8 \pm 3.9$ |
| TVAE | $0.4 \pm 0.1$ | $21.0 \pm 0.2$ | $1.0 \pm 0.1$ | $9.4 \pm 0.2$ | $21.5 \pm 1.5$ | $14.3 \pm 1.1$ |
| GOGGLE | $0.8 \pm 0.9$ | $10.6 \pm 2.4$ | $3.3 \pm 0.7$ | $17.2 \pm 4.9$ | $18.8 \pm 5.3$ | $10.7 \pm 1.6$ |
| All C-models | $\mathbf{0.0 \pm 0.0}$ | $\mathbf{0.0 \pm 0.0}$ | $\mathbf{0.0 \pm 0.0}$ | $\mathbf{0.0 \pm 0.0}$ | $\mathbf{0.0 \pm 0.0}$ | $\mathbf{0.0 \pm 0.0}$ |

The Figures visually demonstrate that the models that incorporate our constrained layer (C-WGAN, C-CTGAN, C-TableGAN, C-TVAE and C-GOGGLE) are guaranteed to produce outputs only within the feasible regions of the real data.

Finally, to further investigate the properties of the generated data by C-DGMs, we study the impact of constraint reparation on the boundary population for the three cases considered above. We defined a band around the boundary whose width $w$ we set to be proportional to the range of the values of the considered features in the real dataset. As in all the considered constraints only two features appear, we set $w = \sqrt{(r_1 p)^2 + (r_2 p)^2}$, where $r_1$ (resp. $r_2$) represents the range of the first (resp. second) feature, while $p$ represents the proportion of the range of values each feature can assume. If $p = 1$, then the width is equal to the diagonal of the rectangle defined by the two points with minimum and maximum coordinates.

The results in Table 10 show the percentage of generated samples that lay on the boundary when $p = \{1\%, 5\%, 10\%\}$ for the real dataset (last row), individual DGMs and C-DGMs, as well as their averages (third and second to last row, respectively). Notice that C-DGMs populate the boundary at a much closer rate to the real data than their unconstrained counterparts. Indeed the maximum difference between the percentage of real data points laying at the boundary and the average number of samples generated by C-DGMs is equal to 14.7% (for dataset WiDS and $p = 1\%$). On the contrary, the maximum difference between the number of real data points laying at the boundary and the average number of samples generated by DGMs is equal to 51.5% (for dataset FSP and $p = 1\%$)

## C.2 Feature Distribution Analysis

Following Kotelnikov et al. (2023) and Zhao et al. (2021) we perform a comparative analysis of the distributions of data generated by the models under study and the real data. The results are presented in Table 11 for continuous features, for which we measure the difference between the real data and generated data distributions using the Wasserstein distance, and in Table 12 for categorical features, for which we measure the difference between the real data and generated data distributions using the Jensen-Shannon divergence.

As we can see from Table 11, for every model and dataset there is very little difference in the Wasserstein distance obtained with the standard DGMs, the P-DGMs, and the C-DGMs. The only big gap is registered for GOGGLE on the WiDS dataset, where the Wasserstein distance obtained with C-GOGGLE is equal to 0.05 while the one obtained with GOGGLE and P-GOGGLE is equal to 0.22. Regarding the results for the categorical features in Table 12 we can see that the differences between DGMs and C-DGMs are very small, while, as expected, there is no difference between the results obtained with the DGMs and the P-DGMs. Indeed, our datasets are annotated with no constraints over the categorical features.

## C.3 Variable Ordering Study

In this subsection, we first discuss how we picked the variable orderings tested in our experimental analysis, and then we conduct an in-detail experimental analysis of how the different orderings perform with each model.

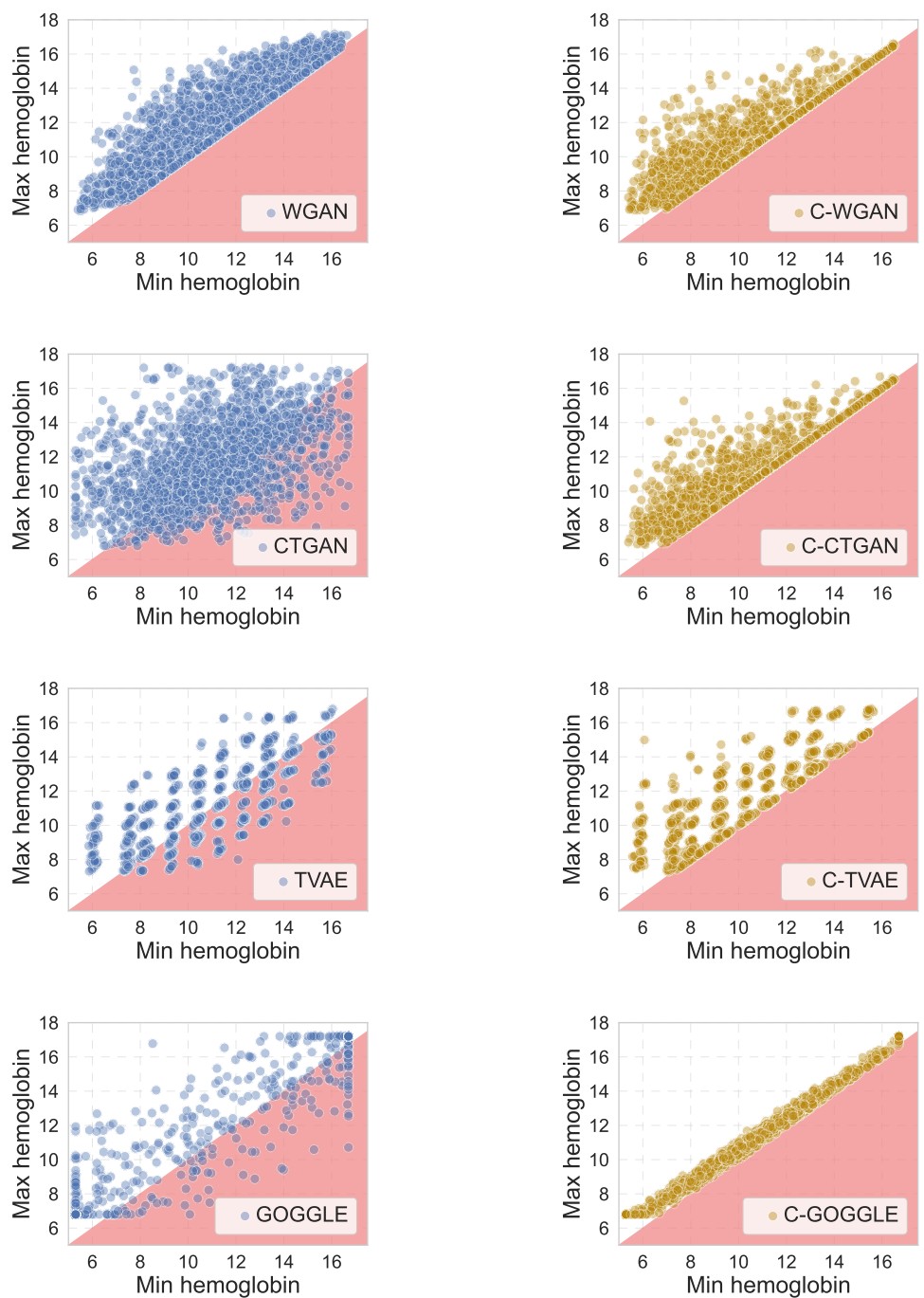

Figure 4: Samples generated by WGAN, CTGAN and TVAE, GOGGLE and their constrained versions for WiDS. The real and TableGAN distributions are given in Section 4.2 in the main text.

It is easy to see that given a sample $\tilde{x}$, the value of $\mathrm{CL}(\tilde{x})$ may depend on the selected variable ordering, i.e., that different variable orderings may lead to different $\mathrm{CL}(\tilde{x})$. Ideally, we would like the orderings to help in generating the highest quality samples possible. Thus, our intuition has been that features whose distributions are easier to learn should be assigned a lower ranking, so that their value can be fruitfully used to compute the value associated with features with higher ranking. With such intuition, we designed two different heuristics to choose such orderings. To

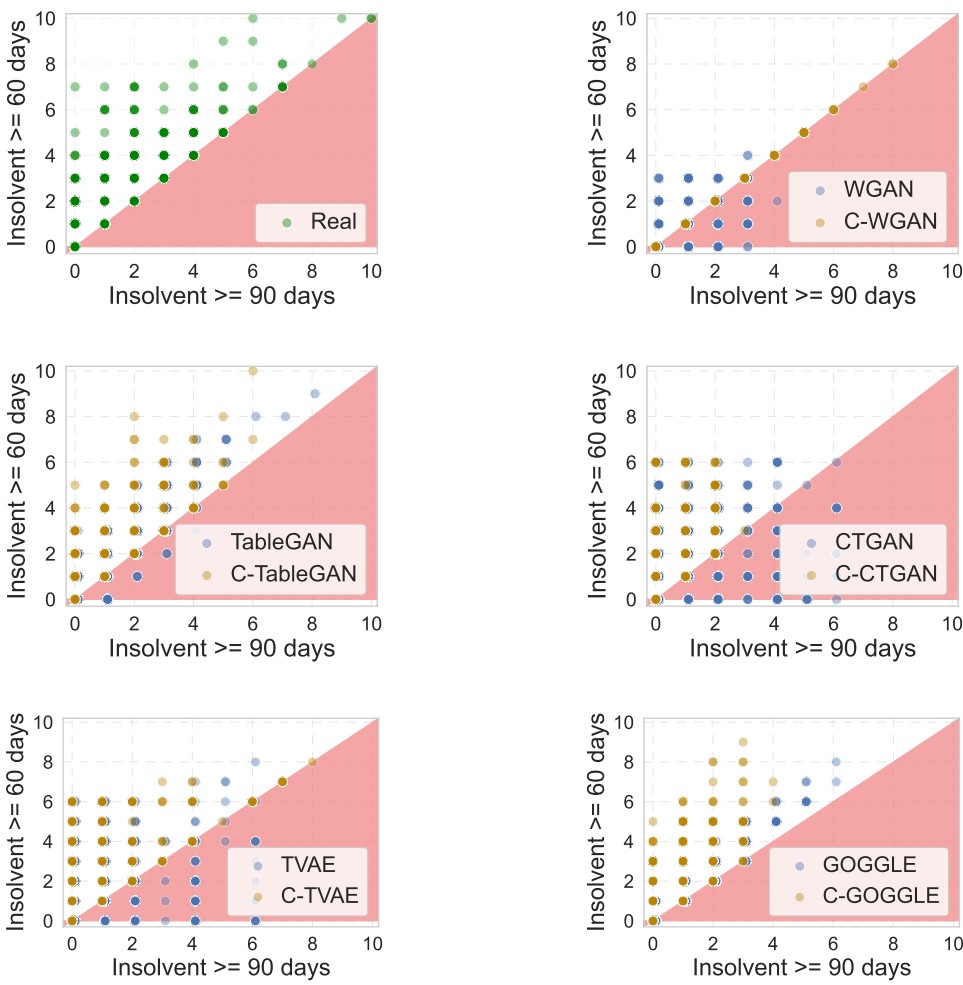

Figure 5: Real data and samples generated by WGAN, TableGAN, CTGAN, TVAE, GOGGLE and their constrained versions for Heloc.

obtain the two orderings described below, for each C-DGM, we first need to train the respective standard unconstrained DGM, and then generate a set of samples $\mathcal{S}$.

The first ordering we propose is a *correlation-based (Corr.) ordering*. Given the initial real data dataset $\mathcal{D}$ and $\mathcal{S}$, for each variable $x_i$ we compute the score $\sigma_i = |\sum_{j \neq i} \text{corr}(C_i^{\mathcal{D}}, C_j^{\mathcal{D}}) - \sum_{j \neq i} \text{corr}(C_i^{\mathcal{S}}, C_j^{\mathcal{S}})|$, where $C_j^{\mathcal{D}}$ (resp. $C_j^{\mathcal{S}}$) is the vector of the values of the $j$th column in $\mathcal{D}$ (resp. $\mathcal{S}$). The lower the value of $\sigma_i$ is, the lower the position $x_i$ gets in the variable ordering, and the sooner its value is computed (as its relations with the other features should be easier to compute for the DGM).

The second ordering we propose is a *kernel density estimation (KDE)-based ordering*, which can be obtained by following the protocol outlined below:

1. Using scikit-learn's [9] (Pedregosa et al., 2011) implementation of KDE, we estimated a probability density function of the multidimensional real data.

2. We then computed the log-likelihood of each sample belonging to $\mathcal{D}$ and $\mathcal{S}$ under the estimated model, using again scikit-learn. It is worth noting that scikit-learn returns probability values normalised over the sample spaces under evaluation. Using these, we treated

---

[9]https://scikit-learn.org/stable/index.html

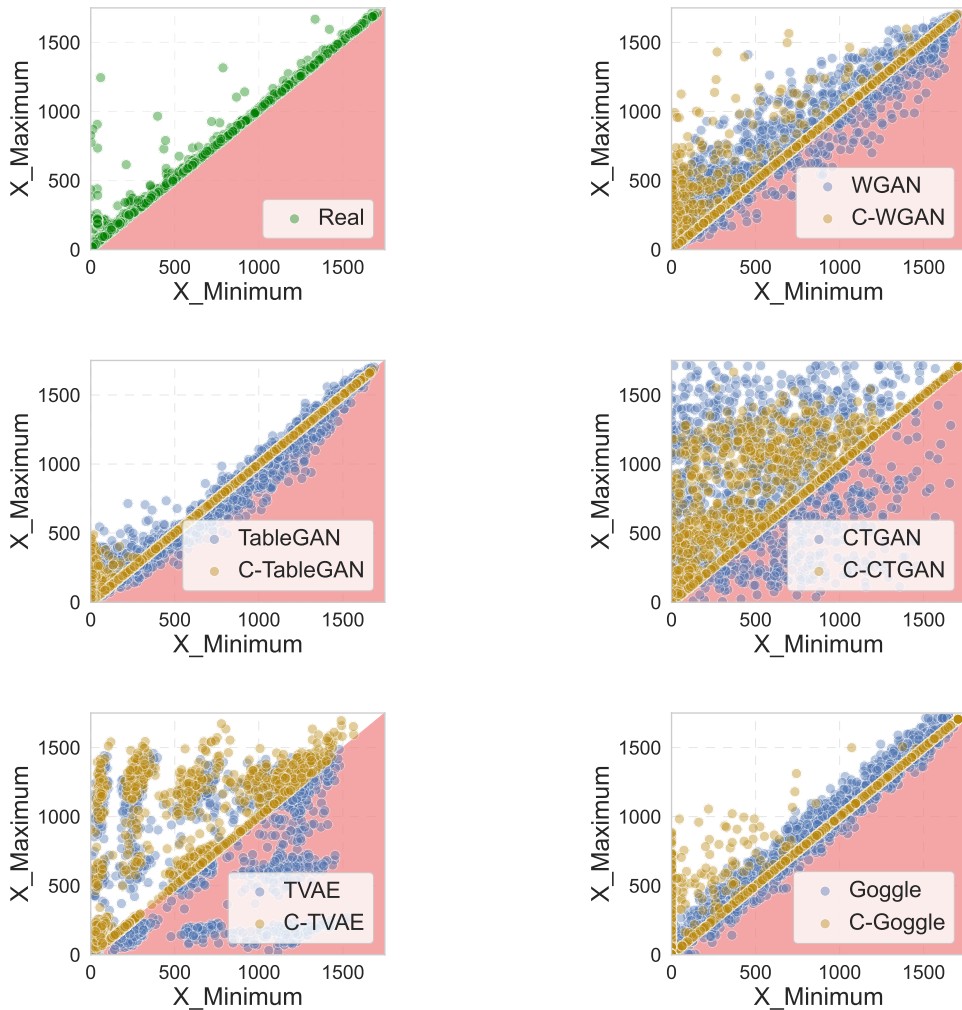

Figure 6: Real data and samples generated by WGAN, TableGAN, CTGAN, TVAE, GOGGLE and their constrained versions for FSP.

the problem in a discrete setting and approximated the marginal probability mass function of each variable. More specifically, for each feature $x_i$, we determined its unique values from $\mathcal{D} \cup \mathcal{S}$ and for each unique value $u_{ij}$ we summed the KDE scores of the samples for which $x_i = u_{ij}$, separately for $\mathcal{D}$ and $\mathcal{S}$. In case no sample was observed with $x_i = u_{ij}$, we set the value of the marginal probability mass function of $x_i$ to 0.

3. Hence, we obtain two marginal probability mass functions for each feature $x_i$ (one for the real data $D$ and one for the generated samples $\mathcal{S}$) and compute the Kullback-Leibler divergence (KL) between them to get a single value $d_i$.

4. Finally, we rank the features $x_i$ based on their KL scores $d_i$ in ascending order.

One disadvantage of this method is that the marginals are computed in a discrete setting. As future work, one interesting direction would be to use a higher-fidelity approximation of the marginal distributions, based on which the KL divergence scores, and hence the variable ranks in the ordering, are computed.

To assess the impact of using different orderings for each C-DGM model, we compared the two orderings proposed above with a random ordering (Rnd). Table 13 summarises the average utility

Table 10: Percentage of the generated samples that lie on the boundary.

| | WiDS | | | Heloc | | | FSP | | |
|---|---|---|---|---|---|---|---|---|---|
| | $p = 1\%$ | $p = 5\%$ | $p = 10\%$ | $p = 1\%$ | $p = 5\%$ | $p = 10\%$ | $p = 1\%$ | $p = 5\%$ | $p = 10\%$ |
| WGAN | 28.5±0.4 | 45.2±0.5 | 67.0±0.5 | 22.5±0.4 | 83.9±0.2 | 99.4±0.1 | 14.8±1.3 | 55.6±1.7 | 81.2±0.7 |
| C-WGAN | 62.1±0.3 | 72.1±0.2 | 83.1±0.2 | 100.0±0.0 | 100.0±0.0 | 100.0±0.0 | 76.8±1.0 | 83.1±0.8 | 89.9±0.5 |
| TableGAN | 19.7±0.1 | 71.9±0.2 | 88.7±0.1 | 37.4±0.2 | 94.7±0.1 | 99.8±0.0 | 25.3±1.2 | 76.0±0.9 | 95.8±0.5 |
| C-TableGAN | 72.2±0.2 | 81.1±0.3 | 88.7±0.3 | 83.8±0.3 | 96.1±0.2 | 99.1±0.1 | 75.4±0.5 | 86.2±0.5 | 94.3±0.3 |
| CTGAN | 6.5±0.0 | 30.5±0.3 | 53.2±0.2 | 80.2±0.4 | 93.1±0.5 | 96.8±0.3 | 2.6±0.3 | 16.4±0.4 | 36.7±1.3 |
| C-CTGAN | 54.2±0.6 | 62.4±0.6 | 73.8±0.5 | 83.2±0.4 | 96.0±0.1 | 98.3±0.1 | 49.2±0.8 | 62.3±1.1 | 72.9±1.4 |
| TVAE | 20.2±0.1 | 42.0±0.2 | 62.4±0.3 | 69.4±0.5 | 90.1±0.4 | 96.5±0.2 | 6.5±0.7 | 37.6±1.6 | 56.7±1.8 |
| C-TVAE | 31.0±0.4 | 56.9±0.5 | 69.7±0.3 | 79.2±0.7 | 92.7±0.4 | 97.1±0.2 | 38.4±1.2 | 57.5±0.4 | 77.4±0.9 |
| GOGGLE | 0.9±0.1 | 29.1±0.2 | 33.1±0.3 | 46.3±0.4 | 99.6±0.1 | 100.0±0.0 | 35.3±0.0 | 82.2±0.0 | 98.9±0.0 |
| C-GOGGLE | 7.7±0.1 | 77.2±0.2 | 99.1±0.1 | 83.9±0.3 | 92.4±0.3 | 97.0±0.1 | 81.1±0.0 | 85.4±0.0 | 89.7±0.0 |
| DGMs | 15.2±0.2 | 43.8±0.3 | 60.9±0.3 | 51.1±0.4 | 92.3±0.3 | 98.5±0.1 | 16.9±0.7 | 53.5±0.9 | 73.9±0.9 |
| C-DGMs | 45.4±0.3 | 69.9±0.4 | 82.9±0.3 | 86.0±0.3 | 95.4±0.2 | 98.3±0.1 | 64.2±0.7 | 74.9±0.6 | 84.8±0.6 |
| Real | 60.1±0.0 | 73.3±0.0 | 85.1±0.0 | 85.7±0.0 | 96.5±0.0 | 98.9±0.0 | 68.4±0.0 | 82.8±0.0 | 98.9±0.0 |

Table 11: Wasserstein distance between numerical features.

| | URL | WiDS | LCLD | Heloc | FSP | News |
|---|---|---|---|---|---|---|
| WGAN | 0.01±0.00 | 0.04±0.00 | 0.02±0.00 | 0.03±0.00 | 0.04±0.00 | 0.02±0.00 |
| P-WGAN | 0.02±0.00 | 0.04±0.00 | 0.02±0.00 | 0.03±0.00 | 0.07±0.00 | 0.02±0.00 |
| C-WGAN | 0.03±0.00 | 0.05±0.00 | 0.02±0.00 | 0.03±0.00 | 0.07±0.00 | 0.02±0.00 |
| TableGAN | 0.01±0.00 | 0.02±0.00 | 0.01±0.00 | 0.01±0.00 | 0.02±0.00 | 0.02±0.00 |
| P-TableGAN | 0.02±0.00 | 0.02±0.00 | 0.02±0.00 | 0.01±0.00 | 0.03±0.00 | 0.02±0.00 |
| C-TableGAN | 0.01±0.00 | 0.02±0.00 | 0.01±0.00 | 0.02±0.00 | 0.02±0.00 | 0.02±0.00 |
| CTGAN | 0.01±0.00 | 0.05±0.00 | 0.03±0.00 | 0.02±0.00 | 0.06±0.01 | 0.01±0.00 |
| P-CTGAN | 0.02±0.00 | 0.05±0.00 | 0.03±0.00 | 0.02±0.00 | 0.08±0.00 | 0.01±0.00 |
| C-CTGAN | 0.01±0.00 | 0.04±0.00 | 0.03±0.00 | 0.02±0.00 | 0.06±0.01 | 0.02±0.00 |
| TVAE | 0.01±0.00 | 0.01±0.00 | 0.02±0.00 | 0.01±0.00 | 0.02±0.00 | 0.01±0.00 |
| P-TVAE | 0.02±0.00 | 0.02±0.00 | 0.02±0.00 | 0.02±0.00 | 0.04±0.00 | 0.01±0.00 |
| C-TVAE | 0.01±0.00 | 0.02±0.00 | 0.01±0.00 | 0.01±0.00 | 0.03±0.00 | 0.01±0.00 |
| GOGGLE | 0.03±0.01 | 0.22±0.10 | 0.04±0.00 | 0.05±0.01 | 0.12±0.01 | 0.08±0.01 |
| P-GOGGLE | 0.03±0.01 | 0.22±0.10 | 0.04±0.00 | 0.05±0.01 | 0.12±0.02 | 0.08±0.01 |
| C-GOGGLE | 0.04±0.01 | 0.05±0.01 | 0.08±0.02 | 0.07±0.02 | 0.14±0.02 | 0.13±0.03 |

performance over the binary and multiclass classification datasets (i.e., all except the News) and the average detection performance over all datasets.

For detection, we can see that the KDE-based ordering performs best for C-WGAN, C-TableGAN and C-CTGAN — according to all three metrics (i.e., F1-score, weighted F1-score, and Area Under the ROC Curve) for the first two, and according to two out of three metrics (i.e. weighted F1-score and Area Under the ROC Curve) for C-CTGAN. On the other hand, the correlation-based ordering yields better results for C-TVAE across all three metrics. Only C-GOGGLE gets best detection results across all three metrics using the random ordering. For utility, we see again that different C-DGM models work best with different orderings. For instance, the KDE-based ordering performs best w.r.t. all three utility metrics for C-WGAN, C-TVAE and C-CTGAN. However, for C-TableGAN and C-GOGGLE, the correlation-based ordering outperforms the other orderings.

Overall, the utility and detection results we obtained using KDE-based or correlation-based orderings as opposed to random ordering highlight the importance of choosing an ordering that takes into account the data distribution and/or the feature relations captured by the C-DGM models in their predictions. Additionally, we can see that a trend emerges for C-WGAN, C-TableGAN, and C-TVAE: each of these models has a clear preference for the ordering that gives the highest overall

Table 12: Jensen-Shannon divergence between categorical features.

|  | URL | WiDS | LCLD | Heloc | FSP | News |
|---|---|---|---|---|---|---|
| WGAN | 0.76±0.00 | 0.79±0.00 | 0.60±0.00 | 0.58±0.01 | 0.33±0.01 | 0.77±0.00 |
| P-WGAN | 0.76±0.00 | 0.79±0.00 | 0.60±0.00 | 0.58±0.01 | 0.33±0.01 | 0.77±0.00 |
| C-WGAN | 0.77±0.00 | 0.80±0.01 | 0.60±0.00 | 0.58±0.00 | 0.33±0.01 | 0.77±0.00 |
| TableGAN | 0.78±0.00 | 0.81±0.00 | 0.69±0.01 | 0.62±0.02 | 0.35±0.01 | 0.77±0.00 |
| P-TableGAN | 0.78±0.00 | 0.81±0.00 | 0.69±0.01 | 0.62±0.02 | 0.35±0.01 | 0.77±0.00 |
| C-TableGAN | 0.78±0.00 | 0.80±0.00 | 0.60±0.01 | 0.59±0.01 | 0.37±0.02 | 0.77±0.00 |
| CTGAN | 0.76±0.00 | 0.75±0.00 | 0.49±0.01 | 0.56±0.00 | 0.33±0.00 | 0.77±0.00 |
| P-CTGAN | 0.76±0.00 | 0.75±0.00 | 0.49±0.01 | 0.56±0.00 | 0.33±0.00 | 0.77±0.00 |
| C-CTGAN | 0.76±0.00 | 0.75±0.00 | 0.49±0.00 | 0.56±0.01 | 0.33±0.00 | 0.77±0.00 |
| TVAE | 0.77±0.00 | 0.79±0.00 | 0.50±0.00 | 0.57±0.00 | 0.33±0.00 | 0.77±0.00 |
| P-TVAE | 0.77±0.00 | 0.79±0.00 | 0.50±0.00 | 0.57±0.00 | 0.33±0.00 | 0.77±0.00 |
| C-TVAE | 0.77±0.00 | 0.80±0.00 | 0.50±0.00 | 0.57±0.01 | 0.33±0.00 | 0.77±0.00 |
| GOGGLE | 0.71±0.02 | 0.76±0.03 | 0.49±0.01 | 0.58±0.01 | 0.37±0.03 | 0.76±0.00 |
| P-GOGGLE | 0.71±0.02 | 0.76±0.03 | 0.49±0.01 | 0.58±0.01 | 0.37±0.03 | 0.76±0.00 |
| C-GOGGLE | 0.71±0.01 | 0.82±0.01 | 0.49±0.01 | 0.57±0.01 | 0.39±0.01 | 0.76±0.01 |

Table 13: Variable orderings comparison for C-DGMs. The best results are in bold.

|  |  | Utility (↑) | | | Detection (↓) | | |
|---|---|---|---|---|---|---|---|
|  |  | F1 | *w*F1 | AUC | F1 | *w*F1 | AUC |
| C-WGAN | Rnd | 0.453 | 0.477 | 0.735 | 0.936 | 0.933 | 0.950 |
|  | Corr | 0.475 | 0.497 | 0.746 | 0.919 | **0.915** | 0.936 |
|  | KDE | **0.485** | **0.504** | **0.748** | **0.917** | **0.915** | **0.935** |
| C-TableGAN | Rnd | 0.346 | 0.409 | 0.710 | 0.908 | 0.904 | 0.926 |
|  | Corr | **0.361** | **0.422** | **0.717** | 0.897 | **0.893** | **0.916** |
|  | KDE | 0.349 | 0.413 | 0.706 | **0.895** | **0.893** | **0.916** |
| C-CTGAN | Rnd | 0.509 | 0.530 | 0.770 | 0.894 | 0.896 | 0.924 |
|  | Corr | 0.513 | 0.536 | **0.773** | **0.887** | 0.891 | 0.920 |
|  | KDE | **0.516** | **0.537** | **0.773** | 0.888 | **0.889** | **0.919** |
| C-TVAE | Rnd | 0.487 | 0.522 | 0.766 | 0.873 | 0.870 | 0.901 |
|  | Corr | **0.504** | 0.534 | 0.771 | **0.868** | **0.868** | **0.898** |
|  | KDE | **0.504** | **0.536** | **0.774** | 0.875 | 0.875 | 0.905 |
| C-GOGGLE | Rnd | 0.384 | 0.406 | 0.653 | **0.922** | **0.916** | **0.937** |
|  | Corr | **0.409** | **0.424** | **0.661** | 0.929 | 0.918 | 0.940 |
|  | KDE | 0.393 | 0.416 | **0.661** | 0.928 | 0.926 | 0.941 |

improvements w.r.t. utility and detection. It is also worth noticing that it is not always the case that an ordering improves both utility and detection, as we have seen for C-TableGAN and C-TVAE.

We also show the effect of using different orderings on P-DGM models, in Table 14. As opposed to the trends we noticed for our C-DGMs, here we find that it is harder to establish clear patterns in the preference of the P-DGMs towards any of the orderings. This is mainly due to the very small differences between the results, as it can be seen from the Table.

**Exploring other variable orderings.** In our work, we explored two variable orderings and conducted extensive experiments with them. However, there are various other ways to define these orderings. One advantage of customizing the order is that it can be tailored to the user's needs. For example, if users require orderings that consider how closely the generated data distribution matches the real data distribution, there are numerous ways to achieve this. In our detailed analysis, we examined a KDE-based ordering that compares the joint distributions of the features. However, a simpler alternative method would be to use the Wasserstein distance, a metric discussed in Section C.2,

Table 14: Variable orderings comparison for P-DGM. Best results are in bold.

|  |  | Utility ($\uparrow$) | | | Detection ($\downarrow$) | | |
|---|---|---|---|---|---|---|---|
|  |  | F1 | *w*F1 | AUC | F1 | *w*F1 | AUC |
| P-WGAN | Rnd | 0.456 | 0.484 | 0.731 | **0.930** | **0.928** | **0.945** |
|  | Corr | 0.462 | **0.489** | **0.732** | 0.931 | 0.930 | 0.947 |
|  | KDE | **0.463** | **0.489** | 0.731 | 0.933 | 0.931 | 0.948 |
| P-TableGAN | Rnd | **0.328** | 0.398 | 0.705 | 0.902 | 0.901 | 0.925 |
|  | Corr | **0.328** | **0.399** | 0.703 | **0.900** | **0.900** | **0.922** |
|  | KDE | 0.326 | 0.398 | **0.706** | 0.901 | 0.901 | 0.924 |
| P-CTGAN | Rnd | 0.507 | 0.524 | 0.769 | **0.898** | **0.901** | **0.926** |
|  | Corr | **0.508** | **0.527** | **0.770** | 0.899 | 0.902 | 0.928 |
|  | KDE | 0.507 | 0.524 | 0.769 | 0.899 | 0.903 | 0.927 |
| P-TVAE | Rnd | 0.490 | 0.521 | 0.764 | 0.879 | 0.879 | 0.905 |
|  | Corr | 0.493 | 0.523 | 0.763 | **0.876** | 0.879 | 0.904 |
|  | KDE | **0.494** | **0.524** | **0.767** | **0.876** | **0.875** | **0.901** |
| P-GOGGLE | Rnd | 0.347 | 0.373 | **0.626** | 0.925 | 0.925 | 0.943 |
|  | Corr | **0.348** | **0.375** | 0.625 | 0.929 | 0.926 | 0.945 |
|  | KDE | 0.347 | 0.374 | **0.626** | 0.929 | 0.927 | 0.944 |

which differs from the KDE approach in that it compares individual distributions of the features rather than joint ones. In yet another scenario, users could define orderings without relying on the outputs of the unconstrained model by considering relations between the features in the real data, which is a different approach to all the previously-mentioned orderings. For instance, one way to do this is to order the variables based on the cause-effect relations between them, which we refer to as a causal-based ordering. More precisely, in such an ordering, there is no instance of a cause-effect pair of features $(x_i, x_j)$ in which $x_j$ appears before $x_i$ in the ordering.

To assess the performance of our layer when constructed using different orderings, we conducted experiments with Wasserstein- and causal-based orderings on the WiDS dataset. For the Wasserstein-based ordering, for each feature we calculated the Wasserstein distance between the real data distribution and the generated data distribution obtained from an unconstrained DGM. We then arranged the features in ascending order based on the calculated distances, such that the features with generated data distributions furthest from the real data distributions would be corrected last by our constraint layer.

We note that the Wasserstein distance can only be defined on a metric space, making it impractical for use with categorical features without additional modifications (i.e. it is possible to define distance metrics for each of the categorical features and then apply Wasserstein distance on these features). However, here we assume that the constraint layer can only be applied on continuous features and, thus, the position of the categorical features in the orderings will not impact the final results, as our constraints exclude such features. As an alternative, we investigated using the Jensen-Shannon divergence to derive another ordering that compares individual feature distributions. And while this method offers has the advantage that it can be applied on both continuous and categorical features, we found that it was unstable in its results, aligning with the observations made by Zhao et al. (2021).

To compute the causal-based ordering, we obtained a directed acyclic graph (DAG), capturing the cause-effect relations between the features, and topologically sorted the features. Since none of the datasets we experimented with provided any information on causal relations between features, we utilized DAG-GNN (Yu et al., 2019) to obtain such relations in the training partition of each dataset. DAG-GNN is a gradient-based causal discovery method employing graph neural networks to learn a DAG structure which encodes cause-effect relations. Importantly for our application, we require that the graph has a topological ordering, implying the graph should be a DAG. However, determining a DAG structure poses a challenge in the causality domain. In fact, the DAG-GNN method does not guarantee that its output structure is a DAG, as the DAG constraint is embedded in a loss term minimised during training. In our experiments, we observed that DAG-GNN produced a few self-loops (edges connecting a node to itself) for most datasets and one cycle for half of the datasets.

Table 15: Comparing DGMs with their C-DGM versions using 5 different orderings, when trained on the WiDS dataset. Best results are in bold.

| | | Utility (↑) | | | Detection (↓) | | |
|---|---|---|---|---|---|---|---|
| | | F1 | wF1 | AUC | F1 | wF1 | AUC |
| C-WGAN | Rnd | 0.303 | 0.360 | 0.797 | 0.995 | 0.996 | 0.999 |
| | Corr | 0.284 | 0.343 | 0.796 | 0.975 | 0.976 | 0.989 |
| | KDE | **0.316** | **0.372** | **0.815** | 0.975 | 0.975 | 0.989 |
| | WD | 0.273 | 0.331 | 0.770 | **0.932** | **0.925** | **0.948** |
| | Caus | 0.275 | 0.334 | 0.775 | 0.944 | 0.929 | 0.956 |
| C-TableGAN | Rnd | 0.213 | 0.279 | **0.777** | 0.984 | 0.984 | 0.996 |
| | Corr | **0.246** | **0.309** | 0.775 | **0.956** | **0.957** | **0.974** |
| | KDE | 0.208 | 0.274 | 0.767 | 0.962 | 0.963 | 0.979 |
| | WD | 0.242 | 0.305 | 0.770 | 0.961 | 0.962 | 0.977 |
| | Caus | 0.225 | 0.289 | 0.770 | 0.962 | 0.963 | 0.979 |
| C-CTGAN | Rnd | 0.364 | 0.408 | 0.836 | 0.990 | 0.990 | 0.997 |
| | Corr | 0.365 | 0.409 | 0.826 | 0.988 | 0.988 | 0.995 |
| | KDE | 0.360 | 0.403 | 0.832 | 0.986 | 0.986 | 0.994 |
| | WD | **0.368** | **0.411** | **0.842** | **0.953** | **0.955** | **0.970** |
| | Caus | 0.365 | 0.409 | 0.838 | 0.954 | 0.956 | **0.970** |
| C-TVAE | Rnd | 0.248 | 0.311 | 0.773 | 0.965 | 0.965 | 0.982 |
| | Corr | 0.305 | 0.363 | 0.804 | 0.959 | 0.960 | 0.977 |
| | KDE | **0.321** | **0.378** | **0.816** | 0.961 | 0.962 | 0.979 |
| | WD | 0.297 | 0.356 | 0.794 | **0.958** | **0.959** | **0.976** |
| | Caus | 0.316 | 0.373 | 0.808 | **0.958** | **0.959** | **0.976** |
| C-GOGGLE | Rnd | 0.139 | 0.210 | 0.643 | **0.965** | **0.965** | **0.979** |
| | Corr | 0.185 | 0.253 | 0.675 | 0.972 | 0.971 | 0.984 |
| | KDE | 0.171 | 0.239 | 0.678 | 0.975 | 0.975 | 0.984 |
| | WD | **0.207** | **0.273** | **0.705** | 0.980 | 0.980 | 0.990 |
| | Caus | 0.175 | 0.244 | 0.681 | 0.966 | **0.965** | **0.979** |

To address these issues, we eliminated the self-loops and broke the cycles by randomly selecting an edge from each cycle and removing it from the graph. This approach enabled us to obtain a DAG structure and, consequently, a topological ordering that we eventually used as the causal-based ordering in our experiments.

For our experiments we considered the medium-sized datasets and selected WiDS. Our choice was mainly guided by the levels of difficulty involved when manually checking the causality relations found by the DAG-GNN model. Compared to the other datasets, WiDS' columns had a clearer and more explanatory description, which allowed us to determine whether the cause-effect relations between the columns were sensible. In Tables 15 and 16 we compared the 5 different orderings on the WiDS dataset for all C-DGMs and P-DGMs, respectively. As we can see in Table 15, both the causal- and Wasserstein distance-based orderings achieved better results than the random ordering in 12 (and 11) out of 15 cases for detection (and utility). On the other hand, we notice that when used during postprocessing, these two new orderings can only bring small performance improvements. However, similarly to the other orderings we explored, it is harder to distinguish trends where the new orderings might be more helpful than the random ordering.

**Future work on variable orderings.** As we can see, using custom-made orderings can improve the performance of C-DGMs which use a random ordering of the variables w.r.t. both utility and detection. However, it is not straightforward to determine which ordering works best in which scenario. To leverage the constraint layer's capabilities to the maximum, for future work we plan to investigate when different feature orderings might work best and uncover patterns in the models' and datasets' preferences towards certain orderings.

Table 16: Comparing DGMs with their P-DGM versions using 5 different orderings, when trained on the WiDS dataset. Best results are in bold.

| | | Utility (↑) | | | Detection (↓) | | |
|---|---|---|---|---|---|---|---|
| | | F1 | *w*F1 | AUC | F1 | *w*F1 | AUC |
| P-WGAN | Rnd | 0.319 | 0.372 | 0.777 | 0.979 | 0.980 | 0.992 |
| | Corr | 0.332 | 0.384 | 0.781 | **0.976** | **0.977** | **0.991** |
| | KDE | **0.334** | **0.386** | **0.783** | 0.978 | 0.978 | **0.991** |
| | WD | 0.332 | 0.384 | 0.776 | 0.979 | 0.979 | 0.992 |
| | Caus | 0.332 | 0.383 | 0.774 | 0.980 | 0.981 | 0.992 |
| P-TableGAN | Rnd | 0.173 | 0.242 | **0.749** | **0.962** | **0.963** | 0.980 |
| | Corr | 0.174 | 0.242 | 0.731 | 0.963 | 0.964 | 0.981 |
| | KDE | 0.171 | 0.240 | 0.735 | **0.962** | **0.963** | 0.980 |
| | WD | **0.181** | **0.250** | 0.738 | **0.962** | **0.963** | 0.979 |
| | Caus | 0.171 | 0.240 | 0.741 | 0.963 | 0.964 | 0.980 |
| P-CTGAN | Rnd | 0.364 | 0.407 | **0.837** | 0.991 | 0.991 | **0.997** |
| | Corr | 0.365 | 0.407 | 0.835 | **0.989** | **0.989** | **0.997** |
| | KDE | 0.357 | 0.400 | 0.835 | 0.991 | 0.991 | **0.997** |
| | WD | **0.375** | **0.419** | 0.836 | 0.990 | 0.990 | **0.997** |
| | Caus | 0.363 | 0.406 | 0.835 | 0.990 | 0.990 | **0.997** |
| P-TVAE | Rnd | 0.266 | 0.327 | 0.785 | **0.962** | 0.963 | **0.978** |
| | Corr | 0.282 | 0.342 | 0.787 | **0.962** | **0.962** | 0.979 |
| | KDE | **0.285** | **0.345** | **0.797** | 0.963 | 0.964 | 0.979 |
| | WD | 0.280 | 0.340 | 0.778 | **0.962** | **0.962** | 0.979 |
| | Caus | 0.269 | 0.329 | 0.793 | **0.962** | **0.962** | 0.979 |
| P-GOGGLE | Rnd | **0.192** | **0.202** | 0.665 | 0.988 | 0.988 | **0.993** |
| | Corr | 0.191 | 0.201 | 0.667 | 0.988 | 0.988 | **0.993** |
| | KDE | 0.189 | 0.197 | 0.667 | **0.987** | **0.987** | **0.993** |
| | WD | 0.191 | 0.200 | 0.663 | 0.988 | 0.988 | **0.993** |
| | Caus | 0.191 | 0.200 | **0.669** | 0.988 | 0.988 | 0.994 |

## C.4 Full results on DGMs vs. C-DGMs

To assess the impact of constraining DGMs with our method, we conducted an extensive experimental analysis where we compared the performance of our C-DGM models with the baseline DGM models in terms of utility and detection, as specified in Section 4.1. In Tables 17-23 we present full utility and detection results. More specifically, in each table we report results for each dataset, each DGM and each C-DGM, using 3 different orderings (i.e., random, correlation-based and KDE-based). Additionally, for each result in the table, we report the standard deviation from the mean. As a reminder, we obtained the results by running each experiment with 5 different seeds. As shown, in most cases at least one of the orderings used for the C-DGMs outperforms the DGMs, particularly for detection.

Comparing WGAN with C-WGAN, we note that the only dataset for which we do not get an improvement in utility with either of the three different metrics is LCLD. However, the gap between the WGAN and the best C-WGAN is small for all the 3 different metrics: 0.7%, 0.1%, 0.2% for F1-score, weighted F1-score and Area Under the ROC Curve, respectively. In most cases, with C-WGAN we see significant utility improvements, e.g. 7.7%, 5% and 3.8% for Heloc in F1-score, weighted F1-score and Area Under the ROC Curve, respectively.

For TableGAN, we see that C-TableGAN outperforms the unconstrained models for all binary and multiclass classification datasets in terms of utility, according to at least 2 out of 3 metrics. For the cases where the performance is not improved under some metric, we notice that the difference in performance between TableGAN and C-TableGAN is small, e.g., 0.5% for the FSP dataset and 0.2% for the Heloc dataset in Area Under the ROC Curve. It is worth noticing that out of all the C-DGMs, C-TableGAN brings the highest improvements overall. For example, C-TableGAN outperforms the

unconstrained TableGAN by at least 4.5% according to the F1-score in 4 datasets, with the highest improvement, of 7.5%, recorded for WiDS.

As opposed to C-TableGAN, C-CTGAN does not show the same trend, with most models giving either moderate improvements or performing close to the unconstrained CTGAN. However, it is still the case that most of the datasets (four out of six) show improvements over all three metrics (with the remaining two datasets showing improvements on two out of three metrics). Notably, with C-CTGAN we did observe a large improvement of 64.9 points in the mean absolute error for utility when training on the News dataset, which is the highest improvement we observed with any of the C-DGM models.

Similarly, to the C-WGAN case, there is only one dataset for which C-TVAE does not improve the utility performance in any of the three metrics, namely the Heloc dataset. Nevertheless, the difference between the overall best C-TVAE model (i.e. using the KDE ordering) and TVAE is small for all the 3 different metrics: 0.4%, 0.2%, 0.1% for F1-score, weighted F1-score and Area Under the ROC Curve, respectively. In most cases, with C-TVAE we see major utility improvements, e.g. 4.0%, 3.6% and 1.5% for WiDS in F1-score, weighted F1-score and Area Under the ROC Curve, respectively, and 2.6%, 2.9% and 1.4% for FSP in F1-score, weighted F1-score and Area Under the ROC Curve, respectively.

For C-GOGGLE, we see that five out of six datasets outperform the standard GOGGLE results on all three metrics, giving major improvements (with the largest improvement being of 17.1% in the F1-score for utility when training on the URL dataset). Only the FSP dataset does not show any improvements on any of the orderings when using the C-GOGGLE version.

For detection, with C-WGAN we get an improvement (or do not change the performance, as it happens in a few cases) over all datasets in all metrics, the only exception being the Area Under the ROC Curve for the URL dataset. With C-TableGAN, C-CTGAN, and C-TVAE, we see an improvement in 4 out of 6 datasets, where we outperform the unconstrained models (i) according to all metrics for 3 datasets and (ii) according to at least two out of three metrics for 1 dataset, respectively. With C-GOGGLE we see again an improvement in 4 out of 6 datasets, where we outperform the unconstrained models (i) according to all metrics for 2 datasets and (ii) according to at least one out of three metrics for 2 datasets, respectively.

### C.4.1    DGMS VS. C-DGMS: UTILITY RESULTS

In Tables 17-19 we present the utility results on all datasets using 3 metrics in the following order: F1-score, weighted F1-score, and Area Under the ROC Curve. Separately, in Table 20, we report the performance of real and synthetic data for the News dataset (which is a regression dataset), using 4 different metrics: Explained Variance (XV) and Mean Absolute Error (MAE).

### C.4.2    DGMS VS. C-DGMS: DETECTION RESULTS

In Tables 21-23 we present the detection results on all datasets using 3 metrics in the following order: F1-score, weighted F1-score and Area Under the ROC Curve.

### C.5    FULL RESULTS ON DGMS VS. P-DGMS

Our constrained layer uses the constraints to correct the predictions both at training and inference time, however, the constraints can simply be used at inference time to correct the predictions only once. This latter approach, which we call P-DGM, is a way of putting guardrails on the output space of the DGMs and could be the preferred option for users who do not want to make any modifications to their models or retrain them. Both methods guarantee that the constraints are satisfied by the predictions, however, their impact on the models' performance might differ. To see this, we compared DGMs with P-DGMs whose predictions have been corrected according to three different label orderings: random, correlation-based, and KDE-based. In each of the following tables, we report the mean and standard deviation from the mean.

### C.5.1 DGMs vs. P-DGMs: Utility Results

In Tables 24, 25 and 26 we compare the performance of the DGM and P-DGM models in terms of utility and report the results w.r.t. 3 different metrics: F1-score, weighted F1-score, and Area Under the ROC Curve. Separately, in Table 27, we report the performance of the synthetic data for the News dataset, using two metrics: XV and MAE.

As we can see, putting guardrails on the output space of the models can help increase the performance, however, the overall gains are not as high as those used by our C-DGM models which correct the outputs at training time.

### C.5.2 DGMs vs. P-DGMs: Detection results

In Tables 28, 29 and 30 we compare the performance of the DGM and P-DGM models in terms of detection. Similarly to the utility case, post-processing the outputs of the unconstrained models can help slightly increase the overall performance.

Table 17: Binary (macro) F1-score utility results with their corresponding stddevs for binary (multiclass) classification datasets.

|  |  | URL | WiDS | LCLD | Heloc | FSP |
|---|---|---|---|---|---|---|
| WGAN | - | $0.756 \pm 0.030$ | $\mathbf{0.329} \pm 0.059$ | $\mathbf{0.239} \pm 0.026$ | $0.634 \pm 0.099$ | $0.357 \pm 0.020$ |
|  | Rnd | $0.792 \pm 0.021$ | $0.302 \pm 0.075$ | $0.196 \pm 0.026$ | $0.643 \pm 0.060$ | $0.333 \pm 0.028$ |
| C-WGAN | Corr | $0.790 \pm 0.026$ | $0.284 \pm 0.040$ | $0.232 \pm 0.050$ | $0.704 \pm 0.039$ | $\mathbf{0.367} \pm 0.027$ |
|  | KDE | $\mathbf{0.801} \pm 0.004$ | $0.316 \pm 0.044$ | $0.232 \pm 0.050$ | $\mathbf{0.711} \pm 0.030$ | $\mathbf{0.367} \pm 0.027$ |
| TableGAN | - | $0.562 \pm 0.051$ | $0.171 \pm 0.037$ | $0.123 \pm 0.041$ | $0.593 \pm 0.058$ | $0.199 \pm 0.044$ |
|  | Rnd | $0.534 \pm 0.096$ | $0.213 \pm 0.027$ | $0.148 \pm 0.058$ | $0.636 \pm 0.073$ | $0.199 \pm 0.031$ |
| C-TableGAN | Corr | $\mathbf{0.611} \pm 0.111$ | $\mathbf{0.246} \pm 0.047$ | $0.130 \pm 0.036$ | $\mathbf{0.638} \pm 0.061$ | $0.179 \pm 0.036$ |
|  | KDE | $0.576 \pm 0.074$ | $0.207 \pm 0.043$ | $\mathbf{0.174} \pm 0.063$ | $0.582 \pm 0.114$ | $\mathbf{0.208} \pm 0.053$ |
| CTGAN | - | $0.822 \pm 0.017$ | $0.362 \pm 0.033$ | $0.247 \pm 0.087$ | $0.736 \pm 0.035$ | $0.374 \pm 0.034$ |
|  | Rnd | $0.833 \pm 0.006$ | $\mathbf{0.365} \pm 0.019$ | $0.235 \pm 0.057$ | $\mathbf{0.744} \pm 0.010$ | $0.370 \pm 0.006$ |
| C-CTGAN | Corr | $0.830 \pm 0.009$ | $\mathbf{0.365} \pm 0.020$ | $0.260 \pm 0.081$ | $0.730 \pm 0.027$ | $\mathbf{0.383} \pm 0.019$ |
|  | KDE | $\mathbf{0.836} \pm 0.002$ | $0.360 \pm 0.022$ | $\mathbf{0.265} \pm 0.040$ | $0.736 \pm 0.010$ | $0.381 \pm 0.030$ |
| TVAE | - | $0.810 \pm 0.008$ | $0.282 \pm 0.029$ | $\mathbf{0.185} \pm 0.021$ | $\mathbf{0.735} \pm 0.010$ | $0.473 \pm 0.016$ |
|  | Rnd | $0.824 \pm 0.004$ | $0.249 \pm 0.038$ | $0.143 \pm 0.018$ | $0.720 \pm 0.014$ | $\mathbf{0.501} \pm 0.012$ |
| C-TVAE | Corr | $\mathbf{0.826} \pm 0.007$ | $0.305 \pm 0.021$ | $0.158 \pm 0.011$ | $0.733 \pm 0.017$ | $0.496 \pm 0.018$ |
|  | KDE | $0.824 \pm 0.004$ | $\mathbf{0.322} \pm 0.041$ | $0.146 \pm 0.023$ | $0.731 \pm 0.008$ | $0.498 \pm 0.014$ |
| GOGGLE | - | $0.622 \pm 0.094$ | $\mathbf{0.189} \pm 0.038$ | $0.163 \pm 0.119$ | $0.596 \pm 0.072$ | $\mathbf{0.152} \pm 0.003$ |
|  | Rnd | $0.782 \pm 0.035$ | $0.139 \pm 0.068$ | $0.157 \pm 0.085$ | $\mathbf{0.723} \pm 0.018$ | $0.117 \pm 0.008$ |
| C-GOGGLE | Corr | $\mathbf{0.793} \pm 0.013$ | $0.185 \pm 0.108$ | $\mathbf{0.219} \pm 0.023$ | $0.714 \pm 0.013$ | $0.136 \pm 0.024$ |
|  | KDE | $0.788 \pm 0.014$ | $0.171 \pm 0.036$ | $0.167 \pm 0.082$ | $0.708 \pm 0.025$ | $0.134 \pm 0.016$ |

Table 18: Weighted F1-score utility results with their corresponding stddevs for binary and multi-class classification datasets.

| | | URL | WiDS | LCLD | Heloc | FSP |
|---|---|---|---|---|---|---|
| WGAN | - | $0.764\pm0.018$ | $\mathbf{0.381}\pm0.057$ | $\mathbf{0.359}\pm0.019$ | $0.599\pm0.050$ | $0.339\pm0.021$ |
| | Rnd | $0.782\pm0.025$ | $0.360\pm0.068$ | $0.339\pm0.020$ | $0.585\pm0.079$ | $0.316\pm0.029$ |
| C-WGAN | Corr | $0.785\pm0.011$ | $0.344\pm0.036$ | $0.358\pm0.037$ | $0.648\pm0.022$ | $\mathbf{0.349}\pm0.028$ |
| | KDE | $\mathbf{0.790}\pm0.007$ | $0.373\pm0.040$ | $0.358\pm0.037$ | $\mathbf{0.649}\pm0.011$ | $\mathbf{0.349}\pm0.028$ |
| TableGAN | - | $0.659\pm0.035$ | $0.240\pm0.034$ | $0.286\pm0.029$ | $0.615\pm0.030$ | $0.199\pm0.043$ |
| | Rnd | $0.644\pm0.061$ | $0.280\pm0.025$ | $0.306\pm0.043$ | $0.618\pm0.042$ | $0.198\pm0.030$ |
| C-TableGAN | Corr | $\mathbf{0.695}\pm0.071$ | $\mathbf{0.309}\pm0.042$ | $0.292\pm0.029$ | $\mathbf{0.633}\pm0.036$ | $0.180\pm0.034$ |
| | KDE | $0.670\pm0.052$ | $0.274\pm0.039$ | $\mathbf{0.314}\pm0.029$ | $0.596\pm0.048$ | $\mathbf{0.209}\pm0.051$ |
| CTGAN | - | $0.799\pm0.033$ | $0.405\pm0.034$ | $0.379\pm0.061$ | $0.675\pm0.015$ | $0.372\pm0.034$ |
| | Rnd | $0.818\pm0.008$ | $0.408\pm0.019$ | $0.369\pm0.044$ | $0.684\pm0.005$ | $0.369\pm0.007$ |
| C-CTGAN | Corr | $0.816\pm0.012$ | $\mathbf{0.409}\pm0.019$ | $0.388\pm0.060$ | $0.688\pm0.010$ | $0.379\pm0.020$ |
| | KDE | $\mathbf{0.820}\pm0.008$ | $0.403\pm0.023$ | $\mathbf{0.392}\pm0.030$ | $\mathbf{0.690}\pm0.010$ | $\mathbf{0.380}\pm0.032$ |
| TVAE | - | $0.802\pm0.012$ | $0.342\pm0.027$ | $\mathbf{0.330}\pm0.016$ | $\mathbf{0.696}\pm0.006$ | $0.463\pm0.017$ |
| | Rnd | $\mathbf{0.817}\pm0.007$ | $0.311\pm0.034$ | $0.301\pm0.012$ | $0.687\pm0.006$ | $\mathbf{0.492}\pm0.013$ |
| C-TVAE | Corr | $0.816\pm0.007$ | $0.363\pm0.019$ | $0.311\pm0.011$ | $0.690\pm0.009$ | $0.488\pm0.019$ |
| | KDE | $0.816\pm0.008$ | $\mathbf{0.378}\pm0.038$ | $0.305\pm0.016$ | $0.694\pm0.003$ | $0.490\pm0.014$ |
| GOGGLE | - | $0.648\pm0.074$ | $0.198\pm0.067$ | $0.315\pm0.086$ | $0.566\pm0.050$ | $\mathbf{0.139}\pm0.002$ |
| | Rnd | $0.741\pm0.032$ | $0.210\pm0.061$ | $0.314\pm0.065$ | $\mathbf{0.663}\pm0.012$ | $0.103\pm0.008$ |
| C-GOGGLE | Corr | $\mathbf{0.752}\pm0.024$ | $\mathbf{0.253}\pm0.098$ | $\mathbf{0.357}\pm0.020$ | $0.637\pm0.023$ | $0.121\pm0.024$ |
| | KDE | $0.749\pm0.029$ | $0.239\pm0.032$ | $0.318\pm0.060$ | $0.656\pm0.014$ | $0.119\pm0.017$ |

Table 19: Area under the ROC curve utility results with their corresponding stddevs for binary and multiclass classification datasets.

| | | URL | WiDS | LCLD | Heloc | FSP |
|---|---|---|---|---|---|---|
| WGAN | - | $0.839\pm0.016$ | $0.775\pm0.038$ | $\mathbf{0.618}\pm0.011$ | $0.677\pm0.033$ | $\mathbf{0.742}\pm0.007$ |
| | Rnd | $0.860\pm0.018$ | $0.798\pm0.028$ | $0.616\pm0.009$ | $0.672\pm0.037$ | $0.728\pm0.010$ |
| C-WGAN | Corr | $\mathbf{0.865}\pm0.011$ | $0.796\pm0.017$ | $0.612\pm0.016$ | $0.714\pm0.017$ | $\mathbf{0.742}\pm0.022$ |
| | KDE | $0.858\pm0.011$ | $\mathbf{0.816}\pm0.015$ | $0.612\pm0.016$ | $\mathbf{0.715}\pm0.011$ | $\mathbf{0.742}\pm0.022$ |
| TableGAN | - | $0.843\pm0.020$ | $0.740\pm0.021$ | $0.587\pm0.027$ | $\mathbf{0.707}\pm0.007$ | $\mathbf{0.642}\pm0.026$ |
| | Rnd | $0.853\pm0.018$ | $\mathbf{0.778}\pm0.017$ | $0.593\pm0.019$ | $0.692\pm0.029$ | $0.637\pm0.022$ |
| C-TableGAN | Corr | $\mathbf{0.868}\pm0.007$ | $0.775\pm0.015$ | $\mathbf{0.605}\pm0.016$ | $0.705\pm0.030$ | $0.630\pm0.037$ |
| | KDE | $0.865\pm0.010$ | $0.767\pm0.016$ | $0.587\pm0.017$ | $0.676\pm0.037$ | $0.635\pm0.028$ |
| CTGAN | - | $0.859\pm0.040$ | $0.835\pm0.012$ | $\mathbf{0.651}\pm0.020$ | $0.744\pm0.009$ | $\mathbf{0.760}\pm0.014$ |
| | Rnd | $\mathbf{0.880}\pm0.004$ | $\mathbf{0.837}\pm0.003$ | $0.626\pm0.028$ | $0.749\pm0.008$ | $0.757\pm0.008$ |
| C-CTGAN | Corr | $0.877\pm0.010$ | $0.827\pm0.013$ | $0.643\pm0.015$ | $\mathbf{0.755}\pm0.007$ | $\mathbf{0.760}\pm0.009$ |
| | KDE | $\mathbf{0.880}\pm0.007$ | $0.833\pm0.007$ | $0.641\pm0.015$ | $0.751\pm0.011$ | $0.759\pm0.012$ |
| TVAE | - | $0.863\pm0.011$ | $0.800\pm0.016$ | $0.631\pm0.004$ | $\mathbf{0.752}\pm0.003$ | $0.789\pm0.007$ |
| | Rnd | $0.876\pm0.008$ | $0.773\pm0.036$ | $0.630\pm0.002$ | $0.750\pm0.005$ | $\mathbf{0.803}\pm0.009$ |
| C-TVAE | Corr | $0.878\pm0.005$ | $0.803\pm0.014$ | $\mathbf{0.633}\pm0.003$ | $0.749\pm0.005$ | $0.791\pm0.015$ |
| | KDE | $\mathbf{0.879}\pm0.007$ | $\mathbf{0.815}\pm0.028$ | $0.632\pm0.003$ | $0.751\pm0.002$ | $0.794\pm0.009$ |
| GOGGLE | - | $0.742\pm0.071$ | $0.656\pm0.049$ | $0.543\pm0.039$ | $0.600\pm0.056$ | $\mathbf{0.577}\pm0.010$ |
| | Rnd | $0.800\pm0.029$ | $0.643\pm0.088$ | $0.569\pm0.055$ | $\mathbf{0.719}\pm0.005$ | $0.535\pm0.016$ |
| C-GOGGLE | Corr | $0.794\pm0.030$ | $0.675\pm0.051$ | $\mathbf{0.593}\pm0.046$ | $0.696\pm0.022$ | $0.546\pm0.011$ |
| | KDE | $\mathbf{0.802}\pm0.016$ | $\mathbf{0.678}\pm0.029$ | $0.572\pm0.051$ | $0.713\pm0.021$ | $0.538\pm0.020$ |

Table 20: Utility performance comparison between DGM and C-DGM models trained on News, using XV and MAE. In the first row, we report the real data performance.

|  |  | XV | MAE |
|---|---|---|---|
| Real data | - | $-0.001 \pm 0.001$ | $3001.1 \pm 55.0$ |
| WGAN | - | $-0.006 \pm 0.003$ | $3133.6 \pm 242.1$ |
| | Rnd | $-0.008 \pm 0.012$ | $3093.0 \pm 202.3$ |
| C-WGAN | Corr | $\mathbf{-0.002} \pm 0.002$ | $\mathbf{3014.0} \pm 79.1$ |
| | KDE | $-0.004 \pm 0.001$ | $3070.0 \pm 98.4$ |
| TableGAN | - | $\mathbf{-0.001} \pm 0.001$ | $\mathbf{2992.2} \pm 35.8$ |
| | Rnd | $\mathbf{-0.001} \pm 0.001$ | $3033.1 \pm 53.9$ |
| C-TableGAN | Corr | $-0.002 \pm 0.002$ | $3015.9 \pm 40$ |
| | KDE | $-0.002 \pm 0.002$ | $3016.8 \pm 70.8$ |
| CTGAN | - | $\mathbf{-0.002} \pm 0.001$ | $3043.8 \pm 37.8$ |
| | Rnd | $\mathbf{-0.002} \pm 0.001$ | $3034.6 \pm 29.4$ |
| C-CTGAN | Corr | $\mathbf{-0.002} \pm 0.001$ | $\mathbf{2978.9} \pm 28.7$ |
| | KDE | $-0.003 \pm 0.001$ | $3029.1 \pm 72.7$ |
| TVAE | - | $\mathbf{-0.001} \pm 0.000$ | $3021.3 \pm 10.0$ |
| | Rnd | $-0.002 \pm 0.001$ | $3067.7 \pm 71.5$ |
| C-TVAE | Corr | $\mathbf{-0.001} \pm 0.001$ | $\mathbf{3005.0} \pm 56.2$ |
| | KDE | $-0.002 \pm 0.001$ | $3011.7 \pm 44.2$ |
| GOGGLE | - | $-0.004 \pm 0.003$ | $3054.5 \pm 44.7$ |
| | Rnd | $\mathbf{-0.001} \pm 0.001$ | $3042.6 \pm 54.1$ |
| C-GOGGLE | Corr | $\mathbf{-0.001} \pm 0.001$ | $3026.0 \pm 34.6$ |
| | KDE | $\mathbf{-0.001} \pm 0.000$ | $\mathbf{2999.0} \pm 27.3$ |

Table 21: Binary F1-score detection results with their corresponding stddevs for all datasets.

|  |  | URL | WiDS | LCLD | Heloc | FSP | News |
|---|---|---|---|---|---|---|---|
| WGAN | - | $0.865 \pm 0.035$ | $\mathbf{0.975} \pm 0.002$ | $0.999 \pm 0.001$ | $0.964 \pm 0.004$ | $0.914 \pm 0.018$ | $0.954 \pm 0.022$ |
| | Rnd | $\mathbf{0.864} \pm 0.046$ | $0.996 \pm 0.003$ | $\mathbf{0.916} \pm 0.013$ | $0.970 \pm 0.024$ | $0.900 \pm 0.078$ | $0.969 \pm 0.015$ |
| C-WGAN | Corr | $0.879 \pm 0.027$ | $\mathbf{0.975} \pm 0.006$ | $0.923 \pm 0.005$ | $0.964 \pm 0.007$ | $\mathbf{0.843} \pm 0.037$ | $0.928 \pm 0.023$ |
| | KDE | $0.877 \pm 0.016$ | $\mathbf{0.975} \pm 0.002$ | $0.923 \pm 0.005$ | $\mathbf{0.957} \pm 0.018$ | $\mathbf{0.843} \pm 0.037$ | $\mathbf{0.926} \pm 0.011$ |
| TableGAN | - | $0.831 \pm 0.029$ | $0.963 \pm 0.006$ | $0.895 \pm 0.024$ | $\mathbf{0.923} \pm 0.011$ | $0.909 \pm 0.021$ | $\mathbf{0.927} \pm 0.009$ |
| | Rnd | $0.840 \pm 0.020$ | $0.984 \pm 0.002$ | $0.870 \pm 0.016$ | $0.963 \pm 0.021$ | $0.839 \pm 0.061$ | $0.953 \pm 0.010$ |
| C-TableGAN | Corr | $0.849 \pm 0.025$ | $\mathbf{0.956} \pm 0.004$ | $0.874 \pm 0.011$ | $0.952 \pm 0.010$ | $\mathbf{0.818} \pm 0.012$ | $0.933 \pm 0.011$ |
| | KDE | $\mathbf{0.828} \pm 0.031$ | $0.962 \pm 0.007$ | $\mathbf{0.869} \pm 0.014$ | $0.948 \pm 0.018$ | $0.830 \pm 0.015$ | $0.933 \pm 0.011$ |
| CTGAN | - | $0.850 \pm 0.027$ | $0.990 \pm 0.002$ | $0.848 \pm 0.016$ | $0.914 \pm 0.024$ | $0.926 \pm 0.011$ | $\mathbf{0.901} \pm 0.018$ |
| | Rnd | $0.834 \pm 0.026$ | $0.990 \pm 0.001$ | $0.863 \pm 0.022$ | $0.910 \pm 0.013$ | $0.859 \pm 0.022$ | $0.909 \pm 0.016$ |
| C-CTGAN | Corr | $\mathbf{0.820} \pm 0.025$ | $0.987 \pm 0.003$ | $\mathbf{0.845} \pm 0.025$ | $0.903 \pm 0.012$ | $0.861 \pm 0.021$ | $0.905 \pm 0.019$ |
| | KDE | $0.838 \pm 0.033$ | $\mathbf{0.986} \pm 0.002$ | $0.848 \pm 0.017$ | $\mathbf{0.897} \pm 0.023$ | $\mathbf{0.857} \pm 0.022$ | $0.902 \pm 0.014$ |
| TVAE | - | $\mathbf{0.813} \pm 0.037$ | $\mathbf{0.926} \pm 0.001$ | $0.842 \pm 0.019$ | $0.914 \pm 0.011$ | $\mathbf{0.843} \pm 0.027$ | $0.877 \pm 0.013$ |
| | Rnd | $0.826 \pm 0.037$ | $0.965 \pm 0.001$ | $\mathbf{0.796} \pm 0.030$ | $\mathbf{0.905} \pm 0.022$ | $0.874 \pm 0.015$ | $0.874 \pm 0.018$ |
| C-TVAE | Corr | $0.815 \pm 0.037$ | $0.959 \pm 0.003$ | $0.808 \pm 0.030$ | $\mathbf{0.905} \pm 0.014$ | $0.855 \pm 0.020$ | $\mathbf{0.863} \pm 0.008$ |
| | KDE | $0.814 \pm 0.030$ | $0.961 \pm 0.002$ | $0.799 \pm 0.020$ | $0.907 \pm 0.021$ | $0.879 \pm 0.008$ | $0.890 \pm 0.019$ |
| GOGGLE | - | $0.892 \pm 0.019$ | $0.987 \pm 0.018$ | $\mathbf{0.890} \pm 0.017$ | $\mathbf{0.924} \pm 0.006$ | $0.912 \pm 0.010$ | $\mathbf{0.949} \pm 0.023$ |
| | Rnd | $\mathbf{0.890} \pm 0.044$ | $\mathbf{0.965} \pm 0.006$ | $0.904 \pm 0.011$ | $0.939 \pm 0.008$ | $\mathbf{0.866} \pm 0.020$ | $0.967 \pm 0.009$ |
| C-GOGGLE | Corr | $0.898 \pm 0.021$ | $0.972 \pm 0.005$ | $0.910 \pm 0.017$ | $0.939 \pm 0.010$ | $0.884 \pm 0.009$ | $0.968 \pm 0.015$ |
| | KDE | $0.903 \pm 0.019$ | $0.975 \pm 0.013$ | $0.911 \pm 0.017$ | $0.938 \pm 0.008$ | $0.887 \pm 0.014$ | $0.957 \pm 0.020$ |

Table 22: Weighted F1-score detection results with their corresponding stddevs for all datasets.

| | | URL | WiDS | LCLD | Heloc | FSP | News |
|---|---|---|---|---|---|---|---|
| WGAN | - | $0.856_{\pm0.008}$ | $0.976_{\pm0.002}$ | $0.999_{\pm0.001}$ | $0.964_{\pm0.004}$ | $0.910_{\pm0.027}$ | $0.954_{\pm0.020}$ |
| | Rnd | $\mathbf{0.851}_{\pm0.017}$ | $0.996_{\pm0.003}$ | $\mathbf{0.912}_{\pm0.013}$ | $0.970_{\pm0.024}$ | $0.900_{\pm0.072}$ | $0.969_{\pm0.016}$ |
| C-WGAN | Corr | $0.861_{\pm0.016}$ | $0.975_{\pm0.006}$ | $0.917_{\pm0.007}$ | $0.964_{\pm0.007}$ | $\mathbf{0.845}_{\pm0.022}$ | $0.929_{\pm0.022}$ |
| | KDE | $0.871_{\pm0.010}$ | $\mathbf{0.975}_{\pm0.002}$ | $0.917_{\pm0.007}$ | $\mathbf{0.957}_{\pm0.019}$ | $\mathbf{0.845}_{\pm0.022}$ | $\mathbf{0.925}_{\pm0.013}$ |
| TableGAN | - | $\mathbf{0.822}_{\pm0.007}$ | $0.964_{\pm0.006}$ | $0.895_{\pm0.022}$ | $\mathbf{0.925}_{\pm0.011}$ | $0.906_{\pm0.028}$ | $\mathbf{0.929}_{\pm0.013}$ |
| | Rnd | $0.830_{\pm0.011}$ | $0.984_{\pm0.002}$ | $\mathbf{0.859}_{\pm0.015}$ | $0.963_{\pm0.021}$ | $0.834_{\pm0.054}$ | $0.953_{\pm0.011}$ |
| C-TableGAN | Corr | $0.835_{\pm0.011}$ | $\mathbf{0.957}_{\pm0.003}$ | $0.866_{\pm0.022}$ | $0.952_{\pm0.010}$ | $\mathbf{0.813}_{\pm0.020}$ | $0.936_{\pm0.010}$ |
| | KDE | $0.827_{\pm0.006}$ | $0.963_{\pm0.007}$ | $0.871_{\pm0.013}$ | $0.947_{\pm0.019}$ | $0.817_{\pm0.021}$ | $0.934_{\pm0.011}$ |
| CTGAN | - | $0.862_{\pm0.014}$ | $0.990_{\pm0.002}$ | $0.850_{\pm0.019}$ | $0.915_{\pm0.023}$ | $0.927_{\pm0.016}$ | $0.896_{\pm0.023}$ |
| | Rnd | $0.843_{\pm0.017}$ | $0.990_{\pm0.001}$ | $0.867_{\pm0.015}$ | $0.911_{\pm0.013}$ | $\mathbf{0.861}_{\pm0.022}$ | $0.904_{\pm0.018}$ |
| C-CTGAN | Corr | $\mathbf{0.839}_{\pm0.015}$ | $0.988_{\pm0.002}$ | $0.846_{\pm0.024}$ | $0.904_{\pm0.013}$ | $0.865_{\pm0.028}$ | $0.901_{\pm0.023}$ |
| | KDE | $0.849_{\pm0.020}$ | $\mathbf{0.986}_{\pm0.002}$ | $\mathbf{0.837}_{\pm0.014}$ | $\mathbf{0.897}_{\pm0.023}$ | $0.868_{\pm0.009}$ | $\mathbf{0.894}_{\pm0.018}$ |
| TVAE | - | $0.831_{\pm0.013}$ | $\mathbf{0.927}_{\pm0.001}$ | $0.832_{\pm0.010}$ | $0.916_{\pm0.010}$ | $\mathbf{0.847}_{\pm0.006}$ | $0.855_{\pm0.019}$ |
| | Rnd | $\mathbf{0.829}_{\pm0.014}$ | $0.966_{\pm0.001}$ | $\mathbf{0.795}_{\pm0.025}$ | $\mathbf{0.908}_{\pm0.020}$ | $0.868_{\pm0.018}$ | $0.856_{\pm0.012}$ |
| C-TVAE | Corr | $0.832_{\pm0.011}$ | $0.960_{\pm0.003}$ | $\mathbf{0.795}_{\pm0.014}$ | $\mathbf{0.908}_{\pm0.013}$ | $0.857_{\pm0.011}$ | $\mathbf{0.854}_{\pm0.010}$ |
| | KDE | $0.829_{\pm0.014}$ | $0.962_{\pm0.002}$ | $0.797_{\pm0.019}$ | $0.909_{\pm0.020}$ | $0.873_{\pm0.014}$ | $0.882_{\pm0.024}$ |
| GOGGLE | - | $\mathbf{0.880}_{\pm0.012}$ | $0.987_{\pm0.018}$ | $\mathbf{0.892}_{\pm0.014}$ | $\mathbf{0.926}_{\pm0.005}$ | $0.916_{\pm0.011}$ | $\mathbf{0.955}_{\pm0.019}$ |
| | Rnd | $0.891_{\pm0.030}$ | $\mathbf{0.965}_{\pm0.005}$ | $0.905_{\pm0.010}$ | $0.940_{\pm0.008}$ | $0.823_{\pm0.030}$ | $0.970_{\pm0.008}$ |
| C-GOGGLE | Corr | $0.898_{\pm0.014}$ | $0.971_{\pm0.005}$ | $0.912_{\pm0.017}$ | $0.939_{\pm0.009}$ | $\mathbf{0.817}_{\pm0.004}$ | $0.972_{\pm0.012}$ |
| | KDE | $0.902_{\pm0.017}$ | $0.975_{\pm0.013}$ | $0.913_{\pm0.018}$ | $0.939_{\pm0.008}$ | $0.865_{\pm0.033}$ | $0.963_{\pm0.015}$ |

Table 23: Area under the ROC curve detection results with their corresponding stddevs for all datasets.

| | | URL | WiDS | LCLD | Heloc | FSP | News |
|---|---|---|---|---|---|---|---|
| WGAN | - | $\mathbf{0.872}_{\pm0.008}$ | $\mathbf{0.989}_{\pm0.002}$ | $1.000_{\pm0.000}$ | $0.983_{\pm0.004}$ | $0.916_{\pm0.016}$ | $0.964_{\pm0.021}$ |
| | Rnd | $0.877_{\pm0.017}$ | $0.999_{\pm0.001}$ | $\mathbf{0.941}_{\pm0.009}$ | $0.983_{\pm0.018}$ | $0.918_{\pm0.060}$ | $0.981_{\pm0.010}$ |
| C-WGAN | Corr | $0.879_{\pm0.009}$ | $\mathbf{0.989}_{\pm0.002}$ | $0.946_{\pm0.006}$ | $0.982_{\pm0.007}$ | $\mathbf{0.874}_{\pm0.020}$ | $0.945_{\pm0.020}$ |
| | KDE | $0.883_{\pm0.009}$ | $\mathbf{0.989}_{\pm0.001}$ | $0.946_{\pm0.006}$ | $\mathbf{0.975}_{\pm0.018}$ | $\mathbf{0.874}_{\pm0.020}$ | $\mathbf{0.941}_{\pm0.011}$ |
| TableGAN | - | $0.850_{\pm0.007}$ | $0.980_{\pm0.004}$ | $0.926_{\pm0.020}$ | $\mathbf{0.953}_{\pm0.009}$ | $0.907_{\pm0.022}$ | $\mathbf{0.940}_{\pm0.011}$ |
| | Rnd | $0.851_{\pm0.006}$ | $0.995_{\pm0.001}$ | $\mathbf{0.900}_{\pm0.012}$ | $0.978_{\pm0.016}$ | $0.866_{\pm0.048}$ | $0.964_{\pm0.012}$ |
| C-TableGAN | Corr | $0.856_{\pm0.004}$ | $\mathbf{0.974}_{\pm0.002}$ | $0.904_{\pm0.019}$ | $0.970_{\pm0.008}$ | $\mathbf{0.845}_{\pm0.005}$ | $0.950_{\pm0.010}$ |
| | KDE | $\mathbf{0.849}_{\pm0.005}$ | $0.978_{\pm0.005}$ | $0.909_{\pm0.012}$ | $0.965_{\pm0.017}$ | $0.849_{\pm0.012}$ | $0.947_{\pm0.010}$ |
| CTGAN | - | $0.879_{\pm0.008}$ | $0.996_{\pm0.001}$ | $\mathbf{0.896}_{\pm0.011}$ | $0.953_{\pm0.016}$ | $0.932_{\pm0.007}$ | $\mathbf{0.912}_{\pm0.021}$ |
| | Rnd | $0.865_{\pm0.013}$ | $0.997_{\pm0.000}$ | $0.908_{\pm0.011}$ | $0.950_{\pm0.008}$ | $0.896_{\pm0.016}$ | $0.925_{\pm0.021}$ |
| C-CTGAN | Corr | $\mathbf{0.864}_{\pm0.009}$ | $\mathbf{0.995}_{\pm0.001}$ | $0.900_{\pm0.020}$ | $0.946_{\pm0.004}$ | $0.894_{\pm0.025}$ | $0.922_{\pm0.019}$ |
| | KDE | $0.867_{\pm0.015}$ | $\mathbf{0.995}_{\pm0.001}$ | $0.897_{\pm0.012}$ | $\mathbf{0.943}_{\pm0.018}$ | $\mathbf{0.893}_{\pm0.007}$ | $0.919_{\pm0.022}$ |
| TVAE | - | $0.854_{\pm0.007}$ | $\mathbf{0.935}_{\pm0.002}$ | $0.861_{\pm0.009}$ | $0.947_{\pm0.005}$ | $\mathbf{0.872}_{\pm0.008}$ | $0.881_{\pm0.019}$ |
| | Rnd | $0.854_{\pm0.008}$ | $0.982_{\pm0.001}$ | $0.844_{\pm0.013}$ | $\mathbf{0.942}_{\pm0.015}$ | $0.904_{\pm0.015}$ | $0.883_{\pm0.012}$ |
| C-TVAE | Corr | $0.853_{\pm0.009}$ | $0.977_{\pm0.001}$ | $\mathbf{0.840}_{\pm0.011}$ | $0.943_{\pm0.011}$ | $0.896_{\pm0.008}$ | $\mathbf{0.880}_{\pm0.014}$ |
| | KDE | $\mathbf{0.849}_{\pm0.013}$ | $0.979_{\pm0.001}$ | $0.843_{\pm0.012}$ | $0.944_{\pm0.014}$ | $0.904_{\pm0.013}$ | $0.909_{\pm0.030}$ |
| GOGGLE | - | $\mathbf{0.891}_{\pm0.009}$ | $0.993_{\pm0.013}$ | $\mathbf{0.928}_{\pm0.010}$ | $\mathbf{0.949}_{\pm0.003}$ | $0.920_{\pm0.006}$ | $0.973_{\pm0.013}$ |
| | Rnd | $0.898_{\pm0.028}$ | $\mathbf{0.979}_{\pm0.007}$ | $0.938_{\pm0.010}$ | $0.958_{\pm0.008}$ | $0.871_{\pm0.009}$ | $0.977_{\pm0.007}$ |
| C-GOGGLE | Corr | $0.906_{\pm0.020}$ | $0.984_{\pm0.005}$ | $0.943_{\pm0.013}$ | $0.960_{\pm0.007}$ | $\mathbf{0.865}_{\pm0.011}$ | $0.980_{\pm0.012}$ |
| | KDE | $0.907_{\pm0.018}$ | $0.985_{\pm0.009}$ | $0.944_{\pm0.013}$ | $0.960_{\pm0.007}$ | $0.880_{\pm0.014}$ | $\mathbf{0.973}_{\pm0.009}$ |

Table 24: Binary (macro) F1-score utility results with their corresponding stddevs for binary (multiclass) classification datasets when postprocessing the unconstrained predictions.

| | | URL | WiDS | LCLD | Heloc | FSP |
|---|---|---|---|---|---|---|
| WGAN | - | $0.756 \pm 0.030$ | $0.329 \pm 0.059$ | $\mathbf{0.239} \pm 0.026$ | $0.634 \pm 0.099$ | $0.357 \pm 0.020$ |
| | Rnd | $\mathbf{0.767} \pm 0.034$ | $0.318 \pm 0.053$ | $0.201 \pm 0.030$ | $\mathbf{0.641} \pm 0.113$ | $0.355 \pm 0.020$ |
| P-WGAN | Corr | $0.766 \pm 0.036$ | $0.332 \pm 0.050$ | $0.214 \pm 0.021$ | $\mathbf{0.641} \pm 0.115$ | $\mathbf{0.358} \pm 0.024$ |
| | KDE | $\mathbf{0.767} \pm 0.035$ | $\mathbf{0.334} \pm 0.059$ | $0.214 \pm 0.021$ | $\mathbf{0.641} \pm 0.112$ | $\mathbf{0.358} \pm 0.024$ |
| TableGAN | - | $0.562 \pm 0.051$ | $0.171 \pm 0.037$ | $0.123 \pm 0.041$ | $0.593 \pm 0.058$ | $\mathbf{0.199} \pm 0.044$ |
| | Rnd | $\mathbf{0.565} \pm 0.048$ | $0.173 \pm 0.049$ | $0.115 \pm 0.028$ | $\mathbf{0.594} \pm 0.058$ | $0.195 \pm 0.046$ |
| P-TableGAN | Corr | $0.553 \pm 0.050$ | $\mathbf{0.174} \pm 0.045$ | $\mathbf{0.125} \pm 0.032$ | $\mathbf{0.594} \pm 0.060$ | $0.194 \pm 0.037$ |
| | KDE | $0.556 \pm 0.060$ | $0.171 \pm 0.042$ | $0.119 \pm 0.025$ | $0.592 \pm 0.060$ | $0.193 \pm 0.042$ |
| CTGAN | - | $0.822 \pm 0.017$ | $0.362 \pm 0.033$ | $0.247 \pm 0.087$ | $0.736 \pm 0.035$ | $\mathbf{0.374} \pm 0.034$ |
| | Rnd | $0.819 \pm 0.014$ | $0.364 \pm 0.032$ | $\mathbf{0.255} \pm 0.089$ | $0.735 \pm 0.038$ | $0.365 \pm 0.021$ |
| P-CTGAN | Corr | $0.814 \pm 0.008$ | $\mathbf{0.365} \pm 0.038$ | $0.246 \pm 0.087$ | $\mathbf{0.743} \pm 0.032$ | $0.372 \pm 0.033$ |
| | KDE | $\mathbf{0.823} \pm 0.017$ | $0.357 \pm 0.040$ | $0.248 \pm 0.077$ | $0.736 \pm 0.031$ | $0.371 \pm 0.035$ |
| TVAE | - | $0.810 \pm 0.008$ | $0.282 \pm 0.029$ | $\mathbf{0.185} \pm 0.021$ | $0.735 \pm 0.010$ | $\mathbf{0.473} \pm 0.016$ |
| | Rnd | $0.810 \pm 0.011$ | $0.266 \pm 0.020$ | $0.172 \pm 0.011$ | $\mathbf{0.739} \pm 0.004$ | $0.464 \pm 0.019$ |
| P-TVAE | Corr | $\mathbf{0.815} \pm 0.009$ | $0.283 \pm 0.026$ | $0.176 \pm 0.028$ | $0.730 \pm 0.006$ | $0.462 \pm 0.018$ |
| | KDE | $\mathbf{0.815} \pm 0.009$ | $\mathbf{0.285} \pm 0.034$ | $0.171 \pm 0.030$ | $0.735 \pm 0.007$ | $0.463 \pm 0.021$ |
| GOGGLE | - | $0.622 \pm 0.094$ | $0.189 \pm 0.038$ | $0.163 \pm 0.119$ | $0.596 \pm 0.072$ | $0.152 \pm 0.003$ |
| | Rnd | $0.626 \pm 0.098$ | $\mathbf{0.193} \pm 0.042$ | $0.164 \pm 0.121$ | $0.600 \pm 0.060$ | $0.151 \pm 0.007$ |
| P-GOGGLE | Corr | $0.623 \pm 0.089$ | $0.191 \pm 0.038$ | $\mathbf{0.169} \pm 0.126$ | $\mathbf{0.601} \pm 0.063$ | $\mathbf{0.155} \pm 0.010$ |
| | KDE | $\mathbf{0.626} \pm 0.096$ | $0.189 \pm 0.039$ | $\mathbf{0.169} \pm 0.126$ | $0.597 \pm 0.066$ | $\mathbf{0.155} \pm 0.011$ |

Table 25: Weighted F1-score utility results with their corresponding stddevs for binary and multiclass classification datasets when post-processing the unconstrained predictions.

| | | URL | WiDS | LCLD | Heloc | FSP |
|---|---|---|---|---|---|---|
| WGAN | - | $0.764 \pm 0.018$ | $0.381 \pm 0.057$ | $\mathbf{0.359} \pm 0.019$ | $0.599 \pm 0.050$ | $0.339 \pm 0.021$ |
| | Rnd | $0.768 \pm 0.028$ | $0.372 \pm 0.052$ | $0.342 \pm 0.019$ | $0.600 \pm 0.057$ | $0.337 \pm 0.022$ |
| P-WGAN | Corr | $\mathbf{0.769} \pm 0.027$ | $0.384 \pm 0.048$ | $0.352 \pm 0.014$ | $0.598 \pm 0.054$ | $\mathbf{0.341} \pm 0.026$ |
| | KDE | $0.762 \pm 0.028$ | $\mathbf{0.386} \pm 0.057$ | $0.352 \pm 0.014$ | $\mathbf{0.603} \pm 0.059$ | $\mathbf{0.341} \pm 0.026$ |
| TableGAN | - | $0.659 \pm 0.035$ | $0.240 \pm 0.034$ | $\mathbf{0.286} \pm 0.029$ | $\mathbf{0.615} \pm 0.030$ | $\mathbf{0.199} \pm 0.043$ |
| | Rnd | $\mathbf{0.660} \pm 0.035$ | $\mathbf{0.243} \pm 0.045$ | $0.281 \pm 0.022$ | $0.614 \pm 0.028$ | $0.195 \pm 0.046$ |
| P-TableGAN | Corr | $0.657 \pm 0.035$ | $\mathbf{0.243} \pm 0.041$ | $0.285 \pm 0.023$ | $0.614 \pm 0.030$ | $0.195 \pm 0.036$ |
| | KDE | $0.658 \pm 0.040$ | $0.240 \pm 0.039$ | $0.285 \pm 0.018$ | $0.613 \pm 0.029$ | $0.193 \pm 0.042$ |
| CTGAN | - | $0.799 \pm 0.033$ | $0.405 \pm 0.034$ | $0.379 \pm 0.061$ | $0.675 \pm 0.015$ | $\mathbf{0.372} \pm 0.034$ |
| | Rnd | $0.794 \pm 0.031$ | $\mathbf{0.407} \pm 0.032$ | $\mathbf{0.383} \pm 0.070$ | $0.671 \pm 0.015$ | $0.363 \pm 0.019$ |
| P-CTGAN | Corr | $0.801 \pm 0.008$ | $\mathbf{0.407} \pm 0.040$ | $0.379 \pm 0.068$ | $\mathbf{0.678} \pm 0.010$ | $0.370 \pm 0.033$ |
| | KDE | $\mathbf{0.802} \pm 0.032$ | $0.399 \pm 0.042$ | $0.380 \pm 0.058$ | $0.671 \pm 0.010$ | $0.368 \pm 0.033$ |
| TVAE | - | $0.802 \pm 0.012$ | $0.342 \pm 0.027$ | $\mathbf{0.330} \pm 0.016$ | $\mathbf{0.696} \pm 0.006$ | $\mathbf{0.463} \pm 0.017$ |
| | Rnd | $0.802 \pm 0.015$ | $0.327 \pm 0.018$ | $0.324 \pm 0.009$ | $\mathbf{0.696} \pm 0.004$ | $0.454 \pm 0.020$ |
| P-TVAE | Corr | $\mathbf{0.806} \pm 0.008$ | $0.342 \pm 0.024$ | $0.325 \pm 0.018$ | $0.691 \pm 0.004$ | $0.452 \pm 0.019$ |
| | KDE | $\mathbf{0.806} \pm 0.008$ | $\mathbf{0.344} \pm 0.031$ | $0.322 \pm 0.020$ | $0.694 \pm 0.004$ | $0.453 \pm 0.022$ |
| GOGGLE | - | $0.648 \pm 0.074$ | $0.198 \pm 0.067$ | $0.315 \pm 0.086$ | $\mathbf{0.566} \pm 0.050$ | $0.139 \pm 0.002$ |
| | Rnd | $0.645 \pm 0.071$ | $\mathbf{0.202} \pm 0.070$ | $0.315 \pm 0.086$ | $\mathbf{0.566} \pm 0.047$ | $0.139 \pm 0.006$ |
| P-GOGGLE | Corr | $0.645 \pm 0.070$ | $0.201 \pm 0.067$ | $\mathbf{0.318} \pm 0.090$ | $\mathbf{0.566} \pm 0.047$ | $\mathbf{0.142} \pm 0.009$ |
| | KDE | $\mathbf{0.650} \pm 0.072$ | $0.197 \pm 0.069$ | $\mathbf{0.318} \pm 0.090$ | $0.565 \pm 0.048$ | $0.141 \pm 0.011$ |

Table 26: Area under the ROC curve utility results with their corresponding stddevs for binary and multiclass classification datasets when post-processing the unconstrained predictions.

|  |  | URL | WiDS | LCLD | Heloc | FSP |
|---|---|---|---|---|---|---|
| WGAN | - | $0.839_{\pm0.016}$ | $0.775_{\pm0.038}$ | $\mathbf{0.618}_{\pm0.011}$ | $0.677_{\pm0.033}$ | $\mathbf{0.742}_{\pm0.007}$ |
|  | Rnd | $\mathbf{0.840}_{\pm0.019}$ | $0.777_{\pm0.037}$ | $\mathbf{0.618}_{\pm0.010}$ | $0.682_{\pm0.042}$ | $0.736_{\pm0.011}$ |
| P-WGAN | Corr | $0.839_{\pm0.020}$ | $0.782_{\pm0.035}$ | $0.617_{\pm0.008}$ | $\mathbf{0.683}_{\pm0.038}$ | $0.740_{\pm0.010}$ |
|  | KDE | $0.834_{\pm0.021}$ | $\mathbf{0.783}_{\pm0.039}$ | $0.617_{\pm0.008}$ | $0.682_{\pm0.041}$ | $0.740_{\pm0.010}$ |
| TableGAN | - | $0.843_{\pm0.020}$ | $0.740_{\pm0.021}$ | $\mathbf{0.587}_{\pm0.027}$ | $\mathbf{0.707}_{\pm0.007}$ | $0.642_{\pm0.026}$ |
|  | Rnd | $0.848_{\pm0.019}$ | $\mathbf{0.749}_{\pm0.019}$ | $0.585_{\pm0.027}$ | $0.703_{\pm0.008}$ | $0.641_{\pm0.030}$ |
| P-TableGAN | Corr | $0.846_{\pm0.020}$ | $0.731_{\pm0.022}$ | $\mathbf{0.587}_{\pm0.028}$ | $\mathbf{0.707}_{\pm0.011}$ | $\mathbf{0.645}_{\pm0.031}$ |
|  | KDE | $\mathbf{0.855}_{\pm0.016}$ | $0.736_{\pm0.016}$ | $\mathbf{0.587}_{\pm0.027}$ | $\mathbf{0.707}_{\pm0.011}$ | $0.644_{\pm0.035}$ |
| CTGAN | - | $0.859_{\pm0.040}$ | $0.835_{\pm0.012}$ | $0.651_{\pm0.020}$ | $0.744_{\pm0.009}$ | $\mathbf{0.760}_{\pm0.014}$ |
|  | Rnd | $0.859_{\pm0.038}$ | $\mathbf{0.837}_{\pm0.012}$ | $\mathbf{0.652}_{\pm0.024}$ | $0.743_{\pm0.008}$ | $0.753_{\pm0.008}$ |
| P-CTGAN | Corr | $\mathbf{0.866}_{\pm0.007}$ | $0.835_{\pm0.012}$ | $0.650_{\pm0.023}$ | $\mathbf{0.745}_{\pm0.008}$ | $0.754_{\pm0.011}$ |
|  | KDE | $0.863_{\pm0.035}$ | $0.836_{\pm0.012}$ | $0.651_{\pm0.018}$ | $0.743_{\pm0.009}$ | $0.753_{\pm0.009}$ |
| TVAE | - | $0.863_{\pm0.011}$ | $\mathbf{0.800}_{\pm0.016}$ | $\mathbf{0.631}_{\pm0.004}$ | $\mathbf{0.752}_{\pm0.003}$ | $\mathbf{0.789}_{\pm0.007}$ |
|  | Rnd | $0.866_{\pm0.013}$ | $0.785_{\pm0.017}$ | $0.630_{\pm0.007}$ | $0.751_{\pm0.003}$ | $0.787_{\pm0.011}$ |
| P-TVAE | Corr | $\mathbf{0.870}_{\pm0.007}$ | $0.787_{\pm0.009}$ | $0.629_{\pm0.004}$ | $\mathbf{0.752}_{\pm0.004}$ | $0.779_{\pm0.014}$ |
|  | KDE | $\mathbf{0.870}_{\pm0.007}$ | $0.797_{\pm0.016}$ | $0.630_{\pm0.002}$ | $0.750_{\pm0.001}$ | $0.788_{\pm0.010}$ |
| GOGGLE | - | $\mathbf{0.742}_{\pm0.071}$ | $0.656_{\pm0.049}$ | $0.543_{\pm0.039}$ | $0.600_{\pm0.056}$ | $0.577_{\pm0.010}$ |
|  | Rnd | $0.738_{\pm0.067}$ | $0.665_{\pm0.043}$ | $\mathbf{0.548}_{\pm0.035}$ | $0.602_{\pm0.056}$ | $0.575_{\pm0.008}$ |
| P-GOGGLE | Corr | $0.740_{\pm0.063}$ | $\mathbf{0.667}_{\pm0.044}$ | $0.542_{\pm0.036}$ | $0.601_{\pm0.058}$ | $0.576_{\pm0.010}$ |
|  | KDE | $0.741_{\pm0.067}$ | $\mathbf{0.667}_{\pm0.060}$ | $0.542_{\pm0.036}$ | $\mathbf{0.603}_{\pm0.058}$ | $\mathbf{0.578}_{\pm0.014}$ |

Table 27: Utility performance comparison between DGM and P-DGM models trained on News.

|  |  | XV | MAE |
|---|---|---|---|
| WGAN | - | $-0.006_{\pm0.003}$ | $3133.6_{\pm242.1}$ |
|  | Rnd | $\mathbf{-0.005}_{\pm0.002}$ | $3151.3_{\pm219.8}$ |
| P-WGAN | Corr | $-0.012_{\pm0.011}$ | $\mathbf{3109.2}_{\pm220.4}$ |
|  | KDE | $-0.006_{\pm0.003}$ | $3159.7_{\pm210.7}$ |
| TableGAN | - | $\mathbf{-0.001}_{\pm0.001}$ | $2992.2_{\pm35.8}$ |
|  | Rnd | $\mathbf{-0.001}_{\pm0.001}$ | $\mathbf{2981.9}_{\pm32.3}$ |
| P-TableGAN | Corr | $\mathbf{-0.001}_{\pm0.001}$ | $3015.3_{\pm43.9}$ |
|  | KDE | $\mathbf{-0.001}_{\pm0.001}$ | $3022.7_{\pm23.6}$ |
| CTGAN | - | $\mathbf{-0.002}_{\pm0.001}$ | $3043.8_{\pm37.8}$ |
|  | Rnd | $-0.003_{\pm0.004}$ | $3013.4_{\pm36.2}$ |
| P-CTGAN | Corr | $-0.006_{\pm0.002}$ | $\mathbf{2999.3}_{\pm27.3}$ |
|  | KDE | $-0.003_{\pm0.003}$ | $3004.8_{\pm47.4}$ |
| TVAE | - | $\mathbf{-0.001}_{\pm0.000}$ | $3021.3_{\pm10.0}$ |
|  | Rnd | $\mathbf{-0.001}_{\pm0.001}$ | $3040.5_{\pm60.4}$ |
| P-TVAE | Corr | $-0.002_{\pm0.001}$ | $\mathbf{3003.8}_{\pm22.7}$ |
|  | KDE | $\mathbf{-0.001}_{\pm0.001}$ | $3032.2_{\pm39.3}$ |
| GOGGLE | - | $\mathbf{-0.004}_{\pm0.003}$ | $3054.5_{\pm44.7}$ |
|  | Rnd | $-0.006_{\pm0.005}$ | $3024.1_{\pm37.2}$ |
| P-GOGGLE | Corr | $-0.005_{\pm0.002}$ | $3077.9_{\pm54.1}$ |
|  | KDE | $\mathbf{-0.004}_{\pm0.003}$ | $\mathbf{3012.3}_{\pm36.6}$ |

Table 28: Binary F1-score detection results with their corresponding stddevs for all datasets when post-processing the unconstrained predictions.

| | | URL | WiDS | LCLD | Heloc | FSP | News |
|---|---|---|---|---|---|---|---|
| WGAN | - | 0.865±0.035 | **0.975**±0.002 | 0.999±0.001 | **0.964**±0.004 | 0.914±0.018 | **0.954**±0.022 |
| | Rnd | **0.856**±0.037 | 0.979±0.003 | **0.916**±0.008 | **0.964**±0.004 | **0.910**±0.015 | **0.954**±0.022 |
| P-WGAN | Corr | 0.857±0.033 | 0.977±0.002 | 0.921±0.004 | 0.968±0.005 | 0.911±0.018 | **0.954**±0.026 |
| | KDE | 0.862±0.037 | 0.978±0.003 | 0.921±0.004 | 0.967±0.004 | 0.911±0.018 | 0.957±0.019 |
| TableGAN | - | **0.831**±0.029 | 0.963±0.006 | **0.895**±0.024 | 0.923±0.011 | 0.909±0.021 | 0.927±0.009 |
| | Rnd | 0.843±0.034 | **0.962**±0.006 | 0.896±0.026 | 0.923±0.010 | 0.858±0.031 | 0.932±0.010 |
| P-TableGAN | Corr | 0.833±0.035 | 0.963±0.005 | 0.900±0.024 | 0.922±0.011 | 0.860±0.029 | **0.920**±0.009 |
| | KDE | 0.835±0.041 | **0.962**±0.006 | 0.901±0.019 | **0.921**±0.011 | **0.857**±0.029 | 0.928±0.004 |
| CTGAN | - | 0.850±0.027 | 0.990±0.002 | 0.848±0.016 | 0.914±0.024 | 0.926±0.011 | 0.901±0.018 |
| | Rnd | **0.845**±0.028 | 0.990±0.002 | **0.839**±0.015 | 0.916±0.028 | 0.902±0.029 | **0.898**±0.012 |
| P-CTGAN | Corr | 0.855±0.014 | **0.989**±0.003 | 0.845±0.019 | **0.913**±0.028 | **0.895**±0.030 | **0.898**±0.006 |
| | KDE | 0.850±0.023 | 0.991±0.002 | 0.841±0.019 | **0.913**±0.027 | **0.895**±0.032 | 0.902±0.010 |
| TVAE | - | 0.813±0.037 | **0.926**±0.001 | 0.842±0.019 | 0.914±0.011 | **0.843**±0.027 | 0.877±0.013 |
| | Rnd | **0.806**±0.039 | 0.962±0.002 | 0.835±0.018 | 0.914±0.007 | 0.881±0.007 | 0.873±0.012 |
| P-TVAE | Corr | **0.806**±0.044 | 0.962±0.001 | **0.829**±0.022 | 0.912±0.011 | 0.881±0.011 | **0.866**±0.014 |
| | KDE | **0.806**±0.044 | 0.964±0.001 | 0.837±0.010 | **0.909**±0.011 | 0.874±0.012 | 0.868±0.012 |
| GOGGLE | - | 0.892±0.019 | **0.987**±0.018 | **0.890**±0.017 | **0.924**±0.006 | 0.912±0.010 | **0.949**±0.023 |
| | Rnd | **0.884**±0.025 | 0.988±0.017 | 0.892±0.012 | 0.927±0.006 | **0.907**±0.005 | 0.952±0.021 |
| P-GOGGLE | Corr | 0.897±0.010 | 0.988±0.016 | 0.902±0.005 | 0.926±0.006 | 0.909±0.006 | 0.951±0.021 |
| | KDE | 0.898±0.015 | **0.987**±0.017 | 0.902±0.005 | 0.928±0.007 | **0.907**±0.009 | 0.951±0.020 |

Table 29: Weighted F1-score detection results with their corresponding stddevs for all datasets when post-processing the unconstrained predictions.

| | | URL | WiDS | LCLD | Heloc | FSP | News |
|---|---|---|---|---|---|---|---|
| WGAN | - | 0.856±0.008 | **0.976**±0.002 | 0.999±0.001 | **0.964**±0.004 | 0.910±0.027 | **0.954**±0.020 |
| | Rnd | **0.852**±0.011 | 0.979±0.003 | **0.907**±0.010 | 0.965±0.004 | **0.907**±0.009 | 0.957±0.020 |
| P-WGAN | Corr | 0.853±0.008 | 0.977±0.002 | 0.914±0.002 | 0.968±0.004 | 0.912±0.018 | 0.955±0.025 |
| | KDE | 0.855±0.010 | 0.978±0.003 | 0.914±0.002 | 0.967±0.004 | 0.912±0.018 | 0.959±0.017 |
| TableGAN | - | **0.822**±0.007 | 0.964±0.006 | 0.895±0.022 | 0.925±0.011 | 0.906±0.028 | 0.929±0.013 |
| | Rnd | 0.823±0.005 | **0.963**±0.006 | **0.888**±0.031 | 0.924±0.011 | 0.873±0.018 | 0.934±0.012 |
| P-TableGAN | Corr | 0.826±0.006 | 0.964±0.005 | 0.898±0.025 | 0.924±0.011 | **0.868**±0.016 | **0.923**±0.009 |
| | KDE | 0.824±0.005 | **0.963**±0.006 | 0.896±0.021 | **0.923**±0.011 | 0.873±0.022 | 0.928±0.006 |
| CTGAN | - | 0.862±0.014 | 0.990±0.002 | 0.850±0.019 | 0.915±0.023 | 0.927±0.016 | 0.896±0.023 |
| | Rnd | **0.859**±0.017 | 0.990±0.002 | **0.840**±0.014 | 0.917±0.027 | 0.905±0.026 | **0.894**±0.006 |
| P-CTGAN | Corr | 0.860±0.012 | **0.989**±0.003 | 0.850±0.009 | 0.914±0.027 | **0.902**±0.017 | **0.894**±0.014 |
| | KDE | 0.861±0.017 | 0.991±0.002 | 0.847±0.006 | **0.913**±0.027 | 0.905±0.025 | 0.899±0.014 |
| TVAE | - | 0.831±0.013 | **0.927**±0.001 | 0.832±0.010 | 0.916±0.010 | **0.847**±0.006 | 0.855±0.019 |
| | Rnd | **0.828**±0.012 | 0.963±0.002 | 0.832±0.010 | 0.915±0.007 | 0.877±0.010 | 0.856±0.017 |
| P-TVAE | Corr | 0.829±0.012 | 0.963±0.001 | **0.825**±0.011 | 0.914±0.011 | 0.882±0.012 | 0.860±0.015 |
| | KDE | 0.829±0.012 | 0.964±0.001 | 0.831±0.014 | **0.910**±0.011 | 0.870±0.016 | **0.848**±0.014 |
| GOGGLE | - | 0.880±0.012 | **0.987**±0.018 | 0.892±0.014 | **0.926**±0.005 | 0.916±0.011 | **0.955**±0.019 |
| | Rnd | 0.878±0.018 | 0.988±0.017 | **0.889**±0.015 | 0.928±0.005 | 0.911±0.007 | 0.957±0.017 |
| P-GOGGLE | Corr | **0.875**±0.021 | 0.988±0.016 | 0.901±0.006 | 0.927±0.005 | 0.907±0.012 | 0.957±0.017 |
| | KDE | 0.883±0.011 | **0.987**±0.017 | 0.901±0.006 | 0.929±0.006 | **0.906**±0.011 | 0.956±0.017 |

Table 30: Area under the ROC curve detection results with their corresponding stddevs for all datasets when post-processing the unconstrained predictions.

| | | URL | WiDS | LCLD | Heloc | FSP | News |
|---|---|---|---|---|---|---|---|
| WGAN | - | $0.872\pm0.008$ | $\mathbf{0.989}\pm0.002$ | $1.000\pm0.000$ | $\mathbf{0.983}\pm0.004$ | $\mathbf{0.916}\pm0.016$ | $\mathbf{0.964}\pm0.021$ |
| | Rnd | $\mathbf{0.867}\pm0.003$ | $0.992\pm0.002$ | $\mathbf{0.936}\pm0.010$ | $0.985\pm0.002$ | $0.927\pm0.006$ | $0.966\pm0.020$ |
| P-WGAN | Corr | $0.869\pm0.004$ | $0.991\pm0.002$ | $0.942\pm0.001$ | $0.985\pm0.003$ | $0.931\pm0.013$ | $0.966\pm0.024$ |
| | KDE | $0.870\pm0.003$ | $0.991\pm0.002$ | $0.942\pm0.001$ | $0.985\pm0.002$ | $0.931\pm0.013$ | $0.969\pm0.017$ |
| TableGAN | - | $0.850\pm0.007$ | $0.980\pm0.004$ | $0.926\pm0.020$ | $0.953\pm0.009$ | $0.907\pm0.022$ | $0.940\pm0.011$ |
| | Rnd | $0.850\pm0.006$ | $0.980\pm0.004$ | $\mathbf{0.925}\pm0.024$ | $0.953\pm0.009$ | $0.894\pm0.016$ | $0.946\pm0.012$ |
| P-TableGAN | Corr | $\mathbf{0.848}\pm0.007$ | $0.980\pm0.003$ | $0.929\pm0.023$ | $0.953\pm0.008$ | $\mathbf{0.886}\pm0.018$ | $\mathbf{0.937}\pm0.011$ |
| | KDE | $0.852\pm0.009$ | $\mathbf{0.979}\pm0.003$ | $0.933\pm0.016$ | $\mathbf{0.952}\pm0.008$ | $0.890\pm0.022$ | $0.939\pm0.006$ |
| CTGAN | - | $0.879\pm0.008$ | $\mathbf{0.996}\pm0.001$ | $0.896\pm0.011$ | $0.953\pm0.016$ | $0.932\pm0.007$ | $\mathbf{0.912}\pm0.021$ |
| | Rnd | $\mathbf{0.876}\pm0.012$ | $0.997\pm0.001$ | $\mathbf{0.892}\pm0.008$ | $0.953\pm0.021$ | $0.925\pm0.019$ | $0.915\pm0.012$ |
| P-CTGAN | Corr | $0.878\pm0.004$ | $0.997\pm0.001$ | $0.896\pm0.005$ | $\mathbf{0.952}\pm0.018$ | $\mathbf{0.922}\pm0.013$ | $0.921\pm0.017$ |
| | KDE | $\mathbf{0.876}\pm0.013$ | $0.998\pm0.001$ | $0.893\pm0.004$ | $\mathbf{0.952}\pm0.020$ | $0.925\pm0.017$ | $0.919\pm0.021$ |
| TVAE | - | $0.854\pm0.007$ | $\mathbf{0.935}\pm0.002$ | $0.861\pm0.009$ | $0.947\pm0.005$ | $\mathbf{0.872}\pm0.008$ | $0.881\pm0.019$ |
| | Rnd | $\mathbf{0.849}\pm0.010$ | $0.978\pm0.002$ | $0.862\pm0.008$ | $0.948\pm0.004$ | $0.907\pm0.008$ | $0.887\pm0.021$ |
| P-TVAE | Corr | $0.850\pm0.009$ | $0.979\pm0.001$ | $\mathbf{0.858}\pm0.007$ | $0.946\pm0.007$ | $0.910\pm0.011$ | $0.884\pm0.014$ |
| | KDE | $0.850\pm0.009$ | $0.979\pm0.001$ | $0.861\pm0.005$ | $\mathbf{0.944}\pm0.006$ | $0.895\pm0.013$ | $\mathbf{0.874}\pm0.018$ |
| GOGGLE | - | $0.891\pm0.009$ | $\mathbf{0.993}\pm0.013$ | $0.928\pm0.010$ | $\mathbf{0.949}\pm0.003$ | $0.920\pm0.006$ | $\mathbf{0.973}\pm0.013$ |
| | Rnd | $\mathbf{0.889}\pm0.017$ | $0.994\pm0.013$ | $\mathbf{0.923}\pm0.008$ | $0.952\pm0.004$ | $0.924\pm0.011$ | $0.976\pm0.012$ |
| P-GOGGLE | Corr | $0.895\pm0.010$ | $0.994\pm0.012$ | $0.931\pm0.005$ | $0.952\pm0.003$ | $0.919\pm0.011$ | $0.977\pm0.011$ |
| | KDE | $0.894\pm0.017$ | $\mathbf{0.993}\pm0.012$ | $0.931\pm0.005$ | $0.952\pm0.003$ | $\mathbf{0.917}\pm0.010$ | $0.976\pm0.012$ |

## C.6 REAL DATA PERFORMANCE

To ensure that the comparisons between our C-DGM models and the baseline unconstrained DGMs are meaningful, we conducted a hyperparameter search as detailed earlier in Section B.5, allowing us to get close to (and sometimes even surpass) the real data utility performance. We report the latter in Table 31 using the same three metrics we used for measuring the synthetic data utility performance (i.e., F1-score, weighted F1-score, and Area Under the ROC Curve) following the same protocol as the one described in Appendix B.4. Comparing the results here with those for the synthetic data in Tables 17-19 we can see that, overall, C-WGAN and C-CTGAN models got utility scores similar to those obtained on the real data. For C-TableGAN and C-TVAE we notice several cases where our method helped in bringing the performance of the synthetic data closer to the real data. In particular, for LCLD we notice that the real data got an F1-score 4.8% higher than the unconstrained TableGAN, but our C-TableGAN model was able to match the real data performance (scoring slightly better than it, i.e., by 0.3%). It is also worth noticing that the gap between real and synthetic performance was reduced the most for the LCLD and WiDS datasets, both of which are highly unbalanced. For instance, all C-WGAN, WGAN, C-CTGAN, and CTGAN models yield a higher F1-score than the real LCLD data. We notice similar trends for the other metrics, weighted F1 and Area Under the ROC Curve. On the other hand, none of the DGM or C-DGM models get close to the real FSP data. One possible reason is the datasets' small size and multiclass nature, which makes it harder for the DGM models to capture patterns that lead to the correct different targets.

Table 31: Utility scores calculated on real data.

| | F1 | wF1 | AUC |
|---|---|---|---|
| URL | $0.884\pm0.007$ | $0.875\pm0.014$ | $0.903\pm0.009$ |
| WiDS | $0.383\pm0.021$ | $0.434\pm0.020$ | $0.832\pm0.009$ |
| LCLD | $0.171\pm0.030$ | $0.316\pm0.013$ | $0.645\pm0.007$ |
| Heloc | $0.772\pm0.003$ | $0.662\pm0.011$ | $0.707\pm0.008$ |
| FSP | $0.662\pm0.011$ | $0.659\pm0.009$ | $0.848\pm0.010$ |

