# OpenReview forum: "How Realistic Is Your Synthetic Data? Constraining Deep Generative Models for Tabular Data"
_ICLR.cc/2024/Conference — ICLR 2024 poster_

### Official Review · Reviewer_72Ff · 2023-10-31

**Soundness:** 3 good
**Presentation:** 3 good
**Contribution:** 3 good
**Rating:** 6
**Confidence:** 3

**Summary:**

The authors propose a method for generating tabular data which respects some domain-specific conditions, by introducing the so-called Constraint Layers (CL), which can enforce linear constraints on the features of the generated data. CLs are tested on GAN models and the results on a selection of tabular problems with constraints show how they are effective at preventing generation of data which violates such constraints.

**Strengths:**

The paper presents an effective way to enforce linear constraints on (and between) variables for tabular data generated with deep Generative models. The manuscript is generally very clear and presents the contributions in great detail. The method is tested on a comprehensive set of problems and compared with three strong baselines.

**Weaknesses:**

From reading the paper, it is not immediately clear how the training procedure with CLs works, and whether CLs can be applied to other generative models, such as Variational Auto Encoders, Normalizing Flows or Diffusion Models. A comparison with methods such as TVAE and TabDDPM (mentioned in the related work) and the application of CLs to them would significantly strengthen the experimental section. A metric to compare the data distribution from the generated distribution is missing from the experiments. Would it be possible to include a metric such as negative log-likelihood or Wasserstein distance?

**Questions:**

- Perhaps a point that I missed from the paper, but how do CLs affect the training? Can gradients backpropagate through these layers? Can CLs be applied to other Generative Models?
- How does the application of CLs shift the distribution of the generated data? Does it result in an "overpopulation" of the regions on the boundaries? If that's the case, the resulting distribution would be skewed from the true distribution especially when the baseline model generates many samples which violate the constraints. Adding other metrics (see weaknesses) would help with investigating this matter.
- Can CLs impose constraints between categorical variables, or between numerical and categorical variables? For example, if x1 = "category 1", then x2 > 5, or similar?
- Are there cases in which assigning a valid variable ordering is not feasible?

---

> ### Author Response · Authors · 2023-11-17
> **Response to Reviewer 72Ff [1/3]**
>
> We thank the reviewer for their feedback, which allowed us to study in more detail the behaviour of our constraint layer.
>
> > **Question:**
> > *How does the CL training procedure work? And how does CL affect the training?*
>
> **Answer:**
> The basic intuition behind our layer is very simple: given a set of constraints expressed as linear inequalities, we compile our constraints into a differentiable layer (CL), which then gets added to the topology of the network itself right after the DGM layer that generates the samples. At training time, the network is trained with CL injected in the topology and the gradients backpropagate seamlessly through it.  We have added a paragraph to explicitly state that the gradients can backpropagate through CL at the beginning of Section 3.
>
>
>
> > **Question:**
> > *Can CL be applied to other generative models?*
>
> **Answer:**
> Yes, we applied it to TVAE [1] and GOGGLE [2], and we added the results in Table 2. As can be seen from the results, adding CL at training time results is an improvement in all metrics but one for TVAE and in all metrics for GOGGLE. As we stated in the general answer, note that the results on GOGGLE are computed on 5 out of 6 datasets. We will update the results with the final values as soon as possible.

---

> > ### Author Response · Authors · 2023-11-17
> > **Response to Reviewer 72Ff [2/3]**
> >
> > > **Question:**
> > > *How does the application of CLs shift the distribution of the generated data? Does it result in an "overpopulation" of the regions on the boundaries? If that's the case, the resulting distribution would be skewed from the true distribution especially when the baseline model generates many samples which violate the constraints. Adding other metrics (see weaknesses) would help with investigating this matter.*
> >
> > **Answer:**
> > We thank the reviewer for this comment, which allowed us to study more in-depth the behavior of our C-DGMs at the boundaries.
> >
> > To conduct this study,  following the suggestion given by the reviewer and the approach of [3], we performed a comparative analysis of the distributions of data generated by the models under study and the real data by measuring the distance between the real data distribution and the generated samples' distribution. We used the Wasserstein distance for the continuous features and Jensen-Shannon divergence for categorical features. We report the results in Appendix C.2, in Tables 11 and 12. As we can see from the Tables, there is almost no difference in terms of Jensen-Shannon divergence (Table 12) between the samples generated by the DGMs, the P-DGMs, and the C-DGMs. The differences are more accentuated when studying the continuous features in terms of the Wasserstein distance, where the C-DGMs achieve the lowest distance in 12 cases (out of a total of 22 comparisons).
> >
> > While interesting, the above did not give us much insight into the behaviour of the models at the boundary. To specifically study this phenomenon, we did the following:
> > - We considered the datasets and constraints for which we plotted Figures 3, 4, 5, and 6. These examples are particularly interesting, because these were the constraints for which we got the highest number of violations, and thus we should expect the overpopulation problem (if we had it) to show its worst effects.
> > - We defined a band around the boundary whose width $w$ we set to be proportional to the range of the values of the considered features in the real dataset. As in all the considered constraints, only two features appear, we set $w = p\sqrt{r_1^2+r_2^2}$, where $r_1$ (resp. $r_2$) represents the range of the first (resp. second) feature, while $p$ represents the proportion of the range of values each feature can assume. For the sake of this study, we considered $p=0.01, p=0.05$ and $p=0.1$. For example, for the constraint *MaxHemoglobingLevel - MinHemoglobinLevel* $\ge 0$, we have that the feature *MaxHemoglobingLevel*'s values range between 6.8 and 17.2 (thus $r_1 = 10.4$), while feature *MinHemoglobingLevel*'s values range between 5.3 and 16.7 (thus $r_2=11.4$). The width of the band in this case for $p=0.01$ is thus $w \sim 0.154$.
> > - We then calculated the percentage of data points falling in the defined band for (i) the real dataset, (ii) the DGMs and (iii) the C-DGMs.
> > - We reported the results in Appendix C.1 in Table 10.
> >
> > As we can see from the results, there is no overpopulation happening. Indeed, for each example considered we have that in **every single case** the average number of samples generated by the C-DGMs falling in the band is closer to the number of real data points falling in the band than the number of samples generated by the standard DGMs.
> > This is due to the fact that the layer is added at training time, and thus the DGMs are aware of the presence of the layer (teaching the model to distance its outputs from the invalid regions).
> >
> > Notice that the same cannot be said when the layer is used at post-processing time. In this case, the DGMs do not have access to the background knowledge during training and thus all the data points for which the DGMs commit a violation will simply end up on the boundary. While we can argue that having points on the boundary is still better than having completely unrealistic data points, and the results are still very good for a post-processing step, it is also interesting future work how to create post-processors (that thus cannot be added at training time) that do not suffer from this shortcoming.

---

> > > ### Author Response · Authors · 2023-11-17
> > > **Response to Reviewer 72Ff [3/3]**
> > >
> > > > **Question:**
> > > > *Can CLs impose constraints between categorical variables, or between numerical and categorical variables? For example, if x1 = ``category 1'', then x2 $>$ 5, or similar?*
> > >
> > > **Answer:**
> > > CL can impose constraints expressed as systems of linear inequalities. As reviewer DAiC pointed out, and as we have added in Section 2, linear constraints enjoy multiple desirable properties that make them an interesting class of constraints. Among such properties there is the fact that systems of linear inequalities always define a convex space. We use this assumption in Equation (2), where we define the lower and upper bound associated to each variable. Notice that if we allow for constraints of the type ``if $x_1 =$ *'category 1', then* $ x_2 > 5$'', then they would no longer define a convex space, and thus our method would not be applicable. Additionally, adding such constraints would ultimately correspond to allowing for disjunctions in the constraints and, as far as we know, it is still an open problem how to compile such constraints in a backtrack-free tree representation (see comments to Reviewer ikQ7).
> > >
> > >
> > >
> > > > **Question:**
> > > > *Are there cases in which assigning a valid variable ordering is not feasible?*
> > >
> > > **Answer:**
> > >  All variable orderings are always allowed, with some leading to better performance than others (as shown in Appendix C.3).
> > >
> > >
> > > **References:**
> > >
> > >
> > > [1] Lei Xu, Maria Skoularidou, Alfredo Cuesta-Infante, and Kalyan Veeramachaneni. Modeling tabular data using conditional GAN. In Proceedings of Neural Information Processing Systems, 2019.
> > >
> > > [2] Tennison Liu, Zhaozhi Qian, Jeroen Berrevoets, and Mihaela van der Schaar. GOGGLE: Generative modelling for tabular data by learning relational structure. In Proceedings of International Conference on Learning Representations, 2022.
> > >
> > > [3] Akim Kotelnikov, Dmitry Baranchuk, Ivan Rubachev, and Artem Babenko. TabDDPM: Modelling
> > > Tabular Data with Diffusion Models. In Proceedings of International Conference on Machine
> > > Learning, 2023

---

> ### Comment · Reviewer_72Ff · 2023-11-21
>
> I thank the authors for answering my questions and adding extensive experiments and results to the paper. While I believe that the quality and clarity of the paper have already improved, there are still a couple of points that remain unclear to me.
> - From the Wasserstein distance results, it seems like the C-DGMs can sometimes have higher distances compared to non-constrained models, which would suggest that the learned distribution becomes worse. In addition, sometimes the distance is also greater than the one obtained with the P-DGMs, which to me is counterintuitive given the considerations made in your answer that when the CL is added at training time, "the DGMs are aware of the presence of the layer, teaching the model to distance its outputs from the invalid regions". Do you have some insight on why in some cases the Wasserstine distance increases for C-DGMs?
> - From answer 3, do I understand correctly that CL cannot be applied to categorical features? Could you please clarify this point, and explain the differences in the Jensen-Shannon divergence between categorical features in Table 12?

---

> ### Author Response · Authors · 2023-11-22
>
> First of all, we would like to thank you for answering our comments and appreciating the efforts we have made to extend the experimental analysis and improve the clarity of the paper.
>
> - Regarding the analysis of the results obtained using the Wasserstein distance (WD), some considerations are in order. Indeed, in the experiment conducted to study the overpopulation problem, we computed the Wasserstein distance following the approach of Zhao et al. 2021 [1] (then replicated by Kotelnikov et al. [2]), which consists in considering every feature separately, computing the Wasserstein distance for that feature, and finally averaging over the values obtained for all features. We then saw that the results were inconclusive (as we were doing better/worse $\sim 50$% of the times) and thus focussed on how to show that our models do not suffer from the overpopulation problem. Upon a closer look though, we realised that while this approach might have been sensible for the data considered in [1,2], it is not in our case. Indeed, our datasets contain features ranging from $0$ to $1$, as well as features spanning from $10^3$ to $10^7$. This entails that our results were solely driven by the features which have a high magnitude (as an example, notice that $WD([0, 1, 3], [5, 6, 8]) = 5$, while $WD([0, 1, 3]*10, [5, 6, 8]*10)=50$). We thus thank the reviewer for asking about this, as it allowed us to look deeper into the metric computation. To overcome this issue, we decided to first apply mix-max scaling to all the features, and then repeat the procedure outlined above. The newest results are in Table 11, where we can see that the differences are very small with only one big gap equal to $0.17$, as C-GOGGLE (resp. GOGGLE) obtains Wasserstein distance equal to $0.05$ (resp. $0.22$) on the WiDS dataset. Additionally, we can see that the biggest "negative" gap (meaning that a DGM obtains a lower distance than its corresponding C-DGM) is equal to $0.05$ and is obtained by GOGGLE on the News dataset. While this might seem to suggest that the distribution produced by C-GOGGLE is worse, we can clearly see that C-GOGGLE does better in terms of utility (according to both XV and MAE). This shows that, while the Wasserstein distance is an indicator of the quality of the captured distribution, it is not by any means an exhaustive metric. This makes sense, as in this metric each feature is considered separately and, thus, it cannot capture the correlations existing among features.
>
>
> - Regarding the Jensen-Shannon divergence, we updated the results in Table 12. Notice that the C-DGM model is a different model from the DGM one, and thus can learn a different distribution from the initial DGM even for the categorical features. On the other hand, the P-DGM is exactly the same model as the DGM with the constraint layer added at inference time, and indeed we get exactly the same values as the DGMs (in our dataset we do not have any constraint over the categorical features).
>
>
> - Regarding the ability of writing constraints over categorical features, we can only constrain the continuous values outputted by the neural networks. For example, given two DGM outputs $x_1, x_2 \in [0,1]$, we can write the constraint $x_1 > x_2$ expressing that if $x_2$ is positively predicted w.r.t. some given threshold, then $x_1$ should also be positively predicted. How to integrate constraints expressed in, e.g. propositional logic, surely represents interesting future work.
>
>
> *References*
>
> [1] Zhao, Z., Kunar, A., Birke, R., and Chen, L. Y. Ctab-gan: Effective table data synthesizing. In Asian Conference on Machine Learning, pp. 97–112. PMLR, 2021.
>
> [2] Akim Kotelnikov, Dmitry Baranchuk, Ivan Rubachev, and Artem Babenko. TabDDPM: Modelling Tabular Data with Diffusion Models. In Proceedings of International Conference on Machine Learning, 2023.

---

> > ### Comment · Reviewer_72Ff · 2023-11-23
> >
> > I thank the authors for adjusting the results and adding more evaluation metrics, as well as for further clarifying the capabilities of the CL layers. I see how the proposed method solves the problem of respecting linear constraints in tabular data, and how this can be useful in practice when respecting such constraints is critical. However, I believe that from the experimental section, it is not clear whether the generative performance, in terms of learning the correct data distribution, is improved or worsened with respect to the non-constrained models.
> > Perhaps testing the method on more complex datasets could shed some light on whether the CL layers provide a substantial improvement, as from the current results, the method does not always outperform the non-constrained models, and most of the time only marginally, with respect to the metrics considered (besides constraints violation coverage). I think it is very important to find a way to show the faithfulness of the learned distribution with respect to the true one and to show that CL layers do improve over standard models. Otherwise, as a practitioner, I would rather naively generate data with a non-constrained model, and discard those samples that violate my constraints, until I obtain the desired amount of samples.

---

> > > ### Author Response · Authors · 2023-11-23
> > >
> > > We thank the reviewer for the answer and for engaging in the discussion. However, we would like to point out that there are cases where the current standard approach of discarding the samples that violate the constraints might be **impossible**. Indeed, even in our experimental analysis, we had one case where $100$% of the samples generated by the standard model violated the constraints and multiple cases where $99$% of the samples violated the constraints. Notice that our datasets are annotated with up to 31 constraints. We can envision that in practice the number of constraints can dramatically increase, and with it the percentage of samples violating the constraints. Also, we have clear improvements with respect to utility and detection which are commonly used metrics in the field (notice that in this field the state-of-the-art progresses only by small successive improvements, see, e.g., [1,2]).  We thus politely ask the reviewer to reconsider our contribution in light of these facts.
> > >
> > >
> > > **References:**
> > >
> > > [1] Tennison Liu, Zhaozhi Qian, Jeroen Berrevoets, and Mihaela van der Schaar. GOGGLE: Generative modelling for tabular data by learning relational structure. In Proceedings of International Conference on Learning Representations, 2022.
> > >
> > > [2] Jayoung Kim, Chaejeong Lee, and Noseong Park. STaSy: Score-based Tabular data Synthesis. In Proceedings of International Conference on Learning Representations, 2023.

---

> > > > ### Comment · Reviewer_72Ff · 2023-11-23
> > > >
> > > > I thank the authors for their answers and clarifications. I agree that for a high number of constraints, it might be infeasible for generative models to respect all of them at the same time. After careful consideration, I decided to raise my score. However, I think the paper would improve significantly if the aforementioned considerations about data distribution were to be discussed in the manuscript.

---

### Official Review · Reviewer_ikQ7 · 2023-11-01

**Soundness:** 1 poor
**Presentation:** 4 excellent
**Contribution:** 1 poor
**Rating:** 5
**Confidence:** 3

**Summary:**

This paper investigates a lesser-explored challenge in the generation of tabular data—adherence to specific rules or constraints for data entries. It highlights a common issue where existing generative models often fail to respect constraints such as linear inequalities (e.g., one column being less than or equal to another), as they are not designed to fulfill these conditions.

To address this, the paper introduces a novel approach to adding a constraint layer to standard DGMs, transforming them into constrained DGMs. The study demonstrates that these constrained DGMs are more effective in producing realistic data that adheres to the specified constraints, further improving downstream performance. Moreover, it shows that even applying these constraint satisfaction layers to a pre-trained DGM can significantly enhance the realism of the generated data.

While the paper tackles an interesting issue, given the marginal novelty of the paper and the limited number of constraints in the datasets under consideration, I will not accept this paper. I would be willing to increase the score if the authors can provide reasonable answers to my concerns.

**Strengths:**

1) The paper is well-written and the idea is laid out with sufficient examples to follow through.
2) Pointing out the fact that many DGMs violate obvious constraints in tabular data is very important and studying it is valuable.
3) The experimental results seem thorough and the theory checks out.

**Weaknesses:**

1) **(Important)** The paper mentions the choice of $\lambda(i) = i$ is for ease of notation, while different orderings can drastically change the performance of a C-DGM. As an illustrative example, consider tabular data with three outputs $\langle x_1, x_2, x_3 \rangle$ with constraints: $x_2 \le 10$ and $x_1 \le x_2 \le x_3$. Now assume the generative model only produces $\langle 10, 5, 5 \rangle$. In this case, if the ordering $\lambda = \langle 1, 2, 3 \rangle$ is considered, then $CL(\tilde{x})$ would be $\langle 10, 10, 10 \rangle$ and if the ordering $\lambda = \langle 2, 1, 3 \rangle$ is considered, then $ CL(\tilde{x}) = \langle 5,5,5 \rangle$ with one being significantly closer to the generated distribution than the other. In fact, the notion of “optimality” is not well defined, as each ordering can produce a different optimal with none of them being comparable for defining an optimum. Moreover, the notion of optimality should also consider the discrepancy between the $CL(\cdot)$ outputs and the generative samples to be minimal.
2) **(Important)** I might have missed something but the reduction defined for constraints $\Pi$ can easily turn the number of constraints exponential, meaning that $|\Pi_1| \in \mathcal{O}(exp(|\Pi|))$. The only reason the current experiments do not hinder the performance is that the number of conditions is fairly small, to begin with. I would require datasets with a much larger number of conditions to be convinced that the post hoc method does not impact the sample generation time.
3) The paper only considers linear constraints. Even though it is pointed out as a limitation some extensions to non-linear constraints are quite simple. For example, by introducing polynomial features, one can add polynomial constraints to the current approach. Having one entire paper on linear constraints seems rather limited in novelty. I would suggest adding simple experiments as proof-of-concept for such extensions.

**Questions:**

1) Even though GANs have achieved popularity for image generation, they are known to fail drastically for tabular data generation. That said, is there any reason why other DGMs such as TVAEs, STaSy, and TabDDPM are not considered in this study given the current limitations of GANs? For a more compelling story, it would be good to include other types of generative models as well. Especially in the post hoc experiments.

2) The reported charts and tables are thorough, but I wasn’t able to find any source code for reproducibility and a footnote claims that the code will be released upon publishing; however, I didn’t find anything to run in the supplementary material.

---

> ### Author Response · Authors · 2023-11-17
> **Response to Reviewer ikQ7 [1/3]**
>
> We thank the reviewer for their feedback. Below are detailed answers to all their comments.
>
> > **Question:**
> > *The paper mentions the choice of $\lambda(i) = i$ is for ease of notation, while different orderings can drastically change the performance of a C-DGM. [...] In fact, the notion of "optimality" is not well defined, as each ordering can produce a different optimal with none of them being comparable for defining an optimum. Moreover, the notion of optimality should also consider the discrepancy between the $CL(\cdot)$ outputs and the generative samples to be minimal.*
>
>
> **Answer:**
> We articulate our answer in three points:
> - In a tabular dataset, the features are not ordered (contrarily to e.g., natural language datasets), and thus we can associate e.g., the variable $x_0$ to any feature. Thus, our choice of notation does not affect our results in any way.
> -  The optimality is well-defined and takes into account the discrepancy between the final CL outputs and the generative samples. Let us consider the example proposed by the reviewer. As it can be seen, both the final outputs $\langle 10, 10, 10 \rangle$ and $\langle 5,5,5 \rangle$ are optimal according to our definition. Indeed, for both the final outputs there does not exist another sample such that (i) it satisfies the constraints and (ii) for each $i=1,2,3$ it associates a value closer to the original output than CL.     The very fair question the reviewer asks is essentially why are both final outputs considered optimal. To answer this question, suppose that we are given a DGM that is not able to capture the distribution of the features $x_2$ and $x_3$, while it is particularly good at capturing the distribution of $x_1$. In this scenario, the output $\langle 10, 10, 10 \rangle$ is much more desirable than $\langle 5,5,5 \rangle$. Thus, ruling out a priori the solution $\langle 10, 10, 10 \rangle$ as ``non-optimal'' would be a mistake. The fact that closeness to the original output is not the only factor to take into account when correcting a deep learning model has already been shown in the neuro-symbolic community (see e.g., [1]), where it has been shown that selecting the closest solution leads to worse performance than selecting the solution taking into account the confidence in the predictions.  We understand though that this might be counter-intutive, and for this reason we have added a paragraph in Section 3 (right above Theorem 3.5).
> -  As a consequence of the point above, in Appendix C.3 a study on the impact of the variable orderings can be found, where we also propose two heuristics to decide favourable variable orderings. We could not include such a discussion in the main paper due to the lack of space. A clear reference to this section has been added to the above-mentioned paragraph that has been included in the main body of the paper.

---

> > ### Author Response · Authors · 2023-11-17
> > **Response to Reviewer ikQ7 [2/3]**
> >
> > > **Question:**
> > > *I might have missed something but the reduction defined for constraints $\Pi$ can easily turn the number of constraints exponential, meaning that $|\Pi_1| = O(exp(|\Pi|)$ . The only reason the current experiments do not hinder the performance is that the number of conditions is fairly small, to begin with. I would require datasets with a much larger number of conditions to be convinced that the post hoc method does not impact the sample generation time.*
> >
> > **Answer:**
> >
> > The reviewer is right in saying that the algorithm is exponential in the number of constraints. **However**, even though the compiled representation of the constraints, in the worst case, can have an exponential size in $|\Pi|$, such representation allows us to correct the prediction with a single forward pass both at training and at inference time. Further, the representation has a depth equal to the number of features, and it is possible to parallelize the operations made at each level (as they are $max$, $min$, etc.). Given the above, it should be clear that we designed our method keeping in mind the specific requirements that arise when working with neural networks.
> > Indeed, notice that the above considerations match the ``Efficiency" requirement explicitly stated in [2], whose approach (which also embeds given constraints into the topology of standard neural networks) may lead to an exponential structure but can compute a prediction in linear time in the size of the predictor. Two last observations:
> > - The exponential blow-up happens in the worst case, and indeed it happens when the constraints have many variables in common.
> > - In many cases the constraints are elicited directly from domain experts and/or from the underlying physics, and thus it is extremely unlikely for them to be deeply intertwined and involve very high numbers of variables.
> >
> > In practice, in all the domains that we considered (which have up to 31 constraints, each including up to 17 different features), we did not experience any problem with the constraints' representation, and the overhead introduced by the constraint layer was always negligible.
> >
> >
> >
> > > **Question:**
> > > *The paper only considers linear constraints. Even though it is pointed out as a limitation some extensions to non-linear constraints are quite simple. For example, by introducing polynomial features, one can add polynomial constraints to the current approach. Having one entire paper on linear constraints seems rather limited in novelty. I would suggest adding simple experiments as proof-of-concept for such extensions.*
> >
> > **Answer:**
> > We thank the reviewer for raising this point, which allows us to discuss the many amenable properties of linear inequalities and the difficulty of extending this line of work to constraints involving polynomial inequalities.
> >
> > Regarding the positive properties of linear constraints:
> > - Linear constraints are already very expressive and are able to capture many different relations among the features. The importance of this class of constraints is underlined by the existence of entire fields dedicated to its study (see, e.g., linear programming).
> > - As pointed out by Reviewer DAiC, the unsatisfiability of linear constraints can be decided in polynomial time using Farkas' lemma [3].
> > - As shown in [4], it is possible to compile the linear inequalities in our tree representation of the constraint, which allows for a backtrack-free computation of a prediction "close" to the original one.
> > - A system of linear inequalities always defines a convex space, a property that we exploit in Equation (2), in order to compute a unique lower and upper bound.
> >
> > We added a discussion on these properties in Section 2 below Example 2.1.
> >
> > Regarding the difficulty of extending our work to polynomial constraints:
> > - Even in the case of a single variable, it is not possible to find solutions in radicals for polynomials of degree higher than 4 (Abel-Ruffini theorem),
> > - To the best of our knowledge, no data structures allow for a backtrack-free representation (in the sense outlined above) of constraints with polynomials,
> > - Polynomials allow defining a non-convex space of solutions (e.g., $x^2 - 1 \geq 0$).

---

> > > ### Author Response · Authors · 2023-11-17
> > > **Response to Reviewer ikQ7 [3/3]**
> > >
> > > > **Question:**
> > > > *Even though GANs have achieved popularity for image generation, they are known to fail drastically for tabular data generation. That said, is there any reason why other DGMs such as TVAEs, STaSy, and TabDDPM are not considered in this study given the current limitations of GANs? For a more compelling story, it would be good to include other types of generative models as well. Especially in the post hoc experiments.*
> > >
> > > **Answer:**
> > > We added our constraint layer to TVAE and included in the revised paper an experimental study for the post-hoc case, but also for the case where our layer is added at training time.
> > > The new results are in Tables 2 and 3. As it can be seen from Table 2, adding our layer at training time improves the utility according to all metrics and the detection according to 2 out of 3 metrics. On the other hand, adding it at post-processing time results in a slight decrease in the performance according to all metrics but one, thus underlying the importance of using our layer at training time.
> > >
> > >
> > > > **Question:**
> > > > *The reported charts and tables are thorough, but I wasn't able to find any source code for reproducibility and a footnote claims that the code will be released upon publishing; however, I didn't find anything to run in the supplementary material.*
> > >
> > > **Answer:**
> > > We are currently trying to anonymize our code and share a link with the reviewers (as we gave priority to running all the requested experiments). We will share a link as soon as possible.
> > >
> > > **References:**
> > >
> > > [1] Eleonora Giunchiglia, Mihaela Catalina Stoian, Salman Khan, Fabio Cuzzolin, and Thomas
> > > Lukasiewicz. ROAD-R: The autonomous driving dataset for learning with requirements. Machine Learning Journal, 2023
> > >
> > > [2] Kareem Ahmed, Stefano Teso, Kai-Wei Chang, Guy Van den Broeck, and Antonio Vergari. Semantic probabilistic layers for neuro-symbolic learning. In Proceedings of Neural Information
> > > Processing Systems, 2022.
> > >
> > > [3] Julius Farkas. Theorie der einfachen ungleichungen. Journal fur die reine und angewandte Mathematik, 124, 1902.
> > >
> > > [4] Rina Dechter. Bucket elimination: A unifying framework for reasoning. Artificial Intelligence, 113, 1999.

---

> > > > ### Author Response · Authors · 2023-11-21
> > > > **Follow-up on sample generation time**
> > > >
> > > > To follow up on Reviewer ikQ7's request to provide an analysis on how the sampling time is affected by our layer when a dataset with a large number of constraints is considered, we provide the runtimes on the Botnet dataset [1]. Botnet is an atypical dataset for the tabular data generation task due to its high number of features (it has 757 features).
> > > > Indeed, as stated in prior work [2], the typical datasets considered for this task have dimensionality $d < 100$.
> > > > However, Botnet provides the ideal setting to test our layer's runtime efficiency, as it comes with **362 linear constraints**, where the maximum number of features appearing in a single constraint is 90.
> > > >
> > > > We measured the generation time of 1000 samples, for the three original models we considered in our paper. Note that for the constrained versions we always used a random ordering.
> > > > Below we report the average results (in seconds) over 5 runs and their standard deviation:
> > > > - CTGAN: unconstrained 1.34 $\pm$ 0.02, constrained 1.73 $\pm$ 0.12,
> > > > - WGAN: unconstrained 0.03 $\pm$ 0.00, constrained 0.59 $\pm$ 0.10,
> > > > - TableGAN: unconstrained 11.50 $\pm$ 3.34, constrained 11.88 $\pm$ 4.63.
> > > >
> > > > As we can see from the results, even in such a complex scenario, our constraint layer results in an overhead that ranges **between 0.38 and 0.56 seconds**.
> > > >
> > > > **References**
> > > >
> > > > [1] Alesia Chernikova and Alina Oprea. Fence: Feasible evasion attacks on neural
> > > > networks in constrained environments. arXiv preprint arXiv:1909.10480, 2019.
> > > >
> > > > [2] Tennison Liu, Zhaozhi Qian, Jeroen Berrevoets, and Mihaela van der Schaar. GOGGLE: Generative modelling for tabular data by learning relational structure. In Proceedings of International Conference on Learning Representations, 2022.

---

> ### Comment · Reviewer_ikQ7 · 2023-11-22
> **Thanks for the reply**
>
> Thank you for the comprehensive response and the additional analysis provided. The inclusion of the analysis on TVAE and GOGGLE is particularly appreciated and it coupled with the addition of the Botnet experiments has positively influenced my evaluation. While I am inclined to adjust my score upwards, I still have some reservations. Addressing these issues could significantly enhance the paper's robustness, at least from a theoretical point of view. I encourage a deeper exploration of these aspects in your future revisions.
>
> * **About the ordering**: I concur that tabular data inherently lack an ordered structure, yet they often exhibit an underlying causal sequence reflective of their data generation process. For example, selecting an order precisely opposite the causal direction would presumably yield non-realistic results. The distinction between correlation and causation is crucial here; choosing an order based solely on correlation seems impractical (the first heuristic), and I foresee scenarios where this method could fail. On the other hand, the KDE-based technique, as it stands, appears somewhat arbitrary. Although Table 13 shows marginal improvements, a more systematic method for determining this ordering would be preferable. Furthermore, I appreciate the additional material that clarifies the limitations of using 'closeness' to model-generated samples as a measure of output realism. While I agree that proximity in terms of some distance in a normed vector space to the model's output does not guarantee realism, the impact of different orderings on the output cannot be overlooked. There's a clear need for a more systematic approach to determine this ordering.
>
> * **About exponential blow-up**: Based on the explanation provided, I am now inclined to believe that the issue of exponential blow-up, as described, is unlikely to occur in practical scenarios. The evidence presented in the current literature, especially the insights from the botnet experiment, has effectively addressed my initial concerns in this regard. However, from a theoretical perspective, I still find the idea of exploring more efficient sampling mechanisms from the polyhedra induced by linear constraints to be a compelling avenue for research. Is there any existing literature on writing the constraints as an optimization problem? For example, an objective that maximizes when the data is "realistic" and some linear constraints showing the constraints that should be satisfied from our knowledge base? If one can linearize the objective, then this would be an LP and solving it would not take exponential time.
>
> * **About linear constraints**: The revised manuscript primarily highlights the advantages of linear constraints, emphasizing their ease of management. I appreciate the authors' inclusion of an example demonstrating how polynomial constraints are not necessarily convex, which adds valuable context to the discussion. While I acknowledge the utility of Farkas' lemma in the world of linear constraints, the reality is that not all constraints we encounter are linear. A potentially interesting approach to circumvent the linear assumption might be to employ an invertible neural network. This network could project data into a space where non-linear constraints are linear, apply the post-hoc method, and subsequently map the data back to its original space. However, the current argument presented by the authors seems to narrowly focus on the convenience of linear constraints without adequately addressing the occurrence and implications of non-linear constraints. This gap in the argumentation leaves room for further exploration and justification regarding the choice of constraints in various scenarios.

---

> ### Author Response · Authors · 2023-11-23
> **Thank you for your reply [1/2]**
>
> We thank the reviewer for the response and for the discussion, which has sparked many ideas about possible directions for future work.
>
> - **Regarding the variable ordering:** We absolutely agree that the ordering is critical. We also thank the reviewer for the suggestion of using the causal information to decide the ordering, which would allow us to compute a favourable ordering without the need for first training an unconstrained model, thus making our method even more convenient and usable for a end-user. However, our intuition is that, if it is possible to first train an unconstrained DGM, then an ordering based on the ability of the model itself to learn the distribution of each feature would probably lead to better results. As an example, consider a DGM that has to learn the distribution of some data collected during a clinical trial which includes the features "High Blood Pressure" and "Heart Attack". In this scenario, we know that there exists a causal relation between these two features and thus a causality-inspired ordering would first consider the "High Blood Pressure" feature and then the "Heart Attack" feature. Now, suppose that the DGM approximates very well the distribution for the feature "Heart Attack" but poorly the feature "High Blood Pressure", then in this scenario it would probably lead to better results to consider the "Heart Attack" feature first. Given this example, an interesting question that arises is whether there is any relation between the causal relationships existing among features and the difficulty of learning them with a DGM. As the above conjectures make very interesting problems to be studied, we agree with the reviewer that a more systematic study on the variable orderings is necessary. For this reason, we will do the following:
>     - We will consider an ordering based on the causal relationships among the features. Unfortunately, none of our datasets comes with such information. However, we can use some of the existing algorithms for causal relation extraction (e.g., the PC algorithm and the package pcalg: https://cran.r-project.org/web/packages/pcalg/index.html) to first extract the causal relations and then use this information to calculate the ordering.
>     - We will consider an ordering based on the Wasserstein distance computed for each feature between the real data distribution and the distribution of the samples obtained with the unconstrained DGMs.
>     - We will better explain the orderings already proposed and put them in context with the newly considered orderings.
>
>   Unfortunately, in the little time remaining before the end of the discussion period, we cannot update the paper with the results above. We will do so in the final version.
>
> - **Regarding the exponential blow-up:** We are very happy that we agree on the issue of exponential blow-up. We also agree on the fact that there are many interesting ways of formulating the problem we studied and we cannot wait to see how the field will develop. For example, in [2] the authors study how to constrain the generation of adversarial attacks by embedding the constraints in the loss function. This formulation is interesting because it does not suffer from the issue of the exponential blow-up. However, notice that by formulating the problem in this way, the authors lose the ability of guaranteeing the satisfaction of the constraints, which is one of the strength of our proposed method. On the other hand, we are not aware of any work where the authors managed to linearize the objective that maximizes when the data is "realistic" while having some linear constraints to satisfy. We again agree with the reviewer that this could be an interesting direction to explore.
>
> - **Regarding the linear constraints:** Certainly there are cases in which there are interesting constraints that are not linear and are polynomial or even transcendental. Even though in some of such cases the complexity of deciding the satisfiability of the constraints may greatly jump up (up to become undecidable), we acknowledge that there are cases in which a higher complexity might not be a problem, e.g., because there are no stringent requirements on the time necessary to produce the output. In any case, the suggested idea of using invertible neural networks seems applicable to a broad range of scenarios and interesting for future work. In our current settings, non-linear constraints did not arise and the time to produce the output is indeed considered an important metric according to which the DGMs are evaluated.

---

> ### Author Response · Authors · 2023-11-23
> **Thank you for your reply [2/2]**
>
> To conclude, we thank the reviewer for these comments which will surely further improve our paper. The insightful discussion we had in the past days highlights the importance of the problem handled and the many directions this formulation of the problem of tabular data generation (where the methods not only have to approximate a distribution but also be compliant with a set of constraints) opens up. We see this paper as the first stepping stone in this direction, and we believe that many people will take up the challenge and build upon it.
>
> **References:**
>
> [1] Peter Spirtes, Clark N Glymour, Richard Scheines, and David Heckerman. Causation, prediction, and search. MIT press, 2000.
>
> [2] Thibault Simonetto, Salijona Dyrmishi, Salah Ghamizi, Maxime Cordy and Yves Le Traon. A Unified Framework for Adversarial Attack and Defense in Constrained Feature Space. Proceedings of IJCAI, 2022.

---

### Official Review · Reviewer_Yiht · 2023-11-01

**Soundness:** 3 good
**Presentation:** 2 fair
**Contribution:** 4 excellent
**Rating:** 8
**Confidence:** 3

**Summary:**

Generating realistic tabular data requires compliance with constraints that encode essential background knowledge on the problem. In this paper, the authors address the limitation and show how deep generative models for tabular data can be transformed into constrained deep generative models, whose generative samples are guaranteed to be compliant with the given constraints. This is achieved by automatically parsing the constraints and transforming them into a constraint layer seamlessly integrated with the dgm. The authors shows the effectiveness of the proposed model with experiments on 6 datasets.

**Strengths:**

The first to handle with the constraints of tabular data, and the method addressed the problem well.

**Weaknesses:**

Tabular data synthesis methods used for the experiments are outdated. Please consider [1] GOGGLE, [2] GReaT, [3] STaSy, [4] CoDi, and [5] TabDDPM if possible.


[1] GOGGLE: Generative Modelling for Tabular Data by Learning Relational Structure, ICLR 2022

[2] Language Models are Realistic Tabular Data Generators, NeurIPS 2021

[3] Stasy: Score-based tabular data synthesis, ICLR 2022

[4] CoDi: Co-evolving Contrastive Diffusion Models for Mixed-type Tabular Synthesis, ICML 2023

[5] Tabddpm: Modelling tabular data with diffusion models, ICML 2023

**Questions:**

How was the performance improvement of the state-of-the-art methods after applying the proposed method?

---

> ### Author Response · Authors · 2023-11-17
>
> We thank the reviewer for their appreciation of our work and for their useful suggestions.
>
> > **Question:**
> > *Tabular data synthesis methods used for the experiments are outdated. Please consider [1] GOGGLE, [2] GReaT, [3] STaSy, [4] CoDi, and [5] TabDDPM if possible. [...] How was the performance improvement of the state-of-the-art methods after applying the proposed method?*
>
> **Answer:**
> We included two new models in our experimental results: the suggested model GOGGLE [1] and TVAE [2] (which was requested by both reviewers ikQ7 and 72Ff). The results for both models can be found in Table 2.
>
>
>
> Regarding GOGGLE, we reported the results for 5 out of 6 datasets in green, and we can see that our method gives an overall improvement in both utility and detection metrics, when using the constraint layer during training.
> We are currently running the last experiments on the remaining dataset, along with the post-processing experimental analysis.
>
> Regarding TVAE, we can see that adding our layer at training time improves the utility according to all metrics and the detection according to 2 out of 3 metrics.
>
>
> **References:**
>
> [1] Tennison Liu, Zhaozhi Qian, Jeroen Berrevoets, and Mihaela van der Schaar. GOGGLE: Generative modelling for tabular data by learning relational structure. In Proceedings of International Conference on Learning Representations, 2022.
>
> [2] Lei Xu, Maria Skoularidou, Alfredo Cuesta-Infante, and Kalyan Veeramachaneni. Modeling tabular data using conditional GAN. In Proceedings of Neural Information Processing Systems, 2019.

---

### Official Review · Reviewer_DAiC · 2023-11-07

**Soundness:** 3 good
**Presentation:** 3 good
**Contribution:** 3 good
**Rating:** 6
**Confidence:** 3

**Summary:**

The main contribution of this paper is to add constraints on generating synthetic data so that it is aligned with available background knowledge. The paper introduces constraint layers in order to enforce a set of linear constraints that encode the background knowledge. They also prove the correctness of the constraint layers introduced

**Strengths:**

The motivation behind the paper is clear and easy to follow. The paper also offers some theoretical justification for their method which makes their paper compelling. The proofs from what I can see are correct. The experiments are well formulated and support the claims of the paper.

**Weaknesses:**

The issue with the paper is the proofs can be difficult to follow. It is possible that the authors can spend more time rewording it to make it easier to flow. This also makes it a bit hard to flow and check.

**Questions:**

This isn't really a question but I think the use of linear constraints can be advantageous as they are logically complete. Any inconsistency can be identified easily through Farkas's lemma. I think the authors can potentially frame the use of linear constraints in a better light to highlight how its logical completeness can be useful when specifying background knowledge.

---

> ### Author Response · Authors · 2023-11-17
>
> We thank the reviewer for recognizing the clarity of the paper, its correctness,
> and the importance of the problem solved.
>
> > **Question:** *The issue with the paper is the proofs can be difficult to follow.
> > It is possible that the authors can spend more time rewording it to make it easier to flow. This also makes it a bit hard to flow and check.*
>
> **Answer:**
> We have reworded all the proofs to make them more readable.
>
>
>
> > **Question:** *I think the authors can potentially frame the use of linear constraints in a better light to highlight how its logical completeness can be useful when specifying background knowledge.*
>
> **Answer:** Thanks for pointing out  the logical completeness of linear inequalities, which further emphasizes
> the advantage of utilizing linear constraints in our work. We have added a paragraph in Section 2 (below Example 2.1) highlighting all the positive properties of linear inequalities that we exploit in our paper.

---

### Author Response · Authors · 2023-11-17
**General Answer**

We thank the reviewers for their comments and their feedback. We have updated the paper (in blue) with the results of the new experiments and the requested clarifications. We list below the major updates to the paper:

-  We tested our solutions on **two additional deep generative models**, namely, TVAE [1] and GOGGLE [2], showing that our constraint layer improves the quality of the generated samples in both cases. The results for GOGGLE are in green, as we completed the experiments on 5 out of 6 datasets. We will add the results for the remaining dataset and for the post-processing analysis as soon as possible.
- We discussed the advantageous properties of considering constraints expressed as linear inequalities, which allow us to seamlessly integrate our constraint layer into the deep generative models.
- We further investigated our method's properties by studying the behaviour of the generated data distribution at the boundaries (defined by the constraints) and by measuring the similarity between the generated and real data distributions.

**References:**

[1] Lei Xu, Maria Skoularidou, Alfredo Cuesta-Infante, and Kalyan Veeramachaneni. Modeling tabular data using conditional GAN. In Proceedings of Neural Information Processing Systems, 2019.

[2] Tennison Liu, Zhaozhi Qian, Jeroen Berrevoets, and Mihaela van der Schaar. GOGGLE: Generative modelling for tabular data by learning relational structure. In Proceedings of International
Conference on Learning Representations, 2022.

---

> ### Author Response · Authors · 2023-11-23
> **Final General Answer**
>
> As the rebuttal period draws to a close, we express our appreciation for the constructive dialogue with the reviewers. Their valuable feedback has significantly shaped the evolution of our paper, resulting in what we believe to be a more comprehensive and improved version. All recent modifications, including those initially highlighted in green, are now marked in blue. Notably, we have completed the experiments for the GOGGLE model.
> The thoughtful comments provided by the reviewers had not only motivated us to
> refine our current work, but have also sparked ideas for potential future improvements, which we plan to explore in the post-rebuttal period.
> As a closing remark, we would like to emphasize that our paper marks an initial step towards generating synthetic tabular data that guarantees the satisfaction of background knowledge expressed as linear constraints, a crucial aspect in diverse applications.
> We envision that such practices will become common in future research within this domain. Furthermore, we anticipate that in the forthcoming works we will broaden the range of background knowledge types supported by our constraint layer integrated into deep generative models.

---

### Meta-Review · Area_Chair_Jb9u · 2023-12-09

**Metareview:**

This paper addresses the challenge of ensuring compliance with constraints when generating realistic synthetic tabular data using Deep Generative Models (DGMs). To achieve this, the authors propose Constrained Deep Generative Models (C-DGMs), integrating Constraint Layers (CL) into DGMs to guarantee generated samples adhere to specified constraints. Experimental results demonstrate that while standard DGMs often violate constraints, C-DGMs consistently maintain compliance, showcasing up to a 4.5% improvement in utility and detection by leveraging background knowledge expressed through constraints during training.

Four reviewers evaluated the paper, with three recommending acceptance and one suggesting a borderline rejection. All reviewers agreed that the generative models satisfying given constraints in tabular data generation is a timely and crucial research area. They also concurred that the paper, excluding some proofs, is easy to follow and presents convincing ideas. In the final version, discussing potential limitations such as 'ordering,' 'exponential blow-up,' and 'linear constraint' from the review would further enhance the paper's quality. In the discussions between the AC and reviewers, the method of discarding samples that do not satisfy constraints after generating with vanilla DGM, akin to rejection sampling, was discussed. While reviewers expressed concerns about high rejection rates of such method on specific datasets leading to prolonged generation times, they agreed that including more such baselines (naive rejection sampling and more sophisticated alternatives such as Metropolis-Hastings algorithms capable of handling high constraint violation rates more effectively) could further elevate its quality.

**Justification For Why Not Higher Score:**

to achieve such quality (as usual spotlight papers), including more baselines as mentioned in the ac-reviewer discussions and meta-review need to be added.

**Justification For Why Not Lower Score:**

all reviewers agreed that the paper is well written and ideas are convincing. the submission seems to be worth publishing.

---

### Decision · Program_Chairs · 2024-01-16

Accept (poster)